# MECHANISM OF TASK-ORIENTED INFORMATION REMOVAL IN IN-CONTEXT LEARNING

**Hakaze Cho**[1,2,3]  **Haolin Yang**[4]  **Gouki Minegishi**[5]  **Naoya Inoue**[3,1]

[1]RIKEN  [2]Tohoku University  [3]JAIST  [4]University of Chicago  [5]University of Tokyo

 Verb_subspace  yufeng.zhao@riken.jp  haolinyang2001@uchicago.edu
minegishi@weblab.t.u-tokyo.ac.jp  naoya-i@jaist.ac.jp

## ABSTRACT

**In-c**ontext **L**earning (ICL) is an emerging few-shot learning paradigm based on modern **L**anguage **M**odels (LMs), yet its inner mechanism remains unclear. In this paper, we investigate the mechanism through a novel perspective of information removal. Specifically, we demonstrate that in the zero-shot scenario, LMs encode queries into non-selective representations in hidden states containing information for all possible tasks, leading to arbitrary outputs without focusing on the intended task, resulting in near-zero accuracy. Meanwhile, we find that selectively removing specific information from hidden states by a low-rank filter effectively steers LMs toward the intended task. Building on these findings, by measuring the hidden states on carefully designed metrics, we observe that few-shot ICL effectively simulates such task-oriented information removal processes, selectively removing the redundant information from entangled non-selective representations, and improving the output based on the demonstrations, which constitutes a key mechanism underlying ICL. Moreover, we identify essential attention heads inducing the removal operation, termed Denoising Heads, which enables the ablation experiments blocking the information removal operation from the inference, where the ICL accuracy significantly degrades, especially when the correct label is absent from the few-shot demonstrations, confirming both the critical role of the information removal mechanism and denoising heads.

## 1 INTRODUCTION

**In-c**ontext **L**earning (ICL) is a promising application of modern **L**anguage **M**odels (LMs), in which a sequence of input-label pairs (demonstrations) concatenated with a query is fed to the LMs to predict the query label. However, the inner mechanism of ICL remains unclear, despite some progress (Zhou et al., 2024), including: linking ICL to specific input components (Min et al., 2022; Yoo et al., 2022; Pan, 2023; Kossen et al., 2024), relating ICL-style inputs to pre-training data (Li & Qiu, 2023; Gu et al., 2023; Han et al., 2023b; Li et al., 2024; Cho et al., 2025b), or analogy to simpler algorithms (Zhang et al., 2023; Dai et al., 2023; Xie et al., 2021; Han et al., 2023a).

Among prior work, mechanistic interpretability studies (Elhage et al., 2021; Chan et al., 2022; Reddy, 2024; Wang et al., 2023; Singh et al., 2024; Cho et al., 2025a; Yang et al., 2025; Minegishi et al., 2025; Yin & Steinhardt, 2025; Bakalova et al., 2025) reduce ICL to functional LM components (e.g., attention heads), capturing many phenomena and establishing causal roots to ICL accuracy. Mainstream interpretations attribute ICL to Induction Heads (Elhage et al., 2021; Reddy, 2024; Cho et al., 2025a; Yang et al., 2025) (introduced in §2), which copy the most relevant label token from demonstrations to the outputs. However, such explanations fail in *unseen label scenarios* (Fig. 15), i.e., the correct label is absent (Minegishi et al., 2025; Cho et al., 2025a) where ICL accuracy drops but remains well above the zero-shot level (Table 6), implying additional supporting mechanisms.

Therefore, we propose a new perspective to interpret ICL, as shown in Fig. 1. Specifically, we argue that ICL should **not** be viewed as "copying new information to the output (Cho et al., 2025a) or learning new tasks (Li et al., 2024)", but as *removing task-irrelevant information from the query to highlight the specified task*. In detail, since the previous works demonstrate that various kinds of information are encoded in various subspaces in LM's hidden states (Saglam et al., 2025; Zhao

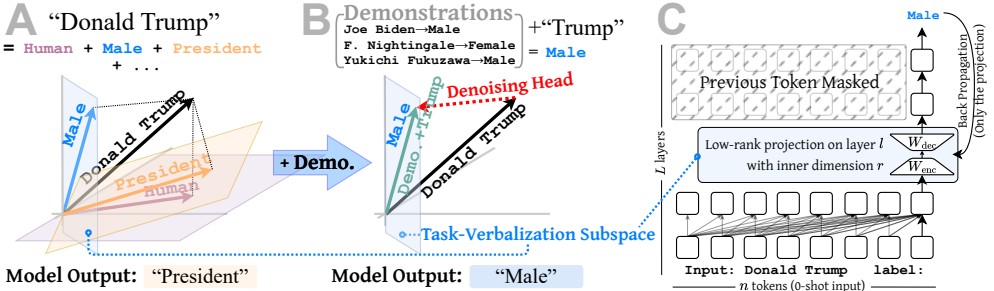

Figure 1: **(A)** A zero-shot query is encoded into a non-selective semantic representation containing all possible label information on various subspaces, making the output arbitrary among these labels. **(B)** Demonstrations help LM filter the label information, saving only the task-related one on the specific subspace (termed **T**ask-**V**erbalization **S**ubspace (TVS)), leading to the task-specific output. **(C)** We explicitly find the TVS by injecting a low-rank filter into the residual stream of zero-shot inputs, and train only the filter to drive the final outputs towards the ground-truth labels.

et al., 2025), we hypothesize that: LMs encode zero-shot queries with superposed information from all possible subspaces that mix task-specific and task-irrelevant information, often causing zero-shot outputs to yield plausible but arbitrary and unexpected answers (e.g., "Donald Trump, label:" → "President" instead of "Male", illustrated in Fig. 1 and demonstrated in Fig. 2). Meanwhile, task-specific information is concentrated in one of these low-rank spaces (termed *Task-Verbalization Subspace* (TVS)) in the hidden states, which leads LM to output in the expected pattern but covered by the noisy information in the zero-shot scenario. To validate this, as shown in Fig. 1 (C), we inject a low-rank filter into the last token's residual stream of zero-shot inputs, and train the filter with the ground-truth output to recover the TVS. Surprisingly, filters preserving only $0.7\%$ of dimensions boost accuracy from near-zero to a remarkable level, showing that removing the TVS-orthogonal irrelevant information suppresses arbitrary outputs and enforces task-specific predictions.

Based on such findings, we generalize the task-oriented information removal from the injected zero-shot settings to vanilla (i.e., un-injected) few-shot settings. As shown in Fig. 1 (B), we observe that LMs, guided by few-shot demonstrations, implicitly drive the hidden states towards TVS calculated independently in the previous low-rank filter injection experiments to produce task-specific outputs. Moreover, we also confirm a consistent behavior in the aforementioned unseen label scenario. Furthermore, we identify a set of attention heads (termed *Denoising Head* in this paper), which are highly independent from the input labels and induction heads, conducting such task-oriented information removal by re-encoding the queries' information. Such head-localization enables ablation experiments by zeroing these heads' outputs, which results in substantial accuracy degradation, especially in unseen label scenarios, where the accuracy drops nearly to zero, strongly indicating the task-oriented information removal as an essential mechanism for ICL.

**In summary, the main contributions of this paper are: (1, §3)** We propose a novel and systematic evaluation framework to measure the task-oriented information removal dynamics in hidden states, which is a versatile methodology for not only the ICL scenario. **(2, §4.2, 4.3)** Based on the proposed evaluation framework, we propose a novel ICL mechanism from the task-oriented information removal perspective, where demonstrations remove the task-irrelevant information from the query to drive LM's output on the specified task. **(3, §4.4)** We identify denoising heads responsible for task-oriented information removal, enabling subsequent ablation to verify their effectiveness. We further address induction heads' limitation on unseen labels by showing that ablating denoising heads (also the information removal) sharply reduces accuracy, and collapses to zero on unseen label scenarios.

## 2 BACKGROUND

**In-context Learning (ICL)** (Radford et al., 2019). In an ICL scenario, given $k$ input-label pairs $\{(x_i, y_i)\}_{i=1}^{k}$ as the *demonstration* and a $x_q$ as the query, a concatenation $[x_1, y_1, \ldots, x_k, y_k, x_q]$ is fed into LMs for $y_q$ corresponding to $x_q$. ICL enables LMs to perform diverse tasks without parameter updates, drawing significant research interest, particularly in mechanism analysis. However, current studies, especially those focusing on attention components such as Induction Heads (Elhage et al., 2021; Wang et al., 2023; Reddy, 2024; Singh et al., 2024; Cho et al., 2025a), cannot account for all ICL phenomena, leaving them an incomplete explanation of ICL behaviors as described below.

**Induction Heads-based Mechanistic Interpretation for ICL.** Induction Head (Elhage et al., 2021) is a family of attention heads identified in LM, primarily responsible for prefix recognition and suffix replication. Specifically, for an input string shaped as $[A, B, A]$, induction heads add the information of B, which is the suffix of the previous A, to the residual stream of the last A, so that drive the output towards B. Such behavior aligns with the form of ICL, where the $x_i$s, especially the consistent structural final token (i.e., ":" in "label:") (Cho et al., 2025a), serve as the As, and the $y_i$s serve as the Bs. The induction heads retrieve the most similar $x_i$s with $x_q$, and copy the corresponding $y_i$ to the output. However, such a framework has a serious issue: it determines the final answer only by copying the tokens that have appeared in the context; thus,

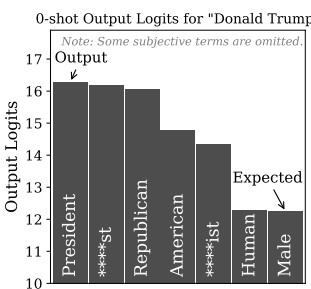

Figure 2: Logits distribution (selected) of zero-shot input "Donald Trump, label:".

if the correct label token does not appear in the demonstrations (termed Unseen Label Scenario), then the framework predicts ICL to fail. But this is not the case in reality, although the accuracy in the unseen label scenario is significantly harmed, but basically not zero (refer to Appendix A.3), which prompts for new investigations to supplement the induction head framework.

## 3 METHOD: TRACING THE TASK-ORIENTED INFORMATION REMOVAL

In this section, we propose a systematic evaluation framework that traces the task-oriented information removal. Following this framework, in §4, we undertake a comprehensive investigation of task-oriented information removal in the ICL setting. While our discussion centers on the ICL scenario as the prototype, we believe the framework extends naturally to a wider range of scenarios.

**Preliminary: Task-Verbalization Subspace (TVS).** As shown in Fig. 1 (A), an LM encodes a zero-shot query $x_q$ into a non-selective semantic representation $h_q^l \in \mathbb{R}^d$ after $l$ transformer blocks, where the output derived from such $h_q$ by the remaining layers is unlikely to be the desired answer to $x_q$, as demonstrated in Fig. 2. In such a scenario, we hypothesize the existence of a low-rank subspace in the representation space encoding the task-related information, equipped with a projection matrix $W \in \mathbb{R}^{d \times d}$ that projects $h_q^l$ to remove task-irrelevant information, thereby redirecting the final output derived from $h_q^l W$ towards the task-specific answer. As shown in §4.2, such $W$ defines the task and also the "verbalization mode" (i.e., how the output is expressed, such as "positive" vs. "+" in sentiment analysis), for which we term $W$ as **T**ask-**V**erbalization **S**ubspace (TVS).

**Step 1: Finding the Explicit TVS in Zero-shot Hidden States.** As shown in Fig. 1 (C), we explicitly assume the existence of TVS parameterized as a low-rank filter $W_{\text{enc}} W_{\text{dec}} \in \mathbb{R}^{d \times r} \cdot \mathbb{R}^{r \times d}$, and verify this assumption by an injection method: we inject $W_{\text{enc}} W_{\text{dec}}$ into the residual stream of the inputs' last token (where the output is expected) in a specific layer $l$, and interrupt this last token's access to previous tokens in the subsequent layers to block the context-related additional information gain from the inputs afterward the filter to keep the attribution clear. Given some zero-shot prompts and ground-truth responses, with all other parts of the LM frozen, we train the $W_{\text{enc}} W_{\text{dec}}$ on zero-shot inputs towards corresponding ground-truth responses (refer to Appendix A.2 for details).

**Step 2: Tracing the Implicit Task-oriented Information Removal towards TVS.** We then assume that LMs intrinsically drive the hidden states towards TVS in the **uninjected** inference with the help of demonstrations. To validate this, we propose two carefully designed geometric metrics of the few-shot hidden states, utilizing the trained $W_{\text{enc}} W_{\text{dec}}$ in Step 1. In detail, given a $N$-amount set of hidden states $H^{l,k} = \{h_i^{l,k}\}_{i=1}^N$ from the last token of $k$-shot ICL inputs after $l$ transformer blocks:

- **Eccentricity**: To measure the *magnitude* of information removal, we calculate the eccentricity as the covariance loaded ratio on the first principal direction of $H^{l,k}$, indicating the enrichment, or anisotropy, of information on a single linear representation (Engels et al., 2025). A higher eccentricity indicates a purer representation with less extraneous information.

- **Covariance Flux on TVS**: To measure the *correctness* of information removal, we calculate the ratio of task-related information, we project all the $h_i^{l,k} \in H^{l,k}$ onto the $W_{\text{enc}} W_{\text{dec}}$ with $r = 8$ trained on layer $l$ in Step 1, and calculate the covariance ratio of $H^{l,k} W_{\text{enc}} W_{\text{dec}}$ against $H^{l,k}$ (details in Appendix A.3). A higher covariance flux suggests that a larger proportion of the information in $H^{l,k}$ is task-related.

In summary, as intuitively shown in Fig. 23, eccentricity quantifies the concentration of information into a single feature, thereby indicating the magnitude of information removal; and covariance flux evaluates whether the task-related information is preserved, i.e., the correctness of the information removal. Appendix G relates these covariance metrics to entropy, a standard measure of information.

**Step 3: Building the Causal Link between Information Removal and ICL Performance by Denoising Heads.** To test whether the detected information removal process indeed has a causal significance, i.e., whether it actively influences accuracy, or merely constitutes a byproduct of accuracy improvement from other mechanisms, we conduct a causality evaluation which actively suppresses such mechanisms. Since the aforementioned two metrics are not directly controllable, we utilize the attention heads as the handle for such causal evaluation. In detail, we quantify each head's contribution to task-oriented information removal by ablating it and re-evaluating the metrics on the hidden states produced by the ablated layer. Then, we label the heads that contribute substantially to these metrics as "**D**enoising **H**eads (DH)", then ablate all the detected DHs and evaluate the ICL performance to verify the causal link. The design of the technical details relies in part on the conclusions from Step 2, as described in §4.4.

# 4 TASK-ORIENTED INFORMATION REMOVAL IN IN-CONTEXT LEARNING

In this section, we conduct comprehensive experiments to apply the aforementioned evaluation framework to trace the task-oriented information removal dynamics in the ICL scenario.

## 4.1 EXPERIMENT SETTINGS

**Models.** We mainly conduct experiments on 6 modern LMs detailed in Appendix A: Llama (v3.2-1B, v3-8B, v3-13B Instruct) (Grattafiori et al., 2024); Qwen 2.5 (3B, 3B Instruct, 7B) (Team, 2024; Yang et al., 2024). We default to reporting the results on Llama 3.2-1B (more results in Appendix I).

**Datasets.** We utilize **6 classification datasets**: SST-2, SST-5 (Socher et al., 2013), MR (Pang & Lee, 2005), FP (Malo et al., 2014), AGNews (Zhang et al., 2015), Subjective (Wang & Manning, 2012); and **3 non-classification datasets**: country-capitals (icyrockcom, 2011), people-profession (Laouenan et al., 2022), opus-100 (Zhang et al., 2020). Unless specified, we report the results on SST-2.

**Others.** To verify whether the LM spontaneously produces task-related outputs, we calculate the accuracy by greedy decoding, where hard matching of the model output among all the vocabularies is utilized for the accuracy, instead of the restricted decoding, where only the most likely output in the pruned output candidates is selected. Refer to Appendix A.1 for the prompt templates.

## 4.2 STEP 1: FINDING THE EXPLICIT TVS IN ZERO-SHOT HIDDEN STATES

In this section, following Step 1 in §3, we show that explicitly removing information from the orthogonal complement of the injected TVS in the zero-shot hidden states guides the LM to recognize the intended task rather than producing arbitrary outputs, suggesting zero-shot hidden states contain task-related information with redundant information interfering with the output, and filtering out such redundancy helps LMs focus on the target task.

**Existence of Explicit TVS in Hidden States.** We conduct the injection-and-train experiment mentioned in Step 1 of §3, with validation accuracy with various $l$ and $r$ on SST-2 and Llama 3.2-1B is shown in Fig. 3, where, globally, compared to zero-shot accuracy, filtering the hidden state while explicitly assuming the existence of a TVS of max-rank[1] 2 can significantly improve the output accuracy from the open-end decoding (mentioned in §4.1). Such results confirm our hypothesis: selectively removing information from zero-shot hidden states steers outputs toward the task, inspiring a possibility that ICL implicitly applies this procedure, as verified in §4.3.

$W_{\mathbf{enc}}W_{\mathbf{dec}}$ **is Information Removal.** One suspicion is that $W_{\mathrm{enc}}W_{\mathrm{dec}}$ may not significantly remove information, such as when the principal component direction of $h_q^l$s is highly aligned with the eigenvector of $W_{\mathrm{enc}}W_{\mathrm{dec}}$. We rule out such doubt by: (1) Calculating the remaining covariance-load ratio

---

[1]Notice that even if the $r$ is sufficient large, the effective rank of $W_{\mathrm{enc}}W_{\mathrm{dec}}$ can also be small, so the $r$ is the maximum possible rank. (Original embedding dimension is 2048.)

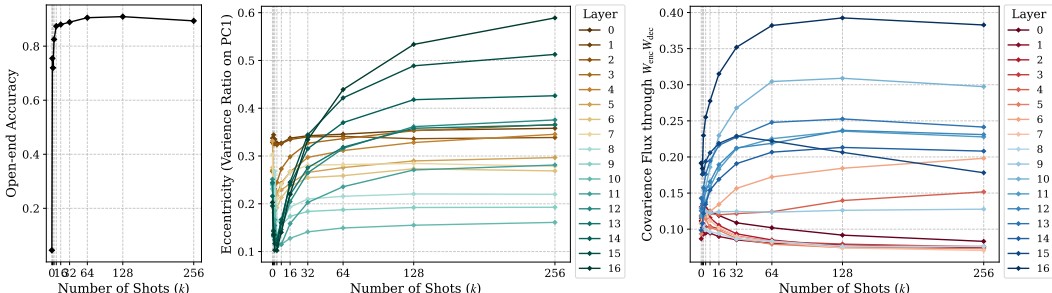

Figure 4: Test results of **(left)** open-end accuracy against demonstration numbers $k$, **(middle)** eccentricity of hidden states point cloud against demonstration numbers $k$ on various layers, **(right)** covariance flux through the catched $W_{\text{enc}}W_{\text{dec}}$ against demonstration numbers $k$ on various layers.

out of the top-$r$ principal components of $h_q^l$s, as a lower bound of information removal by $W_{\text{enc}}W_{\text{dec}}$ with maximum rank $r$ (explained in Appendix B, and numerically visualized in Appendix I), as shown in the diamond size of Fig. 3, suggesting effective information removal by $W_{\text{enc}}W_{\text{dec}}$. (2) Calculating the cosine similarity of $W_{\text{enc}}$'s eigenvectors and the principal components of the hidden states, as an estimation of the positive gap between the actual information removal and the lower limit, shown in Appendix B. In summary, $W_{\text{enc}}W_{\text{dec}}$ is an effective information removal while enhancing the zero-shot accuracy.

$W_{\text{dec}}$ **Controls the Verbalization Mode.** Notice that, in the training of the $W_{\text{enc}}W_{\text{dec}}$, not only is the task-related information (as shown in Fig. 1 (B)), but also the verbalization mode, i.e., how the task-related information is transferred into output tokens (e.g., into "positive/negative" or "+/−") is defined in the trained parameters. This is also why we name $W_{\text{enc}}W_{\text{dec}}$ as "task-verbalization". Moreover, we find that the verbalization mode is mainly controlled by the $W_{\text{dec}}$: In detail, we first train the $W_{\text{enc}}W_{\text{dec}}$ filter on SST-2 using the standard verbalizations "positive" and "negative", then freeze either $W_{\text{enc}}$ or $W_{\text{dec}}$ and fine-tune the remaining one to transfer the outputs to symbolic verbalizations "A" and "B". The results, shown in Table 1, indicate that fine-tuning only $W_{\text{dec}}$ successfully transfers the verbalization into symbolic form, whereas $W_{\text{enc}}$ fails. This suggests that $W_{\text{enc}}$ extracts task-specific but verbalization-irrelevant semantics from hidden states (otherwise, freezing it will not prevent successful verbalization transfer), while $W_{\text{dec}}$ maps them into the target verbalization by aligning hidden states with the specified output unembedding vectors (Yang et al., 2025), consistent with prior findings (Tao et al., 2024).

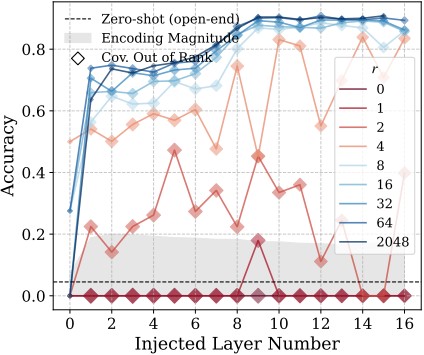

Figure 3: Filter-injection evaluations on various layers and ranks. **Marker size**: the remaining covariance out of the top-$r$ principal components of the point cloud of $h_q^l$. Refer to Appendix I for numerical results. **Encoding Magnitude**: the quality of the hidden states in the current layer as a sentence encoding of the query, emerging simultaneously with the test accuracy, calculated follows Cho et al. (2025b).

### 4.3 STEP 2: TRACING THE IMPLICIT TASK-ORIENTED INFORMATION REMOVAL

In §4.2, removing the TVS-orthogonal information produces task-specific output, which serves as efficient adaptors for LM to specific tasks, and motivates us to hypothesize the mechanism of ICL with few-shot demonstrations similar to such task-oriented information removal. In this section, following Step 2 in §3 to measure the two geometric metrics (i.e., Eccentricity and Covariance Flux) of the few-shot hidden states from uninjected inference processing, thereby verifying the hypothesis shown in Fig. 1 (B): LMs autonomously remove the redundant information out of the TVS with the help of few-shot demonstrations to produce task-specified ICL outputs.

Table 1: Fine-tuning results with parts of the filter frozen, on Llama 3.2-1B and SST-2.

| Trained Part | Both | $W_{\text{enc}}$ | $W_{\text{dec}}$ |
|---|---|---|---|
| **Accuracy** | 0.88 | 0.00 | 0.84 |

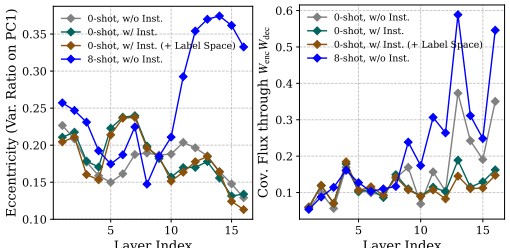
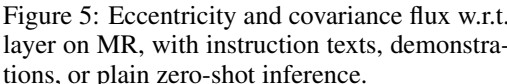
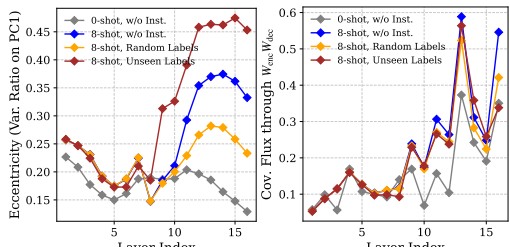

Figure 5: Eccentricity and covariance flux w.r.t. layer on MR, with instruction texts, demonstrations, or plain zero-shot inference.

Figure 6: Eccentricity and covariance flux w.r.t. layer on MR, with random label, unseen label, and normal settings.

**Language Models Intrinsically Compress Hidden States towards TVS.** We test both the metrics on various $k$ and $l$, with results as well as the output accuracy shown in Fig. 4 for Llama 3.2-1B and SST-2 (refer to Appendix I for more cases). Globally, as the amount of demonstration ($k$) increases, the task-oriented information removal primarily occurs in the middle-to-late layers, causing an increasing eccentricity and covariance flux. In detail:

- **Eccentricity:** As shown in Fig. 4 (middle), with sufficient demonstration numbers ($k$), in the later layers, the eccentricity increases against $k$, reaching a maximum of around 60%, showing a highly anisotropic information compression with the help of demonstrations. Specifically, on certain datasets such as SST-2, the eccentricity exhibits a distinctive nonmonotonic *decrease-and-increase* pattern against $k$. This suggests that information removal initially compresses from the first principal component of zero-shot hidden states, which is understandable since the input distribution may be disaligned with the task, and confirms an important conclusion: task-oriented information removal primarily compresses from the task-irrelevant directions (Yang et al., 2025; Kirsanov et al., 2025) of the hidden state point cloud, as discussed in Appendix C.

- **Covariance Flux:** As shown in Fig. 4 (right), also in the later layers, the covariance flux increases against $k$, suggesting that LMs correctly preserve the task-related information while removing the task-irrelevant one with the help of demonstrations. Moreover, the covariance flux of the later layers closely tracks the accuracy (Fig. 4 (left)), with both reaching their peak and then saturating at the same point, which indicates that the alignment of hidden states toward TVS constitutes a crucial mechanism shaping the inference output[2].

Additionally, such findings concur with previous works, where the early layers contribute to task-agnostic low-level semantic encodings (Jawahar et al., 2019; Chen et al., 2023; Wang et al., 2023; Cho et al., 2025a; Yang et al., 2025), thus task-related behavior such as task-oriented information removal, can not be significantly observed in the early layers by our metrics. Conversely, in later layers, high-level features are being processed, causing the task-related operation to be obvious.

**Effect of Instruction.** In practice, instruction text describing the task (e.g., "Please predict the sentiment of this text:") can be utilized instead of the few-shot demonstrations in the inputs. Therefore, we examine whether such instructions produce the task-oriented information removal functionally similar to the few-shot demonstrations by measuring both metrics on the instructed inputs (configurations detailed in Appendix A.1). As shown in Fig. 5, hidden states with 8-shot demonstrations exhibit obvious morphological differences and stronger information removal over the 0-shot inference with or without instruction text. Also, hidden states with instruction, even if the label spaces are indicated (e.g., "Please predict in Positive and Negative"), show information removal dynamics almost consistent to the zero-shot inference (Kirsanov et al., 2025), despite the clearly higher accuracy (refer to Appendix A.3). Such observation suggests that the information removal is only evoked by few-shot demonstrations, with a different mechanism from instruction.

**Effect of Random Labels and Unseen Labels.** Previous works demonstrated that the correctness of labels presented in the demonstrations influences the inference dynamics and output accuracy insignificantly (Min et al., 2022; Yoo et al., 2022; Pan, 2023; Cho et al., 2025a). Therefore, we measure both the metrics with label tokens randomly sampled from all possible labels (detailed in Appendix A.1), as well as the unseen label settings, with results shown in Fig. 6. Focus on the eccentricity results, hidden states with randomized labels exhibit weaker information removal than

---

[2]The two curves differ in form, which is natural since accuracy is a nonlinear metric (Schaeffer et al., 2023).

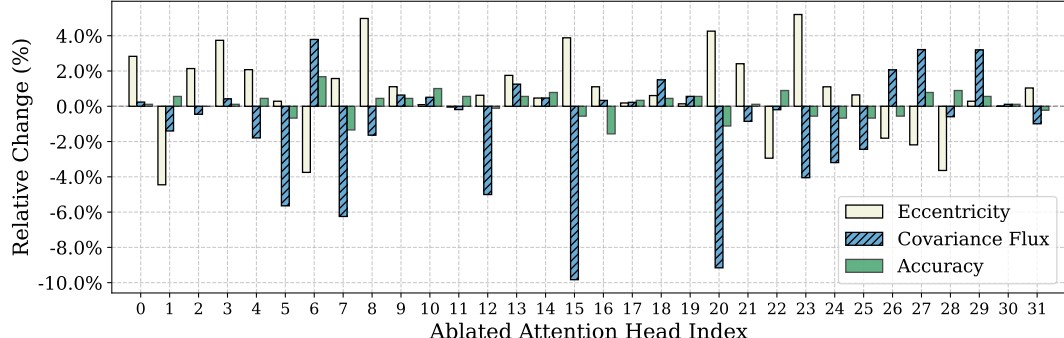

Figure 7: Ablation results for the two metrics and output accuracy of every head in Llama3.2-1B Layer 9 on 8-shot SST-2 (relative to the clean run). A smaller value indicates a more severe drop with ablation, i.e., this attention head contributes more significantly to the metric.

the plain 8-shot inference, suggesting that mismatched labels weaken the impact of demonstration, leading to weaker task-oriented information removal, which echoes previous works (Cho et al., 2025a) showing that mismatched labels neutralize the functionality of demonstrations. In contrast, the demonstrations with unseen labels show a stricter removal, and we infer the reason as: demonstrations with a smaller label space actually specify a narrower task, causing more information to be identified as redundancy. More evidence is shown in Fig. 53, where such an effect is weakened on multi-way tasks than on 2-way tasks, where the removal of each label has a large impact on the task scope. Overall, task-oriented information removal appears across all label settings, differing only in magnitude, regardless of the labels' completeness or alignment to the demonstration. The above degrading configurations merely shift the information removal away from the optimal range, resulting in reduced accuracy (we list the output accuracies in these configurations in Appendix A.3).

## 4.4 Step 3: Causality Test by Denoising Heads

We have shown that selectively removing information from zero-shot hidden states boosts accuracy in the injected settings, and few-shot learning under un-injected settings implicitly simulates a similar process. This section therefore investigates the causal role of such an operation. As outlined in Step 3 of §3, we first identify the model components responsible for this removal, thereby paving the way for subsequent ablation analysis to examine the significant accuracy influence, especially under the unseen label settings. The experiments in this section default to using 8-shot inputs.

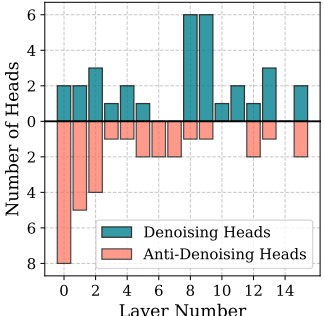

Figure 8: Counting of DHs and anti-DHs w.r.t. layers.

**Identify Denoising Heads in LMs.** Following the ablate-and-remeasure method mentioned in §3, a set of representative results on Layer 9 is shown in Fig. 7 (refer to Appendix I for other settings). Because eccentricity trends have a phase-transition pattern[3], we use covariance flux to identify significantly contributing attention heads (i.e., *Denoising Heads* (DH)), with some (e.g., #5, 7, 12, 15, 20, . . . ) showing clear contributions to both metrics and accuracy. Setting[4] $\pm 3.5\%$ as the thresholds, we observe the distribution of the DHs, and also the anti-DHs whose ablations increase the covariance flux remarkably, as shown in Fig. 8 (refer to Appendix I for more settings). While the DHs can be observed in almost every layer, their numerical advantage against the anti-DHs emerges only in the middle-to-late layers, where task-oriented information removal becomes apparent as discussed in §4.3. This further suggests the transition from early-stage task-agnostic encoding (Cho et al., 2025a) to later-stage task-related filtering: in the former, most

---

[3]As noted in §4.3, a contributive operation may reduce eccentricity, but in later layers (e.g., Layer 10), once redundancy is sufficiently removed and task-relevant information aligns with the first principal component, the contributive information removal begins to increase the eccentricity.

[4]Note that the goal of this experiment is to identify a subset of prominent denoising heads for visualization or causal analysis, rather than to exhaustively capture all potential candidates. Therefore, as discussed in Appendix E, the threshold should be a relatively selective but not extreme value. Based on the visualization in Fig. 20, we consider 3.5% or 5% (mentioned below) to be an ideal choice.

heads conduct non-selective query encoding, leading to anti-DHs; in the latter, more DHs play a selective and decisive role. Moreover, during this experiment, we find more interesting attention heads as shown in Appendix F.

**Denoising Heads are Independent of Induction Heads.** To demonstrate the independence of information removal behavior and also DHs against the induction heads, we observe the overlap of induction heads and the DHs. In detail, we calculate the magnitude of induction for each head as the sum of attention scores from all the label tokens to the last token, and visualize it together with the ablation effects on the covariance flux, as shown in Fig. 9 (refer to Appendix I for other settings), where we find that: although induction heads and task-oriented information removal emerge in similar layers, these attention heads rarely overlap, which clearly suggests that: DHs, and their task-oriented information removal effect, is an independent and novel operation evoked by the ICL-prompted LMs.

**Overlap of Denoising Heads among Tasks.** To identify whether the evoking of the DHs is task-specific or task-irrelevant, which suggests **I**n-**w**eight **L**earning (IWL) (Chan et al., 2022; Reddy, 2024; Chan et al., 2025) or in-context learning characteristics, we calculate the overlap count of the DHs identified on all the task pairs as $|\mathcal{D}_A \cap \mathcal{D}_B|$, where the $\mathcal{D}_A$ or $\mathcal{D}_B$ is the DH set detected on the task $A$ or $B$ by bottom-$K$ covariance flux relative change after ablation, as shown in Fig. 11 with $K = 1\%$ . In the results, we observe several **common** DHs across different dataset pairs, indicating that they exhibit ICL properties by extracting and discriminating at least two task information from the context and leveraging it for operation. In contrast, most DHs are **dataset-specific**, reflecting the IWL property but still evoked by ICL demonstrations: on mismatched datasets, these heads are inactive, neither hindering nor facilitating information removal (e.g., Layer 9 Head 20 as a DH for sentiment-analysis in Appendix I), thus challenging current strict separation of ICL and IWL. These findings suggest that task-oriented information removal is a hybrid of ICL and IWL, supporting the previous works that ICL capability arises from latent tasks acquired during pretraining (Li et al., 2024) and is further strengthened by demonstrations (Yang et al., 2025; Cho et al., 2025c).

**Attention Pattern of Denoising Heads: Local Re-encoding.** To find how the DHs conduct the task-oriented information removal operation, we visualize the attention maps of a common DH (Layer 9 Head 15) of Llama 3.2-1B, as shown in Fig. 10 (refer to Appendix A.4 for detail, Appendix I for more cases). Surprisingly, although task-oriented information removal is a highly context-dependent operation in the ICL-based common DH, the DH exhibits a local attention pattern. Specifically, significant attention scores of the last token are almost confined to the query tokens (ignoring the attention sink (Gu et al., 2025)), thereby forming a local re-encoding pattern. In such processing, as demonstrated in Appendix D, DH appropriately identifies and enforces task-relevant information by selective attention calculation, where the vector extracted from the last-token hidden state by $W_Q^\top W_K$ of the DH may serve as an indicator for locating this information, since it serves an important part of the attention calculation, which is decisive for selecting the key tokens that contain the task-related information.

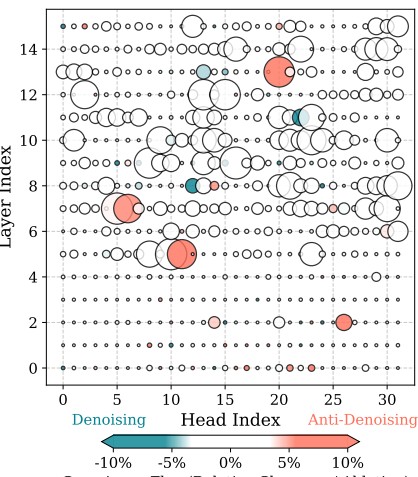

Figure 9: Head-wise (**scatter size**) induction magnitude and (**color**) covariance flux relative change after ablation.

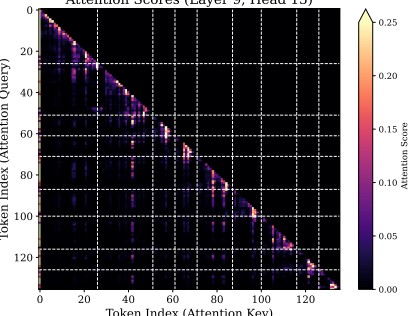

Figure 10: Attention map of a common DH (Layer 9, Head 15). White dotted lines are the position of label tokens.

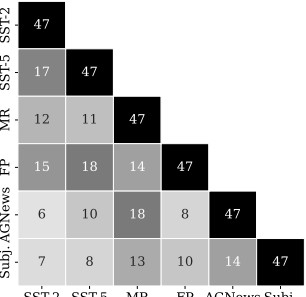

Figure 11: Overlap of DH over all dataset pairs.

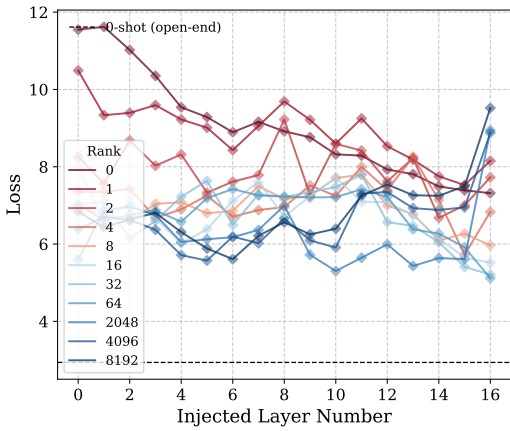

Figure 12: Filter-injection results on country-capitals task. In **cross-entropy loss**.

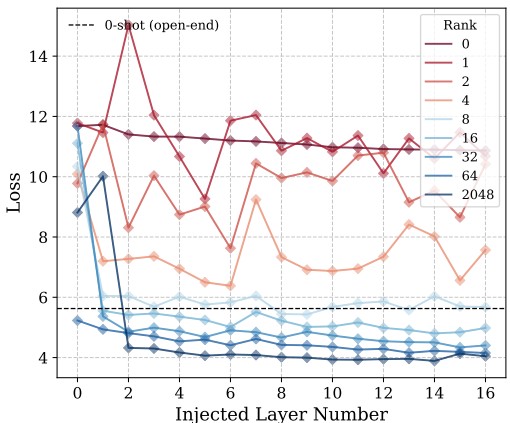

Figure 13: Filter-injection results on the people-profession task. In **cross-entropy loss**.

**Task-oriented Information Removal as a Key Mechanism for ICL with Unseen Label.** Identifying the DHs responsible for the task-oriented information removal enables ablation experiments to evaluate the significance of the operation to ICL. Therefore, we remove DHs from the inference, specifically, setting the outputs of all heads with a covariance flux relative change below $-5\%$ to zero vectors, and report the resulting accuracy in Table 2 (refer to Appendix I for non-average results), where when demonstrations are randomly sampled, or at least one demonstration shares the same label as the query (Seen Label), zeroing out the DHs significantly but not critically reduces accuracy compared to randomly selected heads, since the induction heads provide a strong supportive mechanism in such cases. In the unseen label setting where induction heads fail, DHs become decisive: ablating them almost eliminates accuracy, even without exhaustively identifying all DHs[5]. This shows that DHs are the primary source of accuracy in unseen-label scenarios, effectively acting as the main complement to induction heads (Cho et al., 2025a).

Table 2: Ablated inference accuracy (%) of DHs, averaged on all 6 datasets. **Layer**: the ratio of the layer scanned for DH / total layers. **w/o DH**: accuracy with DHs ablated. **w/o RH**: accuracy with ablation of randomly sampled attention heads of equal amounts to DHs on each layer. **P-value**: the probability for "w/o RH" to sample a value less than "w/o DH" (i.e., the likelihood of DHs being randomly sampled). (Due to computational constraints, we only evaluate Llama 3-13B on SST-2.)

| Model Layer | Demonstration Configuration | 8-shot | 8-shot w/o DH | 8-shot w/o RH | P-value |
|---|---|---|---|---|---|
| Llama 3.2-1B 16 / 16 | Random Sample | 71.02 | 55.04 | $67.29_{3.84}$ | $7\times10^{-4}$ |
| | Seen Label | 73.40 | 56.52 | $69.02_{3.72}$ | $4\times10^{-4}$ |
| | Unseen Label | 14.54 | 1.06 | $10.65_{4.65}$ | $2\times10^{-2}$ |
| Llama 3-8B[5] 7 / 32 | Random Sample | 77.63 | 72.05 | $76.38_{0.96}$ | $3\times10^{-6}$ |
| | Seen Label | 80.47 | 75.11 | $79.70_{1.30}$ | $3\times10^{-4}$ |
| | Unseen Label | 21.66 | 7.52 | $18.94_{3.98}$ | $2\times10^{-3}$ |
| Qwen 2.5-3B[5] 18 / 36 | Random Sample | 73.97 | 68.41 | $74.31_{1.29}$ | $2\times10^{-6}$ |
| | Seen Label | 76.41 | 70.31 | $76.36_{1.57}$ | $1\times10^{-4}$ |
| | Unseen Label | 23.49 | 11.59 | $26.82_{3.82}$ | $3\times10^{-5}$ |
| Qwen2.5 3B-Ins[5] 18 / 36 | Random Sample | 77.24 | 75.57 | $77.28_{1.90}$ | 0.18 |
| | Seen Label | 78.86 | 77.52 | $78.05_{0.81}$ | 0.25 |
| | Unseen Label | 48.70 | 37.51 | $46.32_{4.52}$ | $2\times10^{-2}$ |
| Llama 3 13B Ins[5] 13 / 55 | Random Sample | 76.86 | 71.58 | $77.83_{1.57}$ | $3\times10^{-5}$ |
| | Seen Label | 77.34 | 72.16 | $81.20_{2.12}$ | $1\times10^{-5}$ |
| | Unseen Label | 29.59 | 10.35 | $32.13_{4.89}$ | $4\times10^{-6}$ |

**Amplifying the Outputs of DHs Boosts ICL Accuracy.** As a prototypical empirical application utilizing the discovery of DH, we try to amplify the output of DH on 8-shot inputs, i.e., multiply a scalar $(1 + \lambda)$ on the outputs of DHs detected similarly as the ablation in Table 2. The results with various $\lambda$ are shown in Fig. 14, where a moderate amplification consistently increases ICL accuracy. Especially in the random-label scenario in §4.3, DH amplification yields a markedly stronger effect, consistent with Fig. 6, where information removal is weaker than in the true-label setting. This amplification provides a frugal, plug-and-play way to boost ICL accuracy and mitigate label noise without extra computational overhead, while reinforcing the causal link between information removal and ICL accuracy.

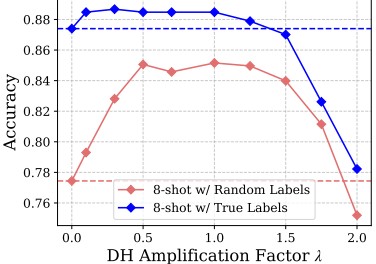

Figure 14: DH amplification results on Llama 3.2-1B and SST-2.

---

[5]Due to computational constraints, we only identify the denoising heads on some of the layers of these models, as shown in Appendix I, so the ablation results here do not cover all denoising heads.

## 5 CONCLUSION AND DISCUSSION

**Conclusions.** This paper proposes a novel perspective to interpret ICL inference as a task-oriented information removal from the query's hidden states. In detail, we first demonstrate that injecting explicit low-rank filters into the zero-shot hidden states to reduce their redundant information improves accuracy. We then show that few-shot ICL spontaneously performs this process, identify the DHs responsible for it, and confirm both the independence from the induction head, and the significant effectiveness of information removal for ICL tasks by ablation experiments.

**Comparison and Connection to Previous Works.** This paper is related to the two categories of previous works: **(1) Task vector-based mechanism.** Our analysis of the attention patterns shows that denoising heads access subspaces of the last hidden states through $W_Q^\top W_K$ for the re-encoding operation. The $W_Q^\top W_K$ thus acts as an operator extracting task representations to trigger and guide the denoising operations. This view suggests a new direction for task-vector research: identifying the subspaces that purify the task representation from the coarse-grained hidden states, and finding the components that mediate their effects (Yang et al., 2025; Yin & Steinhardt, 2025), rather than only relying on coarse steering. **(2) Induction heads-based mechanism.** As mentioned, current induction head-based works slough over the case of unseen labels and resort to vague explanations (Cho et al., 2025a), which our paper addresses. From our ablation study (Table 2), ICL behavior can be attributed almost fully to induction and DHs, with no significant bypasses since ablating DHs in the unseen label scenario reaches near-0 accuracy. Although induction heads contribute more to accuracy, the DH, as the complementary mechanism identified when they fail, remains meaningful.

**Failure Case: Clustering vs. Translation.** Notice that the task-oriented information removal investigated in this paper is essentially a dimensionality reduction operation, where it is conceivable that classification tasks, which cluster a vast input space (i.e., the whole set of the input sentences $x$) to a narrow label space, precisely align with such a dimensionality reduction. While, some works identified translation operation (Merullo et al., 2024; Jiang et al., 2024; Bu et al., 2025) in hidden states directing entities' semantics towards their attributes, especially unique ones (called "Fact Recall" (Zhang et al., 2025), e.g., "Japan"→"Tokyo"), where the clustering operation as presented in classification tasks can not be applied since the mapping from the input space to the output space is a bijection. Therefore, the task-oriented information reduction process observed in this paper is unlikely to apply to the bijection scenario, as information is not removed: input can be losslessly reconstructed from the output. We demonstrate this point by repeating the filter-injection experiment shown in §4.2 on the country-capitals task as shown in Fig. 12, where no loss lower than the 0-shot baseline can be obtained by any filter, even if high-dimensional (not necessarily high-rank) ones. As a comparison, we repeat the experiment on a similar fact recall task but with a clustering structure, which maps a person's name to their profession (e.g., "F. Nightingale"→"Nurse"). As shown in Fig. 13, evaluation results better than 0-shot can be obtained by a low-rank filter, which confirms our idea that the task-oriented information removal does not occur in bijective ICL tasks, while clearly highlighting the distinction between these two types of tasks. Some works on synthetic datasets also discuss the uniqueness of such a full-rank translation scenario (Dong et al., 2025) and obtain similar conclusions. We develop such a discussion on a generative scenario in Appendix H.

**Limitations. (1) Long-term effect of head ablation.** In Fig. 7, we identify DHs by ablated metrics only on the current layer's hidden states. However, since the effect of ablation may propagate to deeper layers, the DH sets could be underestimated. Exhaustively measuring all layers would be costly, yet our ablation already shows that these heads, though not the whole set of DHs, produce clear and significant effects on the outputs. **(2) Fine-grained mechanism.** We propose a prototype mechanism for DHs as the extraction of denoising signals from some subspaces of hidden states. A more detailed investigation is needed to validate such a mechanism prototype for DHs. **(3) Efficient localization of DHs.** In §4.4, we adopt a naive ablation-remeasurement method, which is costly for localization-controlling applications (Cho et al., 2025b). Future work may utilize gradient-based methods, or leverage DH characteristics, such as the attention pattern in Fig. 10, for more efficient localization. **(4) Complex information bottleneck.** In this paper, we utilize a linear low-rank filter as the information bottleneck as shown in Fig. 1 (C). This essentially assumes that the features on these subspaces are linearly separable in the linear space, as illustrated in Fig. 1 (A). However, some works (Engels et al., 2025; Modell et al., 2025) show that features may also be embedded on manifolds and be nonlinearly separable. Therefore, extending the discussion to more complex tasks may require a more complex nonlinear information bottleneck.

ACKNOWLEDGMENTS

This work was supported by the JST FOREST Program (Grant Number JPMJFR232K, Japan) and the Nakajima Foundation. The authors would like to sincerely thank the chairs and reviewers of ICLR 2026 for their thoughtful and insightful feedback on this paper, their perceptive reviews and discussions have greatly improved this work.

THE USE OF LARGE LANGUAGE MODELS

In this paper, LLMs are used and only used to polish writing.

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

# Appendices

Table 3: Prompt templates used in this paper.

| Dataset | Prompt Template (Unit) | Label Tokens |
|---|---|---|
| SST-2 | `sentence: [x] sentiment: [y] \n` | negative, positive |
| MR | `review: [x] sentiment: [y] \n` | negative, positive |
| FP | `sentence: [x] sentiment: [y] \n` | negative, neutral, positive |
| SST-5 | `sentence: [x] sentiment: [y] \n` | poor, bad, neutral, good, great |
| AGNews | `news: [x] topic: [y] \n` | world, sports, business, science |
| Subjective | `review: [x] subjectiveness: [y] \n` | objective, subjective |
| country-capitals | `country: [x], label: [y] \n` | - |
| people-profession | `name: [x], label: [y] \n` | - |
| opus-100 | `sentence: [x], translation: [y] \n` | - |

Table 4: Instruction used in Fig. 5.

| | Dataset | Instruction |
|---|---|---|
| **Basic** | SST-2 | You are a helpful assistant. Please predict the sentiment of the following sentence: |
| | MR | You are a helpful assistant. Please predict the sentiment of the following sentence: |
| | FP | You are a helpful assistant. Please predict the sentiment of the following sentence: |
| | SST-5 | You are a helpful assistant. Please predict the sentiment of the following sentence: |
| | AGNews | You are a helpful assistant. Please predict the category of this news: |
| | Subjective | You are a helpful assistant. Please predict the subjectivity of this sentence: |
| **w/ Label Space** | SST-2 | You are a helpful assistant. Please predict the sentiment of the following sentence in positive and negative: |
| | MR | You are a helpful assistant. Please predict the sentiment of the following sentence in positive and negative: |
| | FP | You are a helpful assistant. Please predict the sentiment of the following sentence in positive, neutral, and negative: |
| | SST-5 | You are a helpful assistant. Please predict the sentiment of the following sentence in poor, bad, neutral, good, and great: |
| | AGNews | You are a helpful assistant. Please predict the category of this news in world, sports, business, science: |
| | Subjective | You are a helpful assistant. Please predict the subjectivity of this sentence in objective and subjective: |

## A  EXPERIMENT DETAILS

All the models and datasets are loaded from HuggingFace, with checkpoint names listed in Table 5. The Llama 3-13B Instruct is quantized to `INT4`.

### A.1  PROMPT TEMPLATE

In this paper, we build the input prompt from Cho & Inoue (2025), detailed in Table 3. Moreover, in the experiments shown in Fig. 5, we utilize the instructions as shown in Table 4, with and without label spaces provided in the instruction.

We illustrate seen vs. unseen label scenarios in Fig. 15. In the seen case, the query's ground-truth answer appears at least once in the demonstration, activating induction heads for direct copying. In the unseen case, the answer is absent, and induction heads lose their functionality (Cho et al., 2025a).

Table 5: Models and corresponding checkpoint names used in this paper.

| Model | Checkpoint |
|---|---|
| Llama 3.2-1B | `meta-llama/Llama-3.2-1B` |
| Llama 3-8B | `meta-llama/Meta-Llama-3-8B` |
| Llama 3-13B Instruct | `elinas/Llama-3-13B-Instruct` |
| Qwen 2.5-3B | `Qwen/Qwen 2.5-3B` |
| Qwen 2.5-3B Instruct | `Qwen/Qwen2.5-3B-Instruct` |
| Qwen 2.5-7B | `Qwen/Qwen 2.5-7B` |

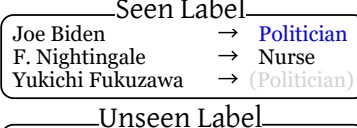

Figure 15: Examples of seen label demonstration and unseen label demonstration.

Also, a random label prompt utilized in §4.3 is built by randomly sampling the labels presented in the demonstrations in the label space. For example, for the prompt shown in Fig. 16, we randomly (with probability 50%) flip the "positive" to "negative" and "negative" to "positive".

### A.2  TRAINING DETAILS OF $W_{\text{enc}}W_{\text{dec}}$ (§4.2)

As shown in Fig. 1 (C), in the experiment of §4.2, we train two contiguous linear layers in the residual stream of the LMs, with only the first linear layer (parameterized by $W_{\text{enc}}$) attached with a bias term. During the training, only the two linear layers are unlocked for parameter update. We sample 2048 zero-shot training examples, and train the filter on Adam (Kingma, 2014) with learning rate $10^{-4}$, momentum factor $\beta_1 = 0.9$, $\beta_2 = 0.999$. After the gradient calculation of every 32 training samples, we update the parameter once, i.e., we utilize a pseudo batch size of 32. The filters are trained for 4 epochs. After the training, we test the model on a 512-size hold-out test set by strict token matching across the entire vocabulary space.

sentence: 's a rollicking adventure for you and all your mateys , regardless of their ages .
sentiment: positive\n
sentence: playing a role of almost bergmanesque intensity ... bisset is both convincing and radiant sentiment: positive\n
sentence: robust and scary sentiment: positive\n
sentence: addictive guilty pleasure sentiment: positive\n
sentence: chance to find love in the most unlikely place sentiment: positive\n
sentence: but because of the startling intimacy sentiment: positive\n
sentence: the movie 's seams may show ... but sentiment: positive\n
sentence: if tragic ) sentiment: negative\n
sentence: likable story sentiment:

Figure 16: The input for attention visualization.

## A.3 DETAILS FOR ECCENTRICITY / COVARIANCE FLUX AND EXPERIMENTS OF §4.3

We intuitively illustrate the eccentricity and co-variance flux metrics in Fig. 23.

**The Calculation of Covariance Flux.** Given the last token hidden state set $H^{l,k} = \{h_i^{l,k}\}_{i=1}^N$ from layer $l$ and $k$-shot ICL prompts, we recall the low-rank filter obtained in §4.2 (independently trained from 0-shot setting) as $W_{\text{enc}}$ and $W_{\text{dec}}$. Then, we calculate the mapped[6] hidden state set as $H_1^{l,k} = \{h_i^{l,k} W_{\text{enc}} W_{\text{dec}}\}_{i=1}^N$, and calculate the covariance matrix of the mapped set as $D_1^{l,k} = \text{Cov}[H_1^{l,k}]$, also the covariance matrix of the original set as $D^{l,k} = \text{Cov}[H^{l,k}]$. Then, we calculate the covariance flux as:

$$\text{Covariance Flux} = \frac{\|D_1^{l,k}\|_*}{\|D^{l,k}\|_*}, \quad (1)$$

where the $\|\cdot\|_*$ is the nuclear norm, i.e., the sum of the singular values.

**Experiment Details.** Based on the same 512-size test set with Appendix A.2, we sample 2 demonstration sequence for each test sample with $k$ demonstrations. Then, on the specified layer, we extract the hidden states from these 1024 test samples to measure the 2 metrics.

**Accuracy with Instruction and Label Configurations.** As supplementary information, we list the accuracies in the input configurations shown in Fig. 5 and 6 in Table 6.

## A.4 DETAILS FOR ATTENTION VISUALIZATION IN FIG. 10

In the attention visualization, we input an 8-shot ICL prompt as shown in Fig. 16 into Llama 3.2-1B, and extract the attention scores from the specified attention head.

Table 6: Accuracies (%) in all configurations shown in Fig. 5 and 6. **Ins.**: with basic instruction; **Ins. (LS)**: with instructions and label space prompt; **Unseen**: with unseen label settings; **Rand** $y$: with randomly sampled demonstrations label tokens in the original label space.

| | Dataset | 0-shot | 0-shot Ins. | 0-shot Ins. (LS) | 8-shot | 8-shot Unseen | 8-shot Rand $y$ |
|---|---|---|---|---|---|---|---|
| **Llama 3.2-1B** | SST-2 | 4.49 | 22.27 | 48.83 | 87.40 | 15.14 | 77.44 |
| | MR | 0.00 | 42.19 | 46.68 | 90.23 | 44.92 | 88.77 |
| | FP | 1.17 | 12.30 | 15.82 | 73.14 | 0.39 | 65.04 |
| | SST-5 | 0.00 | 0.78 | 17.97 | 42.19 | 2.93 | 37.60 |
| | AGNews | 0.20 | 2.34 | 15.23 | 71.78 | 23.83 | 66.21 |
| | Subjective | 0.00 | 0.00 | 0.00 | 61.43 | 0.00 | 50.78 |
| | Average | 0.98 | 13.31 | 24.09 | 71.03 | 14.53 | 64.31 |
| **Llama 3-8B** | SST-2 | 17.38 | 66.60 | 24.61 | 91.31 | 27.15 | 82.71 |
| | MR | 0.00 | 65.04 | 65.23 | 93.07 | 52.83 | 83.89 |
| | FP | 27.34 | 25.98 | 22.66 | 82.71 | 9.57 | 57.03 |
| | SST-5 | 0.20 | 0.00 | 23.63 | 45.31 | 6.74 | 40.23 |
| | AGNews | 0.98 | 12.11 | 9.77 | 75.59 | 33.69 | 48.73 |
| | Subjective | 0.00 | 0.00 | 0.00 | 77.83 | 0.00 | 53.81 |
| | Average | 7.65 | 28.29 | 24.32 | 77.64 | 21.66 | 61.07 |
| **Llama 3-13B Ins.** | SST-2 | 63.48 | 5.66 | 58.79 | 76.86 | 29.59 | 80.08 |
| | MR | 11.13 | 1.56 | 49.80 | 79.79 | 35.64 | 79.39 |
| | FP | 70.70 | 6.25 | 21.29 | 83.11 | 65.43 | 75.10 |
| | SST-5 | 4.30 | 0.00 | 42.58 | 47.85 | 7.03 | 44.34 |
| | AGNews | 13.67 | 24.22 | 14.65 | 68.75 | 58.79 | 60.16 |
| | Subjective | 0.00 | 0.00 | 0.00 | 79.79 | 0.00 | 53.81 |
| | Average | 27.21 | 6.28 | 31.18 | 72.69 | 32.75 | 65.48 |
| **Qwen 2.5-3B** | SST-2 | 43.55 | 47.66 | 77.34 | 92.58 | 56.05 | 81.84 |
| | MR | 28.13 | 61.52 | 84.38 | 91.31 | 50.98 | 83.01 |
| | FP | 8.40 | 31.05 | 72.85 | 78.81 | 25.00 | 55.86 |
| | SST-5 | 0.00 | 0.00 | 41.99 | 48.14 | 5.37 | 41.41 |
| | AGNews | 0.59 | 26.95 | 67.19 | 75.39 | 41.50 | 58.20 |
| | Subjective | 0.00 | 0.00 | 0.00 | 62.79 | 0.00 | 52.54 |
| | Average | 13.44 | 27.86 | 57.29 | 74.84 | 29.82 | 62.14 |
| **Qwen 2.5-3B Ins.** | SST-2 | 56.05 | 52.93 | 82.81 | 90.33 | 69.04 | 85.16 |
| | MR | 74.21 | 24.02 | 79.49 | 89.84 | 64.84 | 80.37 |
| | FP | 13.48 | 53.71 | 66.21 | 87.89 | 77.73 | 76.27 |
| | SST-5 | 0.39 | 1.17 | 33.01 | 50.78 | 16.50 | 43.85 |
| | AGNews | 1.37 | 38.48 | 59.57 | 75.29 | 48.92 | 53.91 |
| | Subjective | 0.00 | 0.00 | 0.00 | 69.33 | 15.14 | 53.71 |
| | Average | 13.44 | 27.86 | 57.29 | 74.84 | 29.82 | 62.14 |
| **Qwen 2.5-7B** | SST-2 | 48.05 | 20.12 | 84.77 | 93.07 | 54.98 | 75.20 |
| | MR | 69.53 | 0.59 | 4.30 | 92.48 | 65.72 | 83.40 |
| | FP | 0.78 | 14.65 | 71.88 | 68.65 | 26.07 | 41.80 |
| | SST-5 | 0.00 | 0.20 | 50.59 | 50.20 | 7.42 | 43.07 |
| | AGNews | 0.20 | 0.20 | 35.74 | 75.68 | 46.09 | 44.82 |
| | Subjective | 0.00 | 0.00 | 0.00 | 66.41 | 0.00 | 55.47 |
| | Average | 24.25 | 28.38 | 53.52 | 77.25 | 48.70 | 65.54 |

---

[6]Notice that the bias terms in the linear layer do not affect the covariance, so we omit them here.

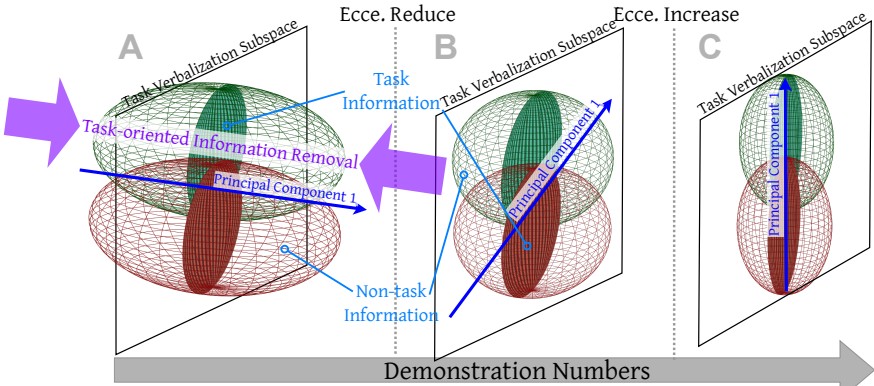

Figure 17: An illustration of task-oriented information removal with nonmonotonic eccentricities against the number of demonstrations. The ellipsoids refer to the hidden state point cloud with two ground-truth labels (distinguished in colors), which is separated even no demonstrations are given, according to previous works (Cho et al., 2025a; Yang et al., 2025).

## B    THE INFORMATION REMOVING MAGNITUDE OF LOW-RANK FILTER

**Principle of Covariance Loaded as the Lower Bound of Information Removal.**    Given that information theory suggests the covariance of a distribution can approximate its differential entropy (i.e., information content), we can compute the upper bound of the covariance remained after the filtering as a lower bound of the information removed by the filter. Evidently, this covariance upper bound can be taken as the variance captured by its top-$r$ principal components.

We utilize such a principle to estimate the gap between the real information removal to the lower bound: if the eigenvectors of $W_{\text{enc}}$ fall within the top principal component subspace of the hidden states point cloud, thus we judge that $W_{\text{enc}}$ can better preserve the information of point clouds, and vice versa. Therefore, we map every eigenvector of $W_{\text{enc}}$ to the orthonormal basis spanned by the top-64 principal components of the hidden state point cloud, and calculate the ratio of vector norms before and after the mapping, as shown in Fig. 25 - 30 for all the datasets on Llama 3.2-1B. In the results, the norm ratios remain low, suggesting that not all the covariance on the top-64 principal components of the hidden state point cloud can be preserved through the $W_{\text{enc}}$ filter, i.e., the $W_{\text{enc}}$ filter is an effective information removal towards the hidden states.

Although the TVS is composed of two parts $W_{\text{enc}}$ and $W_{\text{dec}}$, merely confirming that $W_{\text{enc}}$ performs significant information removal is sufficient to demonstrate that projecting the hidden state onto the $W_{\text{enc}}W_{\text{dec}}$ substantially reduces the amount of information contained in the hidden state point cloud.

## C    VISUALIZATION: THE DIRECTION OF INFORMATION REDUCTION

In §4.3, we observe that the eccentricities have a nonmonotonic pattern against the number of demonstrations on SST-2 and Llama 3.2-1B, while in some of augmentation results of Appendix I, e.g., results on MR and Llama 3.2-1B (Fig. 47), such nonmonotonic pattern can not be observed.

Our explanation is that: as shown in Fig. 17 (A), in some datasets, the distribution of the hidden states (as sentence embeddings (Cho et al., 2025a)) of zero-shot queries does not dominate on the task subspace, i.e., the first principal direction of such hidden state point clouds is orthogonal to the TVS[7]. So, the early information removal when a few demonstrations are given is from a direction near the first principal direction, which reduces the covariance loaded on the first principal direction, i.e., reduces the eccentricity. Such removal gradually reduces the dominance of task-irrelevant information, causing the first principal component of the hidden state point cloud to progressively approach the TVS, as shown in Fig. 17 (B). At this point, the eccentricity reaches its minimum

---

[7]In this visualization and the experiments of this section, we implicitly make the assumption that TVS maximizes, or at least preserves, the separability of clusters produced by queries of different labels. This assumption is acceptable: if TVS were to confound this separability, the mapping results on TVS would fail to effectively distinguish different labels, which would contradict the high accuracy observed in Fig. 3.

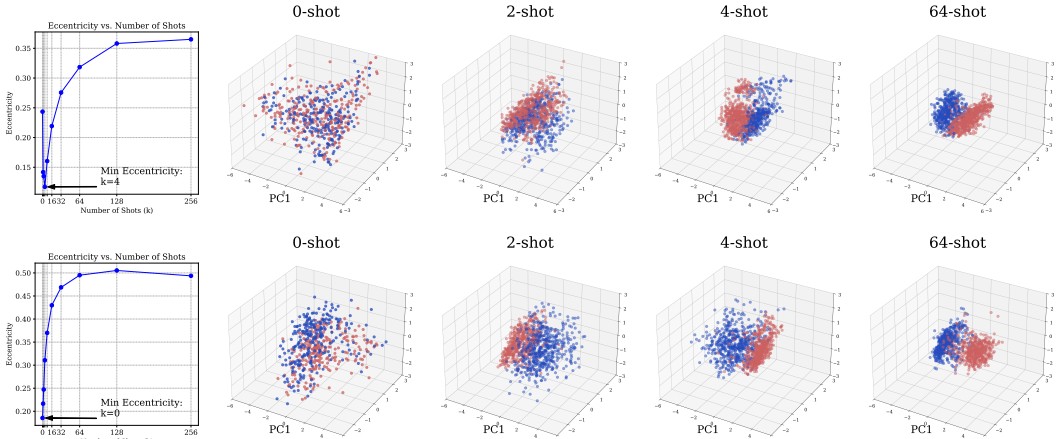

Figure 18: Visualization of the last token's hidden states from Layer 13 of Llama 3.2-1B, (**upper**) on SST-2 with a nonmonotonic eccentricity, and (**lower**) on MR with a monotonic eccentricity. The colors refer to the queries' ground-truth label.

and then begins to increase (Fig. 17 (C)) gradually. While, in other datasets, the distribution of the hidden states of zero-shot queries naturally dominates on or near the TVS, that is, the information removal begins from Fig. 17 (B), causing a monotonic increase in the eccentricity.

We prove our aforementioned inference by visualizing the last token hidden state point clouds from Layer 13 of Llama 3.2-1B on SST-2 (nonmonotonic) and MR (monotonic) with various demonstration numbers, as shown in Fig. 18. In this visualization, on SST-2, the separability occurs on the 3rd principal component at the beginning of demonstrating, and turns to the 1st principal component on $k = 4$, where the eccentricity reaches the minimum. However, on MR, the separability occurs on the 1st principal component at the beginning, causing a monotonic eccentricity. Such observations corroborate the process illustrated in Fig. 18 and confirm an implicit yet important property: task-oriented information removal preserves the distribution along the separable directions of the hidden state point cloud, while compressing the distribution along the non-separable directions.

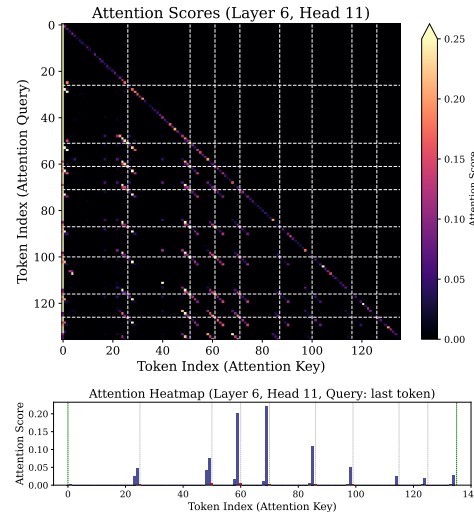

Figure 19: (Upper) attention map; (lower) attention scores to the last token (":", as the attention query) of Layer 6 Head 11.

Moreover, we consider the reason for such a distribution difference on SST-2 and MR as the difference in the inputs. We observe some inputs in both datasets, as shown below:

(Test samples from SST-2)
- video, and.
- stirs us as well.
- all of Dean's mannerisms and self-indulgence,

(Test samples from MR.)
- Audiard successfully maintains suspense on different levels throughout a film that is both gripping and compelling.
- The problem with the mayhem in Formula 51 is not that it's offensive, but that it's boring.
- Doesn't deliver a great story, nor is the action as gripping as in past Seagal films.

<|begin_of_text|> sentence : most ple asurable expressions of pure movie sentiment : positive
sentence : by a central plot that ' s pepper ed with false starts and populated by characters who are nearly impossible to care about sentiment : negative
sentence : is a small but rewarding comedy that takes aim at contemporary southern adolescence and never lets up . sentiment : positive
sentence : she ' s cut open a vein sentiment : negative
sentence : la con ic and very st ilt ed in its dialogue , this indie flick never found its audience , probably because it ' s extremely hard to relate to any of the characters . sentiment : negative
sentence : can watch , gig gle and get an adrenaline boost without feeling like you ' ve completely lowered your entertainment standards sentiment : positive
sentence : wo n 't join the pan theon of great monster /sc ience fiction flick s that we have come to love sentiment : negative
sentence : called `` jar -j ar b inks : the movie sentiment : negative
sentence : barely shocking , barely interesting and most sentiment :

<|begin_of_text|> sentence : takes itself all too seriously sentiment : negative
sentence : talented enough and sentiment : positive
sentence : the movie starts with a legend and ends with a story that is so far -f etched it would be impossible to believe if it were n 't true . sentiment : negative
sentence : a surprisingly an emic disappointment sentiment : negative
sentence : like this movie sentiment : positive
sentence : a ro te exercise in both animation and storytelling . sentiment : negative
sentence : the emperor ' s club any time sentiment : positive
sentence : the sentimental script has problems , but the actors pick up the slack sentiment : positive
sentence : `` home movie '' is a sweet treasure and something well worth your time . sentiment :

<|begin_of_text|> sentence : heart w arming , non jud gment al kind sentiment : positive
sentence : it lacks the utter authority of a genre gem sentiment : negative
sentence : riv eting memories are rendered with such clarity that it ' s as if it all happened only yesterday sentiment : positive
sentence : freedom the ir anian people already possess , with or without access to the ballot box sentiment : positive
sentence : filled with more holes than c ly de bar row ' s car sentiment : negative
sentence : really , really good sentiment : positive
sentence : ` hungry -man portions of bad ' sentiment : negative
sentence : the wonderfully lush mor vern call ar is pure punk existential ism , and ms . r ams ay and her co -w riter , li ana d ogn ini , have dram at ized the alan war ner novel , which itself felt like an answer to ir vine wel sh ' s book trains pot ting . sentiment : positive
sentence : can be classified as one of those ` alternate reality ' movies ... except that it would have worked so much better dealing in only one reality . sentiment :

Figure 21: Attention score visualization of DH (Llama 3.2-1B Layer 9 Head 15), where the last token serves the attention query.

MR examples show clear sentiment tendencies, absent in SST-2, likely explaining the hidden state distribution difference between the two datasets. Moreover, such a distributional property, i.e., signal-noise-ratio describing whether the zero-shot hidden states are in the status of Fig. 17 (A) or (C), can be designed into a metric to characterize the task difficulty. We leave such a discussion to future work.

## D  CASE ANALYSIS FOR DHs' MECHANISM

In this section, we provide prototypical evidence for the mechanism of DH, i.e., locating the important query tokens with task-related information by the attention scores, where the hidden states processed by the $W_Q^\top W_K$ serve as a detector of such tokens. In detail, we visualize the attention scores of Llama 3.2-1B Layer 9 Head 15 on the last tokens in some cases from SST-2, as shown in Fig. 21. In the visualization, it is obvious that the attention scores are concentrated on some sentiment-related tokens, which contain the task-related information for the sentiment analysis task defined on SST-2. This result confirms that DHs can leverage attention scores to correctly filter hidden states generated by tokens containing task-relevant information, thereby amplifying task-related representations, so

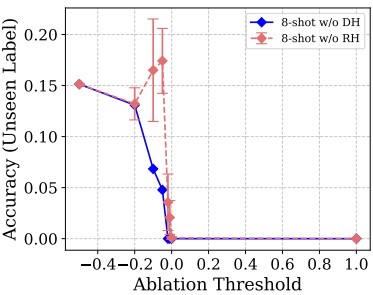

Figure 20: DH / RH ablation results against ablation threshold.

as to relatively reduce task-irrelevant information. In this process, we can equivalently interpret $W_Q^\top W_K$ as extracting information from a certain subspace of the last-token hidden state, which encodes the criterion for determining task information, enabling the subsequent dot-product operations with each hidden state vector to select task-relevant information.

## E  DISCUSSION ON THE SELECTION OF ABLATION THRESHOLD

The ablation experiments in §4.4 utilize a fixed threshold of $-5\%$, which might be suspected to be arbitrary. Therefore, this section aims to demonstrate that the ablation results are insensitive to this threshold, showing that any *mild* threshold is sufficient to distinguish DH ablation from random ablation. In detail, we vary this threshold and retest the results of ablating DHs and random heads

with the same amount of DH. The results on Llama 3.2-1B SST-2 unseen setting are shown in Fig. 20, where thresholds between -0.2 and 0 (note that we use the relative change rate of covariance flux as the metric, so this is a fairly broad range) all successfully identify the causal effect of DHs against RHs. While, if the threshold is too strict (extremely small values), no head will be ablated, and an overly loose threshold will cause many or even all heads to be ablated. Therefore, we select -0.05 within this range as the experimental setting.

## F  "IMPLICIT STEERING HEADS" DISCOVERED IN §4.4

Since we test the ablation effect on the accuracy of all the attention heads in §4.4, we can detect more important heads rather than induction heads or denoising heads as an interesting supplement. As shown in Fig. 59 - 64, ablating Head 11 in Layer 6 causes a very significant drop in accuracy; therefore, we examine its attention map under the aforementioned setting shown in Appendix A.4, as shown in Fig. 19. We find that the attention map exhibits a localized diagonal pattern that recurs periodically with the location of label tokens. Ignoring the attention sink (Gu et al., 2025), this pattern reflects the behavior of the head: locating tokens in the preceding context that are identical to the current token (more evident in Fig. 19 (lower), where nonzero attention scores appear almost exclusively at the ":" positions, identical to the current token). Consequently, the attention map produces a periodic diagonal pattern around each label token, since the tokens at these positions are exactly the same.

Intuitively, at the final token (":"), this attention head copies information from all preceding ":" hidden states to the current position, where, each ":" stores the inference result of the previous shot round (e.g., in an 8-shot input, due to the causal mask, the 8th ":" position is effectively the output position of all preceding tokens, i.e., a 7-shot ICL input, and thus stores the previous round's inference result). In other words, this attention head functions similarly to "task vector steering": it implicitly steers the hidden state of the previous ":" (commonly used as a task vector in prior work (Hendel et al., 2023)) to the current ":" and aggregates it into an updated task vector. This may partly explain why more demonstrations can yield better accuracy, requiring further discussion.

## G  GROUNDING THE METRICS TO INFORMATION MEASUREMENT

Since we utilize covariance-based metrics to measure the information-based quantity, which is usually measured by entropy, here we prove that the covariance is a linear-scale surrogate of entropy.

Notice that we need to handle the scenario with low-rank inputs (e.g., the point set mapped by the low-rank filter $W_{\text{enc}}W_{\text{dec}}$), we define entropy on Hausdorff measure, instead of the typical differential entropy defined on Lebesgue measure, to avoid the entropy calculated being $-\infty$.

**Definition 1** (Hausdorff differential entropy on $d$-dimensional $r$-rank subspace). *Let $x \in \mathbb{R}^d$ be a random vector sampled from Gaussian $X \sim \mathcal{N}(\mu, \Sigma)$, where the rank of $\Sigma$ is $r \leqslant d$. Let $\mathcal{H}^r$ denote the $r$-dimensional Hausdorff measure, embedded on the subspace $\mathcal{M} = \mu + \text{im}(\Sigma)$. If $p$ is the density of $X$ on $\mathcal{H}^r$, the **Hausdorff differential entropy** of $X$ is defined as*

$$h_{\mathcal{H}^r}(X) = - \int_{\mathcal{M}} p(x) \log p(x) \, \mathrm{d}\mathcal{H}^r(x). \tag{2}$$

On such an entropy measurement, we can build the link to the covariance.

**Theorem 1** (Non-zero eigenvalues as Hausdorff entropy estimator). *Let $x \in \mathbb{R}^d$ be a random vector sampled from Gaussian $X \sim \mathcal{N}(\mu, \Sigma)$, where the rank of $\Sigma$ is $r \leqslant d$. Let $\lambda_1, \ldots, \lambda_r > 0$ as the non-zero eigenvalues of $\Sigma$. Then the Hausdorff differential entropy $h_{\mathcal{H}^r}(X)$ is given by*

$$h_{\mathcal{H}^r}(X) = \frac{r}{2}(\log 2\pi + 1) + \frac{1}{2}\sum_{i=1}^{r} \log \lambda_i \tag{3}$$

The Theorem 1 shows that: the covariance loaded on each principal direction ($\lambda_i$) can be a positive correlated measurement of the information (entropy) contribution ($\log \lambda_i$) on such direction, which is the theoretical grounding of our metrics design.

Table 7: Fine-grained data for Table 2.

| | | Random Sample | | | Unseen Labels | | | Seen Labels | | |
|---|---|---|---|---|---|---|---|---|---|---|
| | | 8-shot | 8-shot w/o DH | 8-shot w/o RH | 8-shot | 8-shot w/o DH | 8-shot w/o RH | 8-shot | 8-shot w/o DH | 8-shot w/o RH |
| Llama 3.2-1B | SST-2 | 87.40 | 61.42 | $84.04_{1.61}$ | 15.14 | 4.78 | $17.42_{3.18}$ | 87.79 | 62.10 | $68.73_{1.84}$ |
| | MR | 90.23 | 81.25 | $88.60_{4.67}$ | 44.92 | 1.27 | $28.01_{11.81}$ | 90.53 | 81.64 | $65.54_{2.10}$ |
| | FP | 73.11 | 67.38 | $73.26_{3.69}$ | 0.39 | 0.00 | $0.68_{0.71}$ | 75.00 | 67.97 | $59.42_{4.41}$ |
| | SST-5 | 42.19 | 30.08 | $39.28_{2.08}$ | 2.93 | 0.29 | $3.46_{1.34}$ | 48.63 | 33.98 | $61.79_{7.77}$ |
| | AGNews | 71.78 | 36.72 | $63.87_{9.17}$ | 23.83 | 0.00 | $14.32_{10.83}$ | 76.95 | 39.94 | $77.32_{2.25}$ |
| | Subjective | 61.42 | 53.42 | $54.71_{1.81}$ | 0.00 | 0.00 | $0.00_{0.00}$ | 61.52 | 53.51 | $81.30_{3.93}$ |
| | Average | 71.02 | 55.05 | $67.29_{3.84}$ | 14.54 | 1.06 | $10.65_{4.65}$ | 73.40 | 56.52 | $69.02_{3.72}$ |
| Llama 3-8B | SST-2 | 91.31 | 86.23 | $90.94_{0.76}$ | 27.14 | 2.83 | $26.46_{4.39}$ | 91.89 | 86.91 | $91.50_{0.45}$ |
| | MR | 93.07 | 91.31 | $92.80_{0.48}$ | 52.83 | 33.69 | $40.67_{14.61}$ | 93.46 | 91.60 | $93.12_{0.54}$ |
| | FP | 82.71 | 79.59 | $82.35_{0.55}$ | 9.57 | 4.69 | $8.84_{0.76}$ | 84.96 | 82.13 | $85.47_{0.46}$ |
| | SST-5 | 45.31 | 43.16 | $45.73_{0.48}$ | 6.73 | 3.91 | $6.01_{1.29}$ | 54.39 | 51.95 | $54.83_{0.42}$ |
| | AGNews | 75.58 | 60.74 | $74.95_{0.96}$ | 33.69 | 0.00 | $31.67_{2.83}$ | 80.27 | 66.70 | $78.30_{1.32}$ |
| | Subjective | 77.83 | 71.29 | $71.56_{2.54}$ | 0.00 | 0.00 | $0.00_{0.00}$ | 77.83 | 71.39 | $74.97_{4.61}$ |
| | Average | 77.64 | 72.05 | $76.38_{0.96}$ | 21.66 | 7.52 | $18.94_{3.98}$ | 80.47 | 75.11 | $79.70_{1.30}$ |
| Qwen 2.5-3B | SST-2 | 91.80 | 70.60 | $92.30_{0.49}$ | 21.88 | 0.68 | $43.49_{6.79}$ | 92.09 | 71.09 | $92.65_{0.54}$ |
| | MR | 89.06 | 81.05 | $89.94_{1.14}$ | 46.00 | 0.88 | $48.24_{10.51}$ | 91.11 | 81.44 | $90.33_{1.43}$ |
| | FP | 77.73 | 79.00 | $79.45_{1.01}$ | 23.92 | 27.25 | $25.31_{1.19}$ | 79.49 | 80.37 | $80.35_{1.07}$ |
| | SST-5 | 49.71 | 40.53 | $47.13_{1.81}$ | 5.47 | 1.07 | $5.43_{2.19}$ | 55.76 | 46.09 | $53.44_{2.04}$ |
| | AGNews | 74.80 | 74.90 | $74.22_{1.63}$ | 43.65 | 38.96 | $38.46_{2.21}$ | 78.02 | 78.52 | $77.42_{1.80}$ |
| | Subjective | 60.74 | 64.36 | $62.81_{1.68}$ | 0.00 | 0.68 | $0.00_{0.00}$ | 62.01 | 64.36 | $63.94_{2.52}$ |
| | Average | 73.97 | 68.41 | $74.31_{1.30}$ | 23.49 | 11.59 | $26.82_{3.82}$ | 76.41 | 70.31 | $76.36_{1.57}$ |
| Qwen 2.5-3B Ins | SST-2 | 90.33 | 89.94 | $90.78_{0.61}$ | 69.04 | 38.28 | $66.95_{7.40}$ | 90.43 | 90.04 | $89.96_{0.65}$ |
| | MR | 89.84 | 85.53 | $88.65_{1.08}$ | 64.84 | 29.49 | $66.54_{6.87}$ | 89.94 | 86.13 | $89.26_{0.52}$ |
| | FP | 87.89 | 85.94 | $88.61_{0.39}$ | 77.73 | 73.82 | $73.65_{3.01}$ | 88.57 | 87.89 | $88.24_{0.86}$ |
| | SST-5 | 50.78 | 48.54 | $49.96_{0.55}$ | 16.50 | 10.06 | $13.97_{2.32}$ | 56.44 | 54.59 | $56.33_{0.34}$ |
| | AGNews | 75.29 | 75.29 | $74.31_{1.20}$ | 48.92 | 43.26 | $43.65_{4.06}$ | 78.32 | 78.22 | $78.09_{1.28}$ |
| | Subjective | 69.33 | 68.16 | $71.35_{7.54}$ | 15.14 | 30.17 | $13.18_{3.43}$ | 69.43 | 68.26 | $66.44_{1.16}$ |
| | Average | 77.24 | 75.57 | $77.28_{1.90}$ | 48.70 | 37.51 | $46.32_{4.51}$ | 78.86 | 77.52 | $78.05_{0.80}$ |

*Proof.* Since $\Sigma$ has rank $r$, we write its eigen-decomposition as

$$\Sigma = Q \begin{pmatrix} \Lambda_r & 0 \\ 0 & 0 \end{pmatrix} Q^\top, \qquad \Lambda_r = \mathrm{diag}(\lambda_1, \ldots, \lambda_r). \tag{4}$$

On the Hausdorff measure $\mathcal{H}^r$ on $\mathcal{M}$, the probability density function of $X$ is

$$p(x) = \frac{1}{(2\pi)^{r/2} \det(\Lambda_r)^{1/2}} \exp\left(-\frac{1}{2}(x - \mu)^\top \Sigma^+ (x - \mu)\right), \tag{5}$$

where $\Sigma^+$ is the Moore-Penrose pseudoinverse of $\Sigma$.

The Hausdorff differential entropy is therefore

$$h_{\mathcal{H}^r}(X) = -\mathbb{E}[\log p(X)] \tag{6}$$

$$= \frac{r}{2}\log(2\pi) + \frac{1}{2}\log\det(\Lambda_r) + \frac{1}{2}\mathop{\mathbb{E}}_{x \sim X}\left[(x - \mu)^\top \Sigma^+ (x - \mu)\right]. \tag{7}$$

Since $\Sigma^+$ inverts $\Lambda_r$ on the image of $\Sigma$ and annihilates the orthogonal complement, we obtain

$$\mathop{\mathbb{E}}_{x \sim X}\left[(x - \mu)^\top \Sigma^+ (x - \mu)\right] = \mathrm{tr}(\Sigma^+ \Sigma) = \mathrm{tr}(I_r) = r. \tag{8}$$

Therefore,

$$h_{\mathcal{H}^k}(X) = \frac{r}{2}\log(2\pi) + \frac{1}{2}\log\det(\Lambda_r) + \frac{r}{2} \tag{9}$$

$$= \frac{r}{2}(\log 2\pi + 1) + \frac{1}{2}\sum_{i=1}^{r} \log \lambda_i \tag{10}$$

$\square$

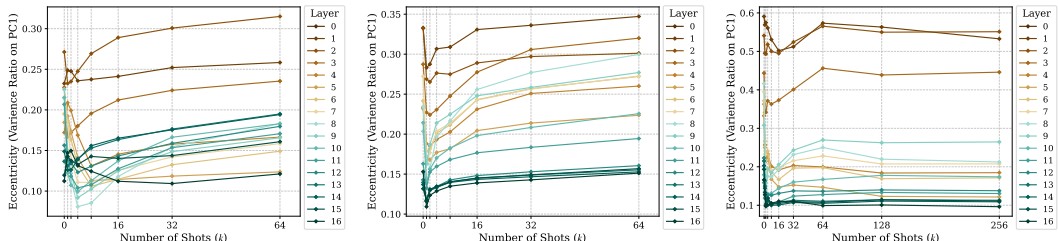

Figure 22: Eccentricity results of 3 non-classification tasks. **(Left)** people-profession; **(Middle)** opus-100 translation: English to Chinese; **(Right)** country-capital.

# H  PROTOTYPE OF INFORMATION REMOVAL ON GENERATIVE SCENARIO

In §5, we discuss some infinite label space settings, i.e., the country-capitals task and the people-profession task, but still on one-token or few-token labels. Therefore, in this section, we generalize these discussions to the generative tasks. In summary, we observe similar information removal dynamics on generative tasks with classification tasks (Fig. 4 (middle)). However, as acknowledged in the Limitation (4) (§5), we can not obtain a linear information bottleneck to evaluate the correctness of such information removal (i.e., Covariance Flux). Therefore, this experiment serves only as a prototypical observation intended to motivate subsequent research within our framework.

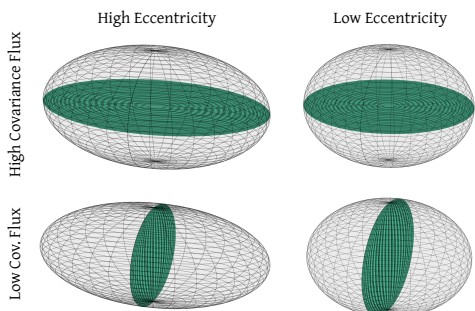

Figure 23: Diagram for eccentricity and covariance flux metrics. The green plain refers to the TVS ($W_{enc}W_{dec}$) obtained in §4.2.

Specifically, we perform tests on the opus-100 (Zhang et al., 2020; Tiedemann, 2012) English–Chinese translation dataset and Llama 3.2-1B that follow the same procedure of eccentricity calculation of Step 2 described in §3, yielding the results shown in Fig. 22 (middle). Note that translation is not a bijective task (i.e., a single translation may correspond to multiple possible source sentences). Therefore, the translation task exhibits a clear information-removal trend. This is intuitive: with the prompt template "sentence: [original text], translation: [translated text]", the target language is not explicitly specified in zero-shot inputs. Thus, at the last token, the model encodes verbalization patterns corresponding to all potential target languages, with the *default* pattern dominating. Once some demonstrations are provided, non-target verbalizations are removed. If the specified target language differs from the "default" pattern, this produces the decrease-and-increase pattern as observed, following the description in Appendix C.

Such results directly confirm the information removal dynamics in task recognition driven by ICL demonstrations for generative and non-bijective tasks. However, it is conceivable that the outputs of such generative tasks are distributed on a manifold (to be compared, the outputs of classification tasks are distributed in the space spanned by the unembedding vectors of their labels). Therefore, utilizing a simple linear filter to actively remove task-irrelevant information and locate denoising heads is difficult, as shown in Fig. 24, which echoes our Limitation 4. Future work can intuitively utilize more complex information bottlenecks to extend our framework to the non-linear tasks.

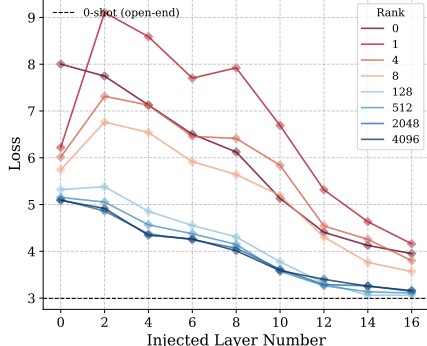

Figure 24: Filter-injection evaluations on opus-100 English–Chinese translation task. **In cross-entropy loss.**

Moreover, we repeat the experiment on the people-profession task and country-capital shown in Fig. 22 (left) and (right). In these results, we can observe a clear information removal dynamic in the people–profession task, while the country–capital task shows no visible interval of increasing eccentricity, which is consistent with our earlier discussion in §5.

# I  SUPPLEMENTARY MATERIALS OUTLINE

**Augmentation Results for Fig. 3 ("Filter Injection").**   We repeat the experiment shown in Fig. 3 on all the mentioned settings, as shown in Fig. 31 - 36, which are consistent with Fig. 3.

**Augmentation Results for the Scatter Size of Fig. 3 ("Filter Injection: Covariance out of Rank $r$").**   We explicitly visualize the numerical results of the lower-bound of information removal (i.e., the covariance rate out of rank) shown as the scatter size in Fig. 3, as shown in Fig. 42 - 46[8].

**Augmentation Results for Table 1 ("Symbolic Fine-tuning").**   We repeat the experiment shown in Table 1 on Llama 3-8B and Qwen 2.5-3B, as shown in Fig. 39, which are consistent with Table 1.

**Augmentation Results for Fig. 4 ("Eccentricity and Covariance Flux against $k$").**   We repeat the experiment shown in Fig. 4 on all the mentioned settings, as shown in Fig. 47 - 49. The results are consistent with Fig. 4.

**Augmentation Results for Fig. 5 and 6 ("Eccentricity and Covariance with Instruction and Various Labels").**   We repeat the experiment shown in Fig. 5 and 6 on all the mentioned settings, as shown in Fig. 53 - 58. The results are consistent with Fig. 5 and 6.

**Accuracy with Instruction and Label Configuration in Fig. 5 and 6.**   We report the test accuracies on the input configurations in Fig. 5 and 6 in Table 6.

**Augmentation Results for Fig. 7 ("Finding Denoising Heads").**   We repeat the experiment shown in Fig. 7 on all the datasets with 8-shot inputs and Llama 3.2-1B, Llama 3-8B, Qwen 2.5-3B, Qwen 2.5-3B Instruct in Fig. 59 - 82.

**Augmentation Results for Fig. 8 ("Layer Distribution of Denoising Heads").**   We repeat the visualization shown in Fig. 8 on all the datasets and Llama 3.2-1B, Llama 3-8B, Qwen 2.5-3B, Qwen 2.5-3B Instruct in Fig. 83 - 86.

**Augmentation Results for Fig. 11 ("Denoising Head Overlap among Datasets").**   We repeat the visualization shown in Fig. 11 on Llama 3-8B and Qwen 2.5-3B in Fig. 40.

**Augmentation Results for Fig. 9 ("Overlap of Induction Heads and Denoising Heads").**   We repeat the visualization shown in Fig. 9 on all the datasets and Llama 3.2-1B, Llama 3-8B, Qwen 2.5-3B, Qwen 2.5-3B Instruct in Fig. 87 - 90.

**Augmentation Results for Fig. 10 ("Attention Map Visualization").**   We repeat the visualization shown in Fig. 10 on more 2 denoising heads as shown in Fig. 41.

**Augmentation Results for Table 2 ("Ablation Experiment").**   We list the ablation results on individual datasets rather than the average ones in Table 7.

**Augmentation Results for Fig. 12 and 13 ("Factor-recall Filter Injection").**   We repeat the experiment shown in Fig. 12 and 13 on Llama 3-8B, Qwen 2.5-3B and 7B as shown in Fig. 37 and 38.

---

[8]The curve with $r = 8$ is usually plotted at a higher resolution, i.e., one point is drawn for each layer (if $r \neq 8$, we sometimes draw one point for each 2 layers). Therefore, in some cases, the $r = 8$ curve appears above curves with lower ranks since the scree plot is usually convex.

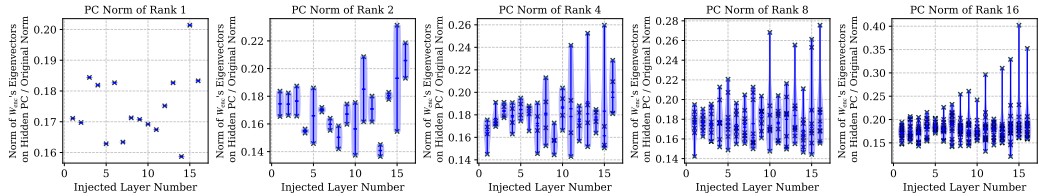

Figure 25: The cosine similarity between all the eigenvectors (each for a scatter) of $W_{\text{enc}}$ on specific layers and the subspace spanned by the top-64 principal components of the hidden state point cloud, measured by the ratio of vector norms before and after mapping to the principal subspace. On Llama 3.2-1B, SST-2, repeated on various inner ranks of TVS, i.e., the column dimensionality of $W_{\text{enc}}$.

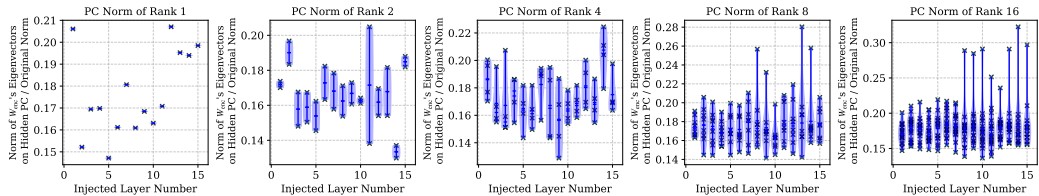

Figure 26: The similar result with Fig. 25 on Llama 3.2-1B, MR.

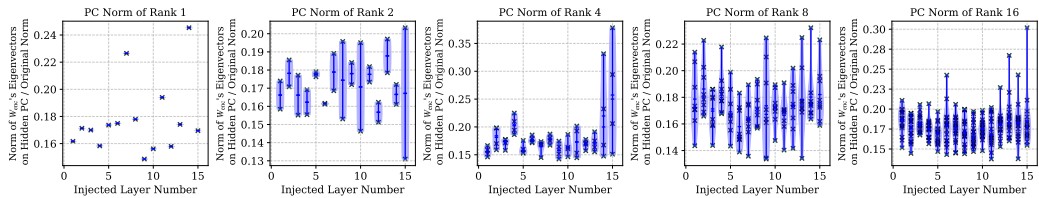

Figure 27: The similar result with Fig. 25 on Llama 3.2-1B, FP.

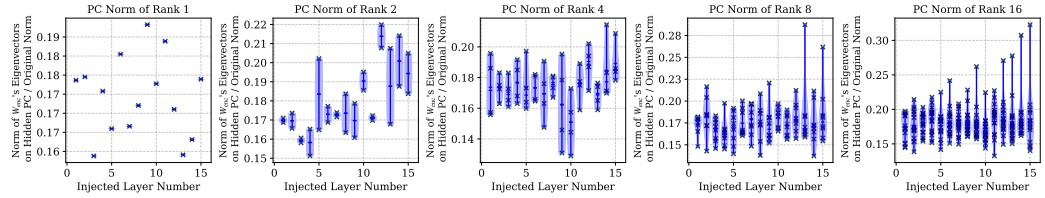

Figure 28: The similar result with Fig. 25 on Llama 3.2-1B, SST-5.

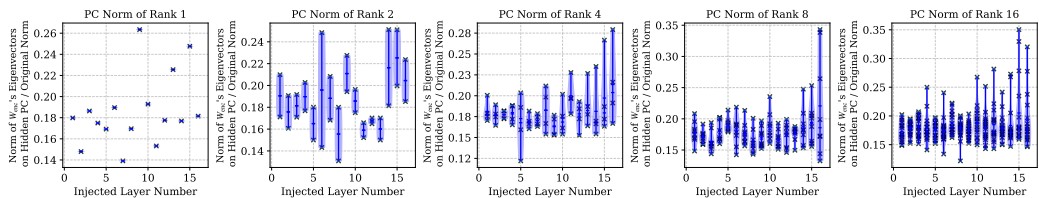

Figure 29: The similar result with Fig. 25 on Llama 3.2-1B, AGNews.

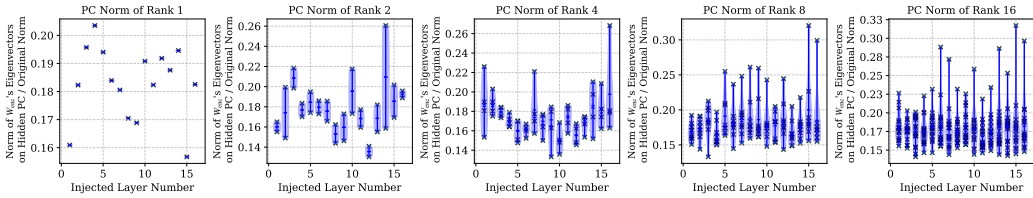

Figure 30: The similar result with Fig. 25 on Llama 3.2-1B, Subjective.

# Supplementary Materials

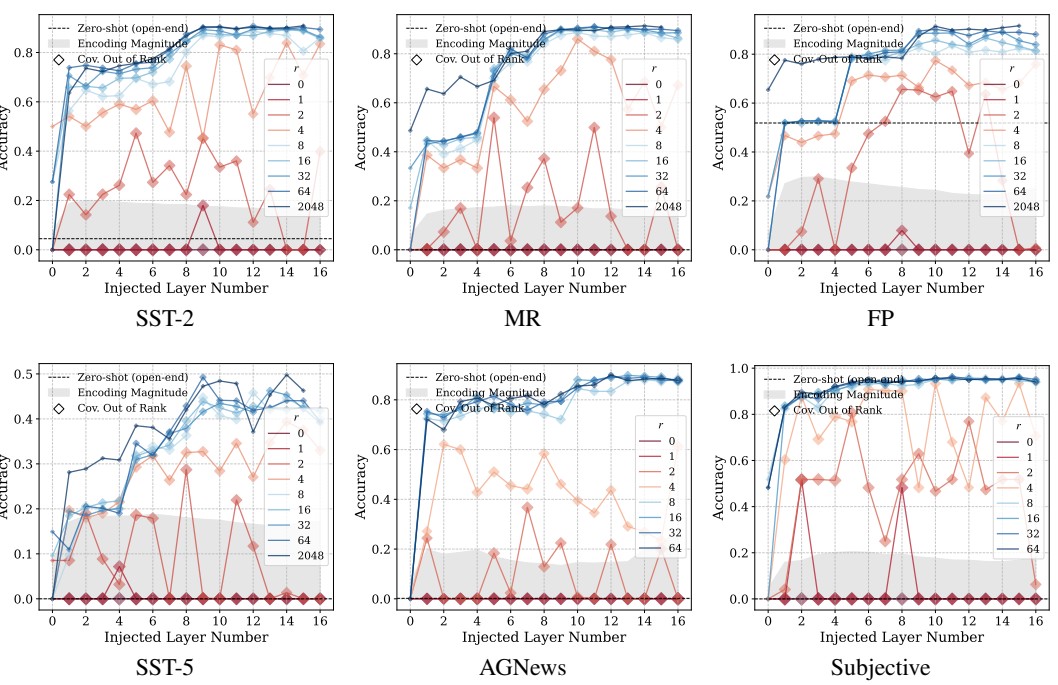

Figure 31: Augmentation results for Fig. 3 on Llama 3.2-1B.

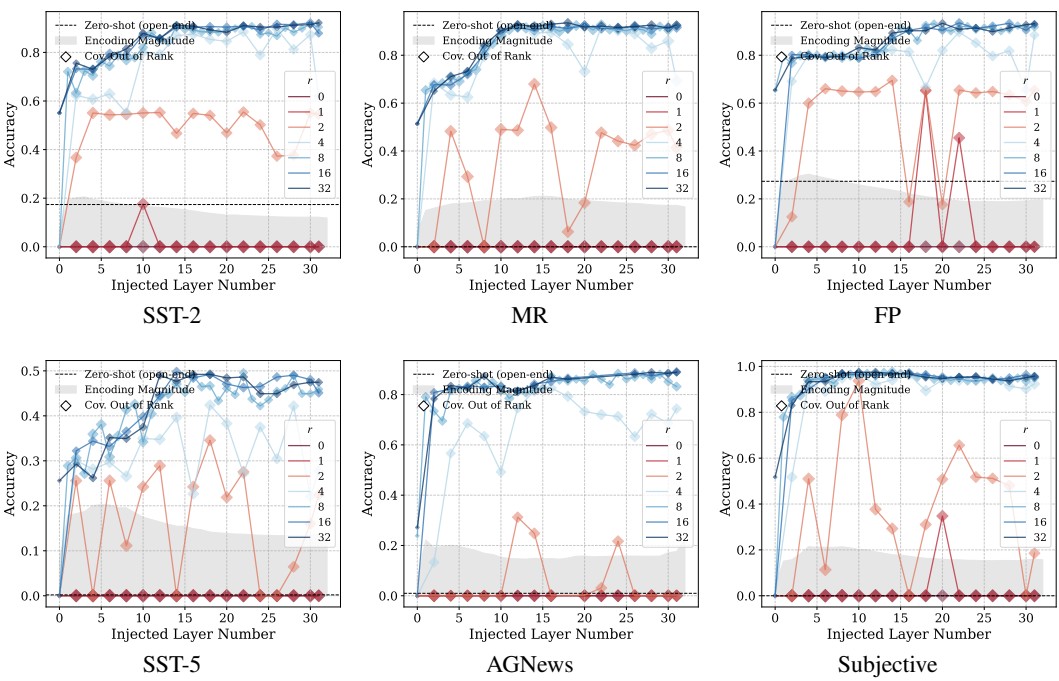

Figure 32: Augmentation results for Fig. 3 on Llama 3-8B.

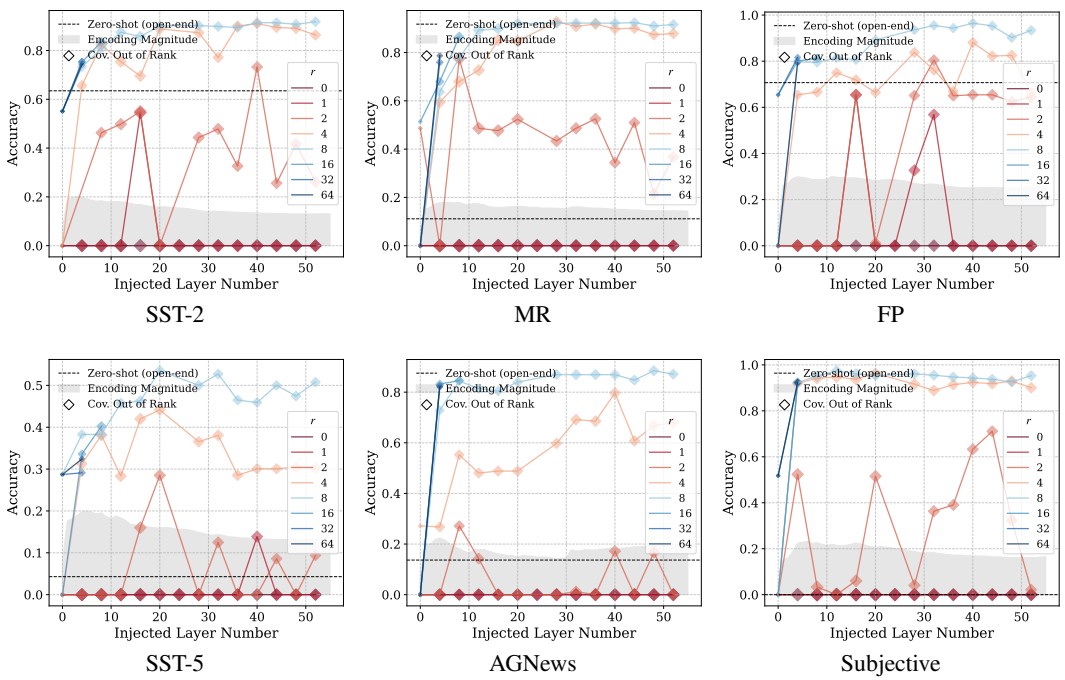

Figure 33: Augmentation results for Fig. 3 on Llama 3-13B.

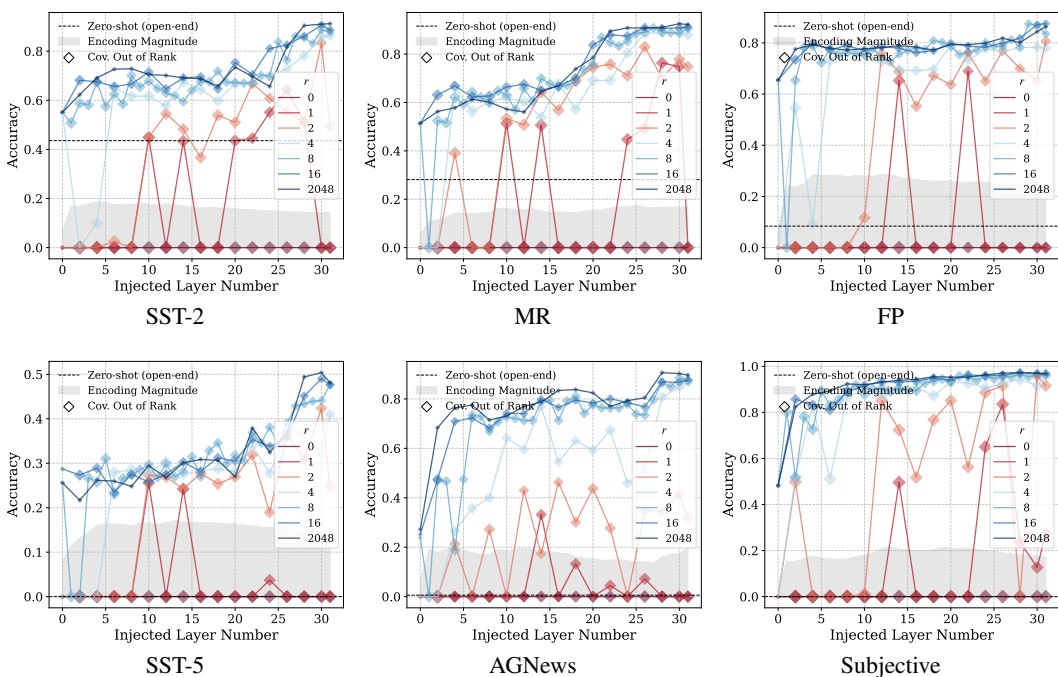

Figure 34: Augmentation results for Fig. 3 on Qwen 2.5-3B.

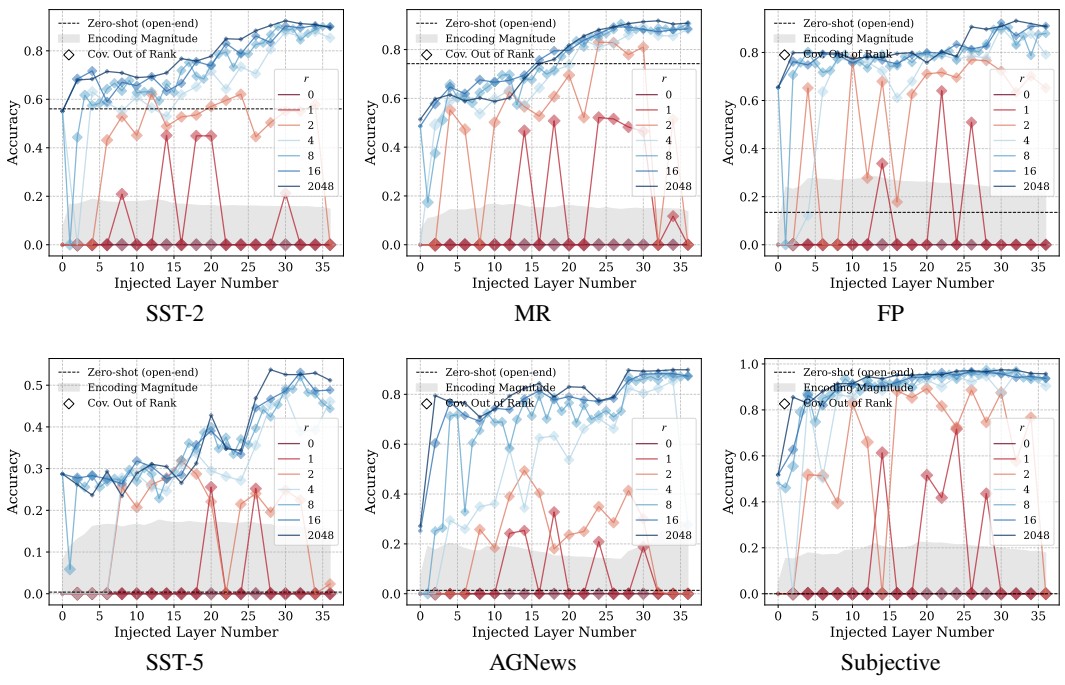

Figure 35: Augmentation results for Fig. 3 on Qwen 2.5-3B Instruct.

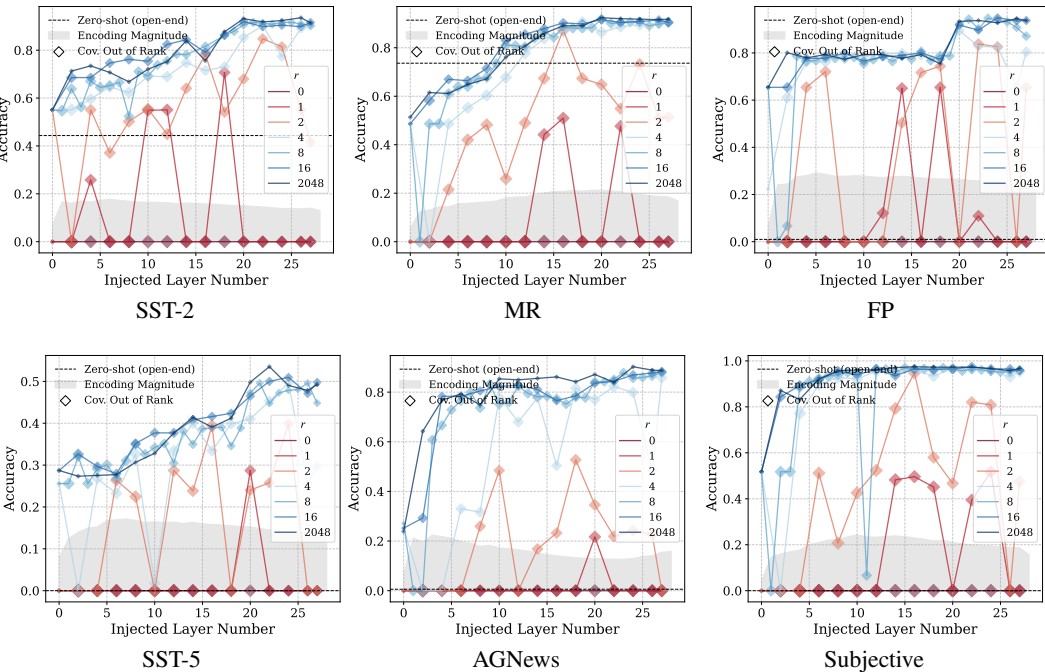

Figure 36: Augmentation results for Fig. 3 on Qwen 2.5-7B.

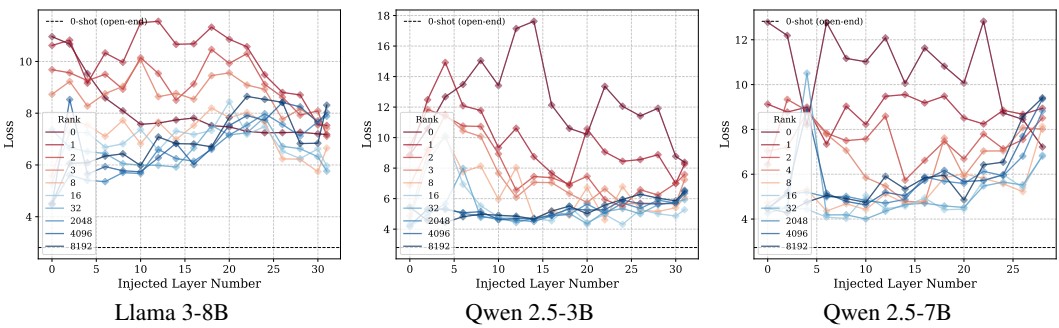

Figure 37: Augmentation results for Fig. 12 on 3 models.

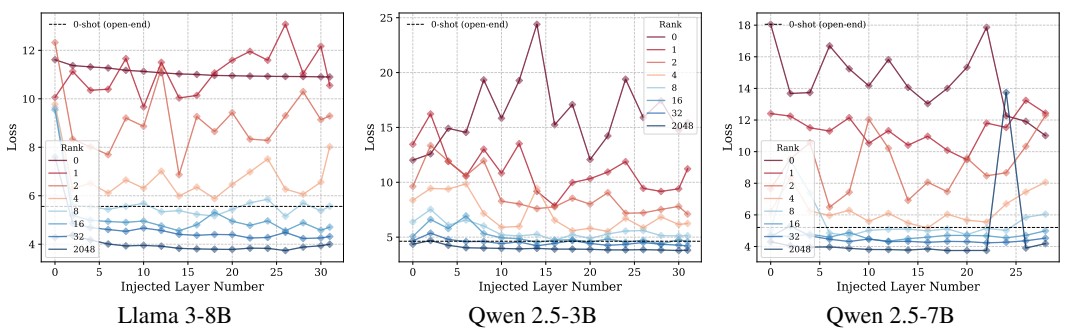

Figure 38: Augmentation results for Fig. 13 on 3 models.

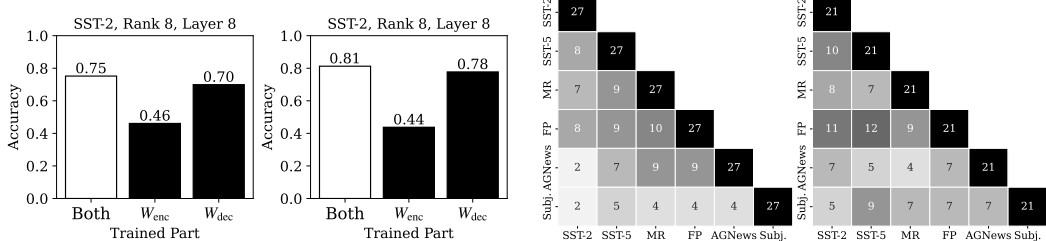

Figure 39: Augmentation results for Table 1, (left) on Qwen 2.5-3B, (right) on Llama 3-8B.

Figure 40: Augmentation results for Fig. 11, (left) on Qwen 2.5-3B, (right) on Llama 3-8B.

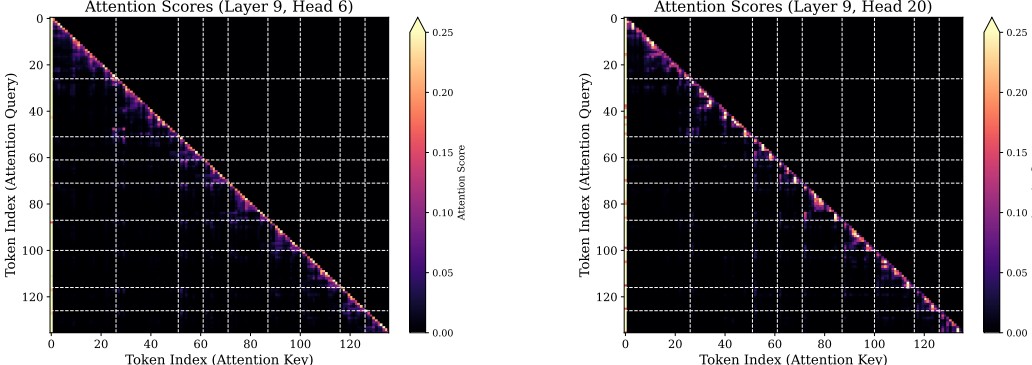

Figure 41: Augmentation results for attention visualization on (left) Layer 9, Head 6, (right) Layer 9, Head 20, with the inputs shown in Fig. 16.

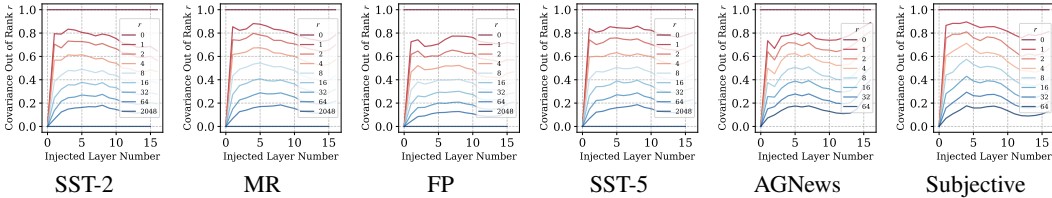

Figure 42: Numerical results of covariance out of rank on Llama 3.2-1B.

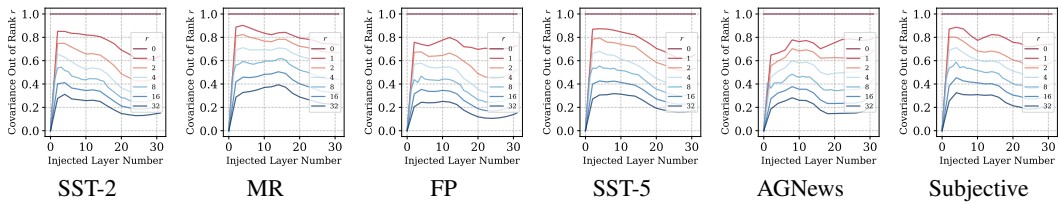

Figure 43: Numerical results of covariance out of rank on Llama 3-8B[8].

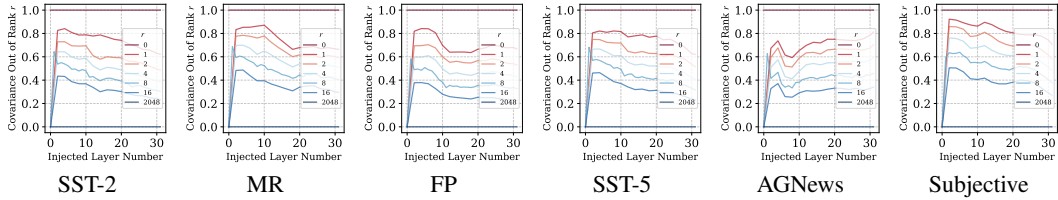

Figure 44: Numerical results of covariance out of rank on Qwen 2.5-3B[8].

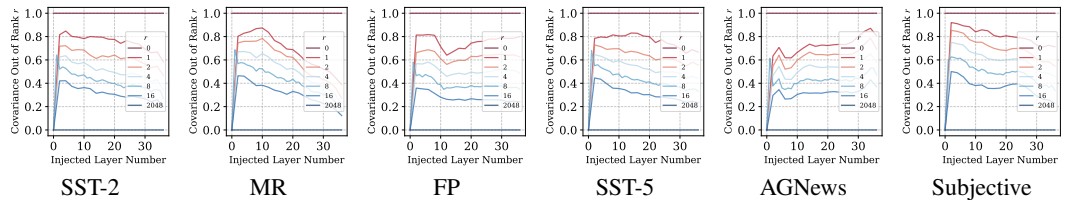

Figure 45: Numerical results of covariance out of rank on Qwen 2.5-3B Instruct[8].

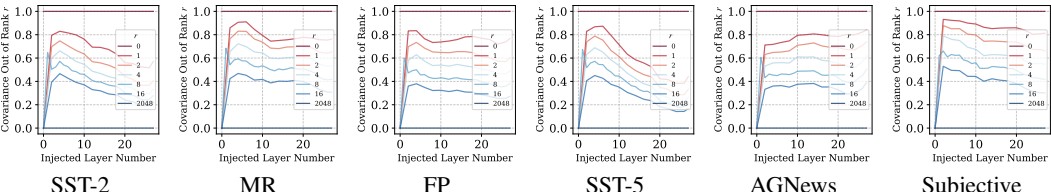

Figure 46: Numerical results of covariance out of rank on Qwen 2.5-7B[8].

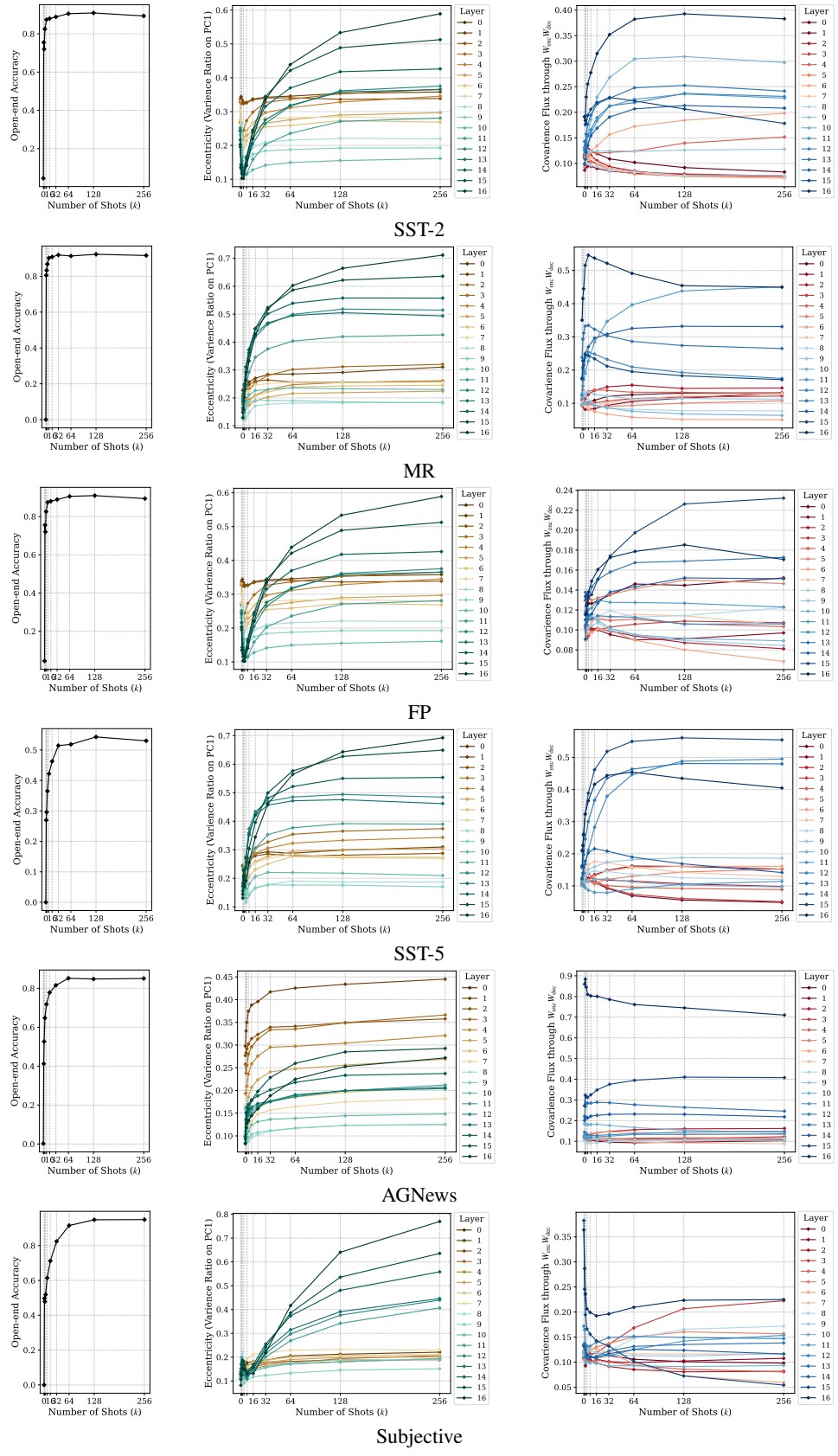

Figure 47: Augmentation results for Fig. 4 on Llama 3.2-1B.

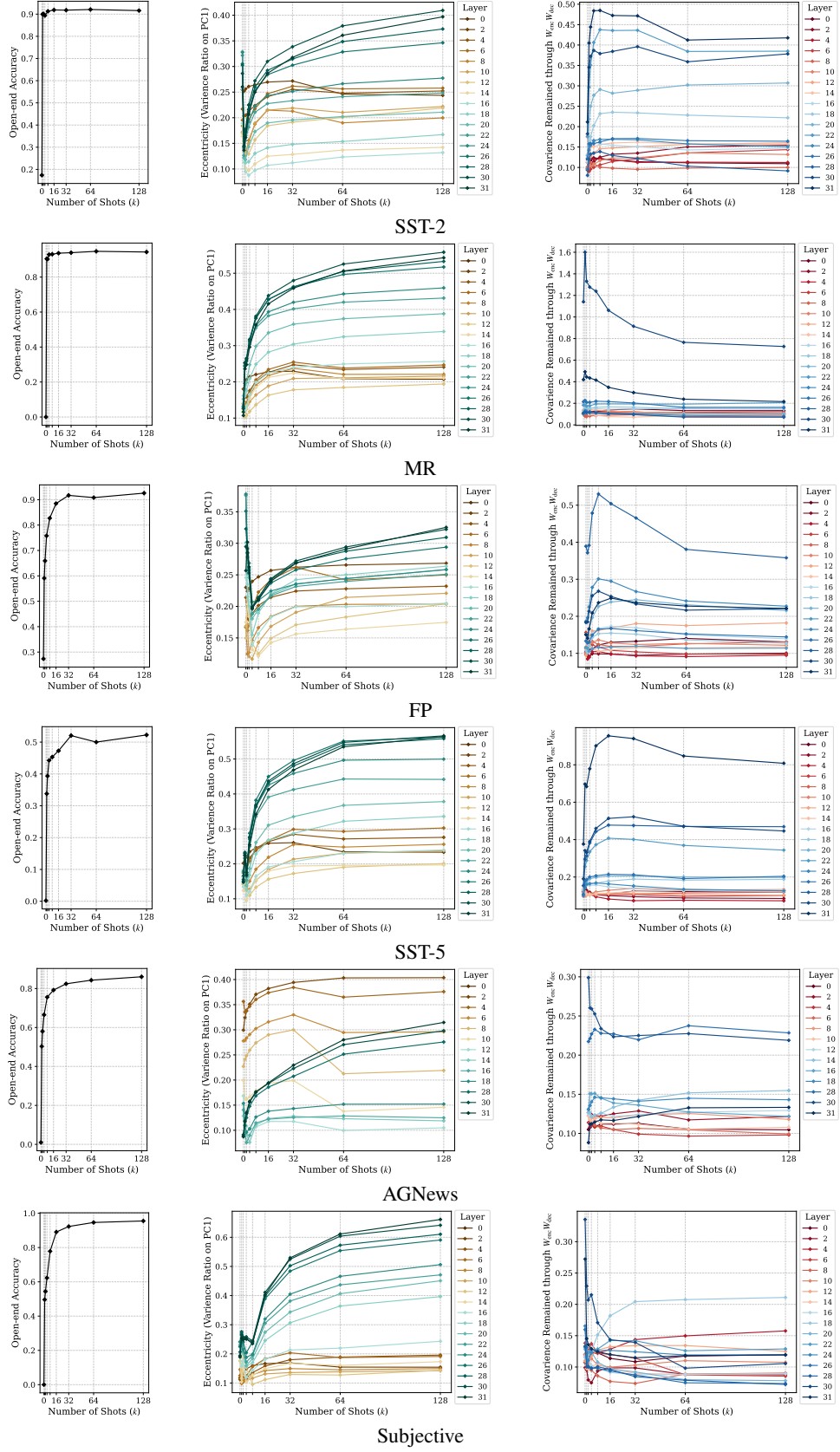

Figure 48: Augmentation results for Fig. 4 on Llama 3-8B.

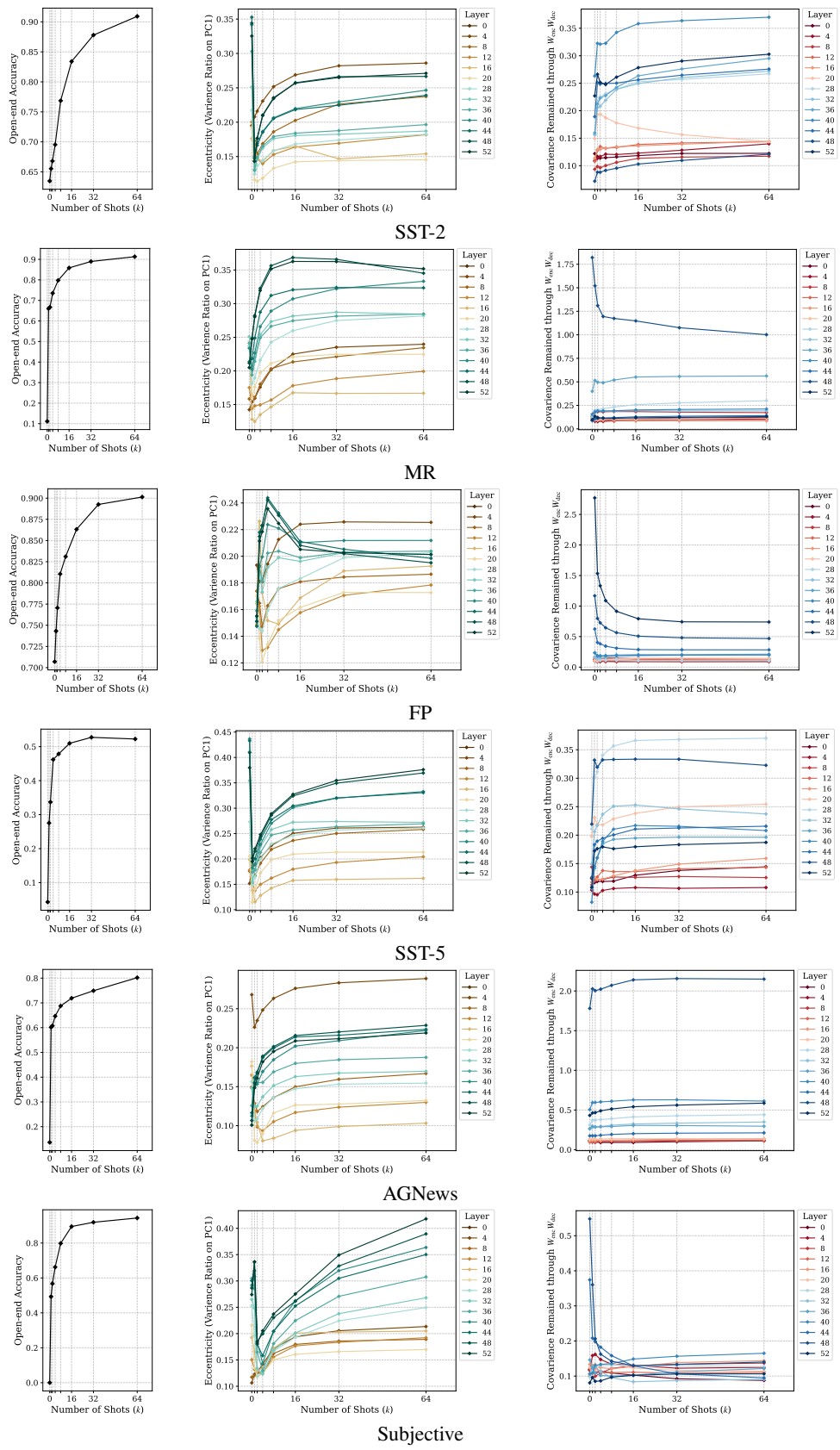

Figure 49: Augmentation results for Fig. 4 on Llama 3-13B.

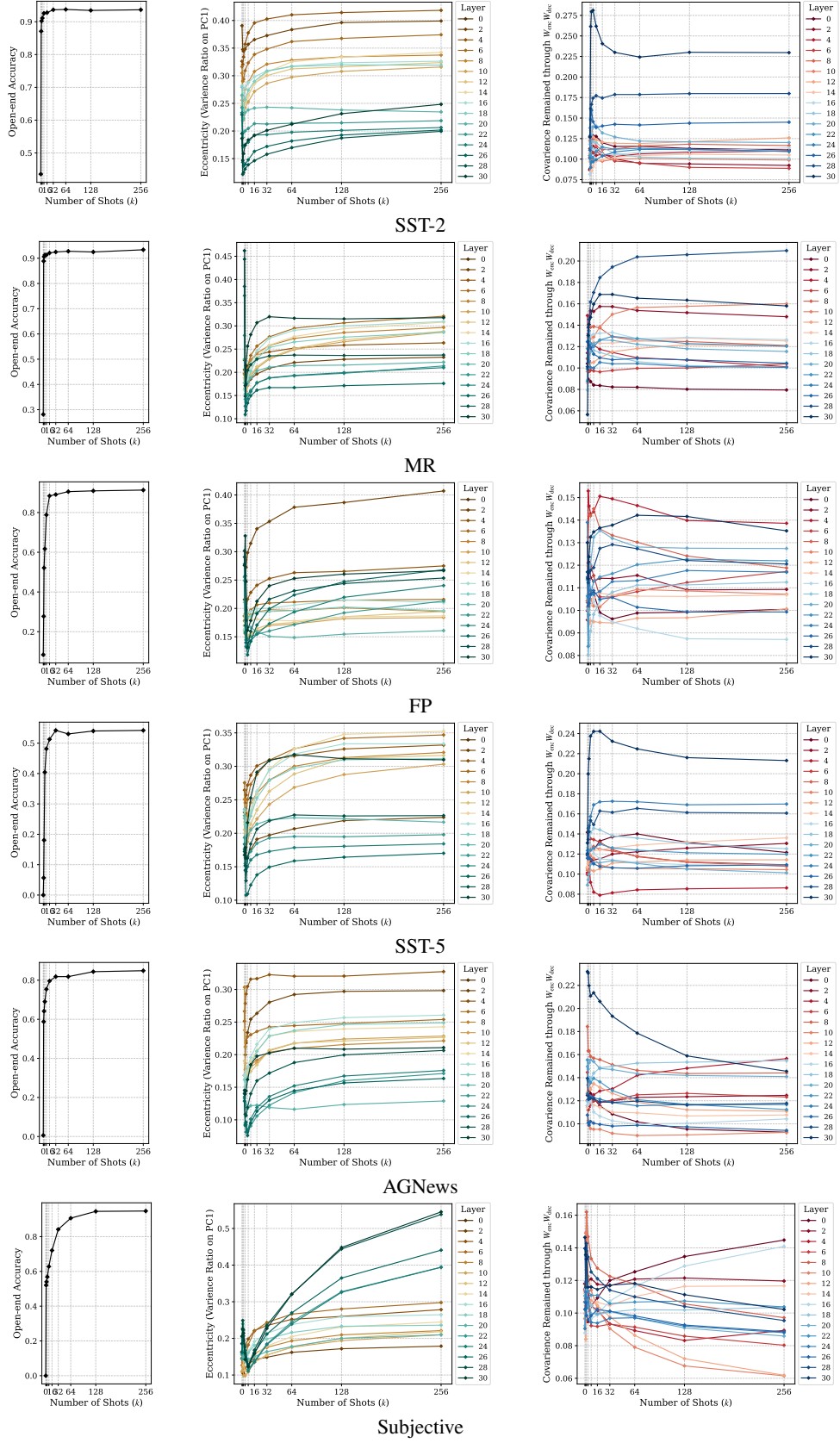

Figure 50: Augmentation results for Fig. 4 on Qwen 2.5-3B.

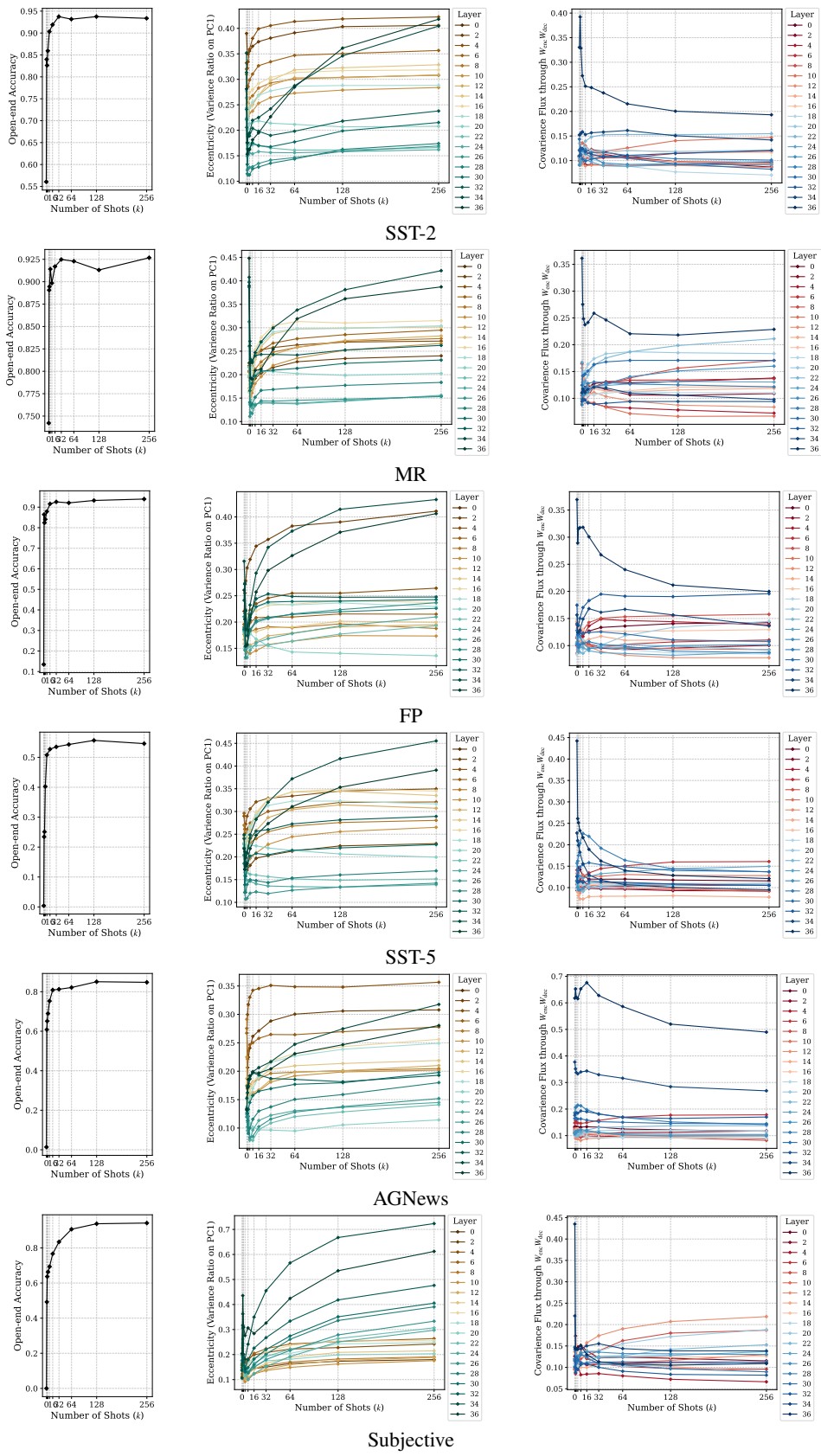

Figure 51: Augmentation results for Fig. 4 on Qwen 2.5-3B Instruct.

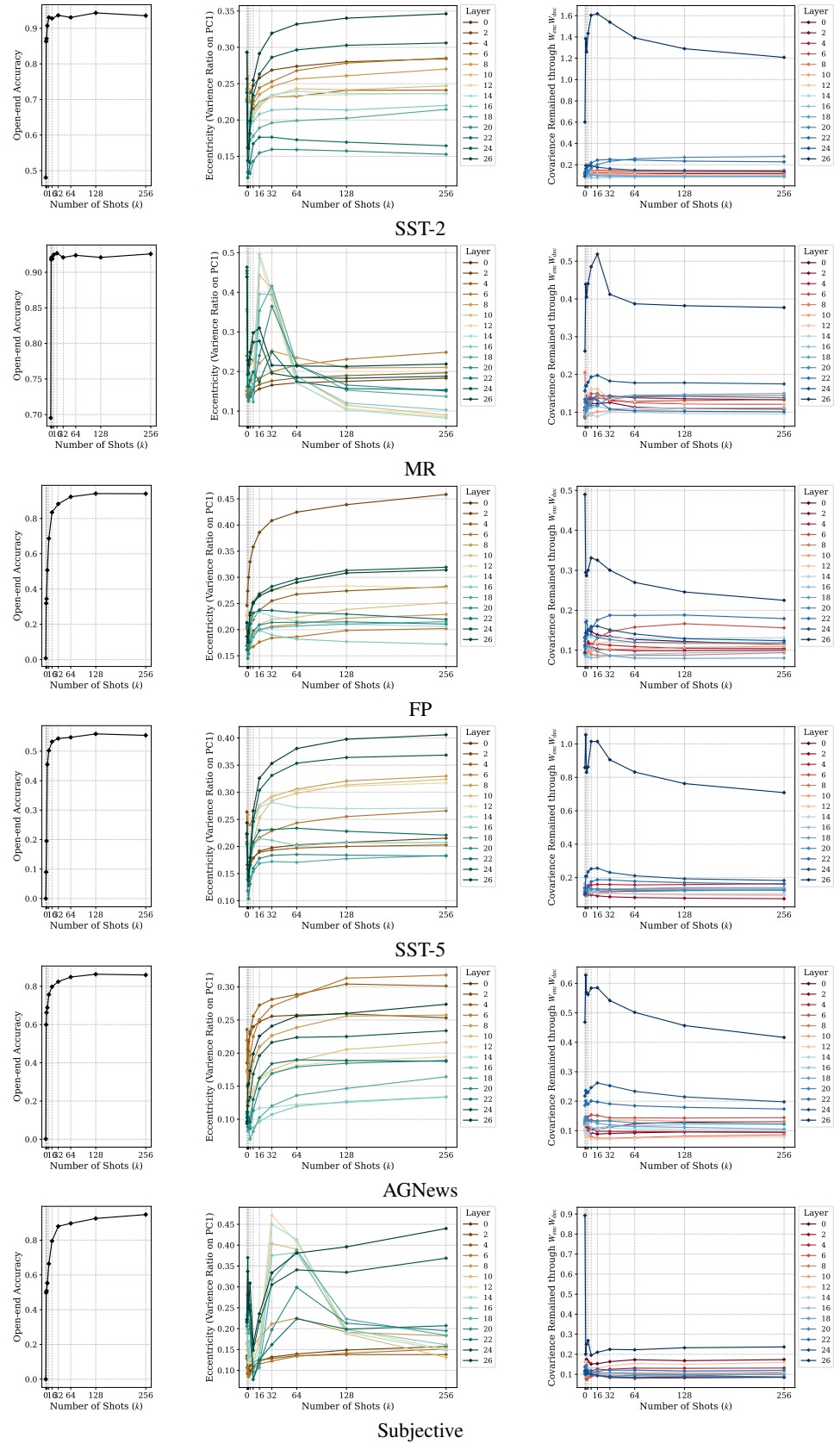

Figure 52: Augmentation results for Fig. 4 on Qwen 2.5-7B.

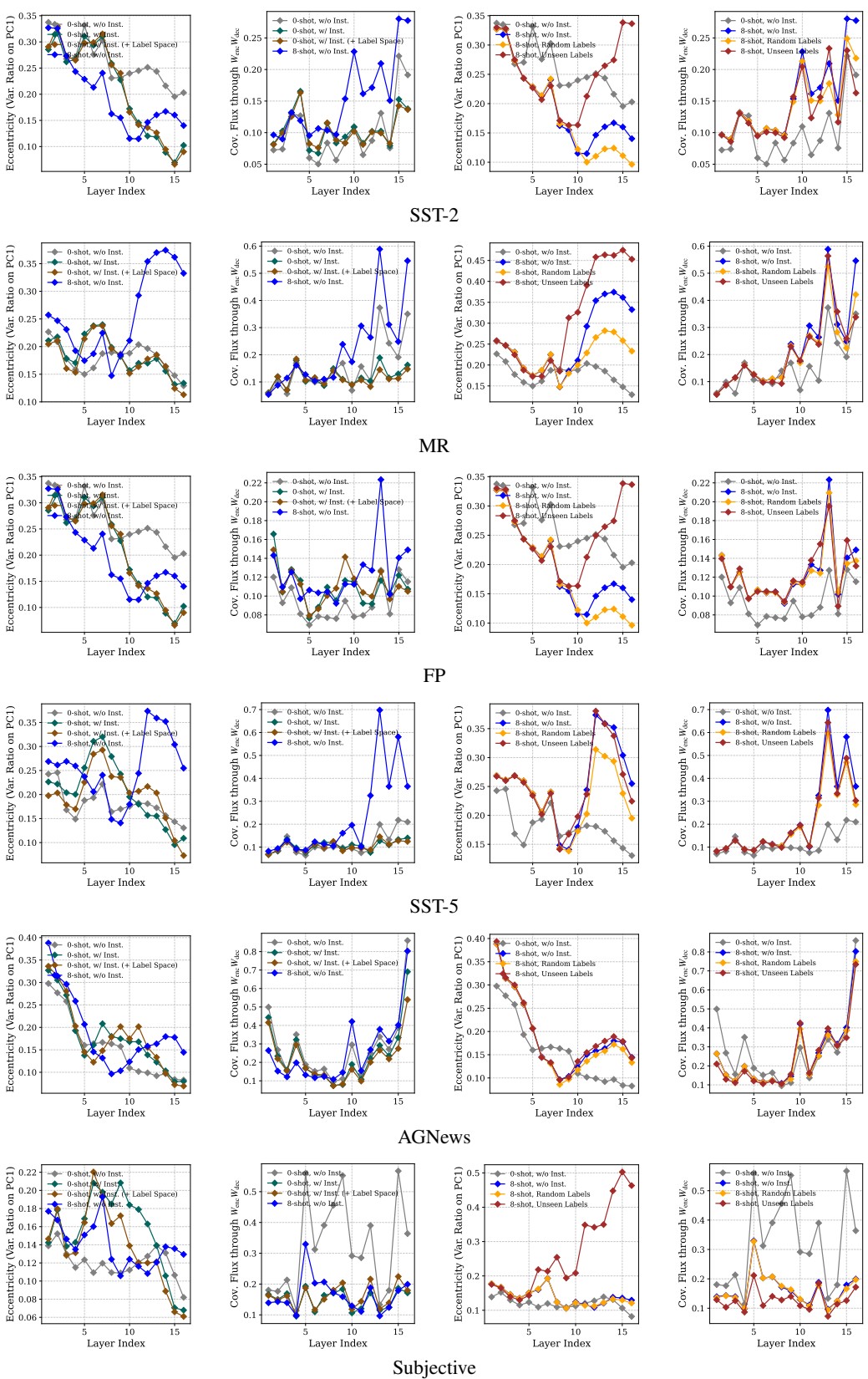

Figure 53: (Left 2) augmentation results for Fig. 5, (right 2) for Fig. 6 on Llama 3.2-1B.

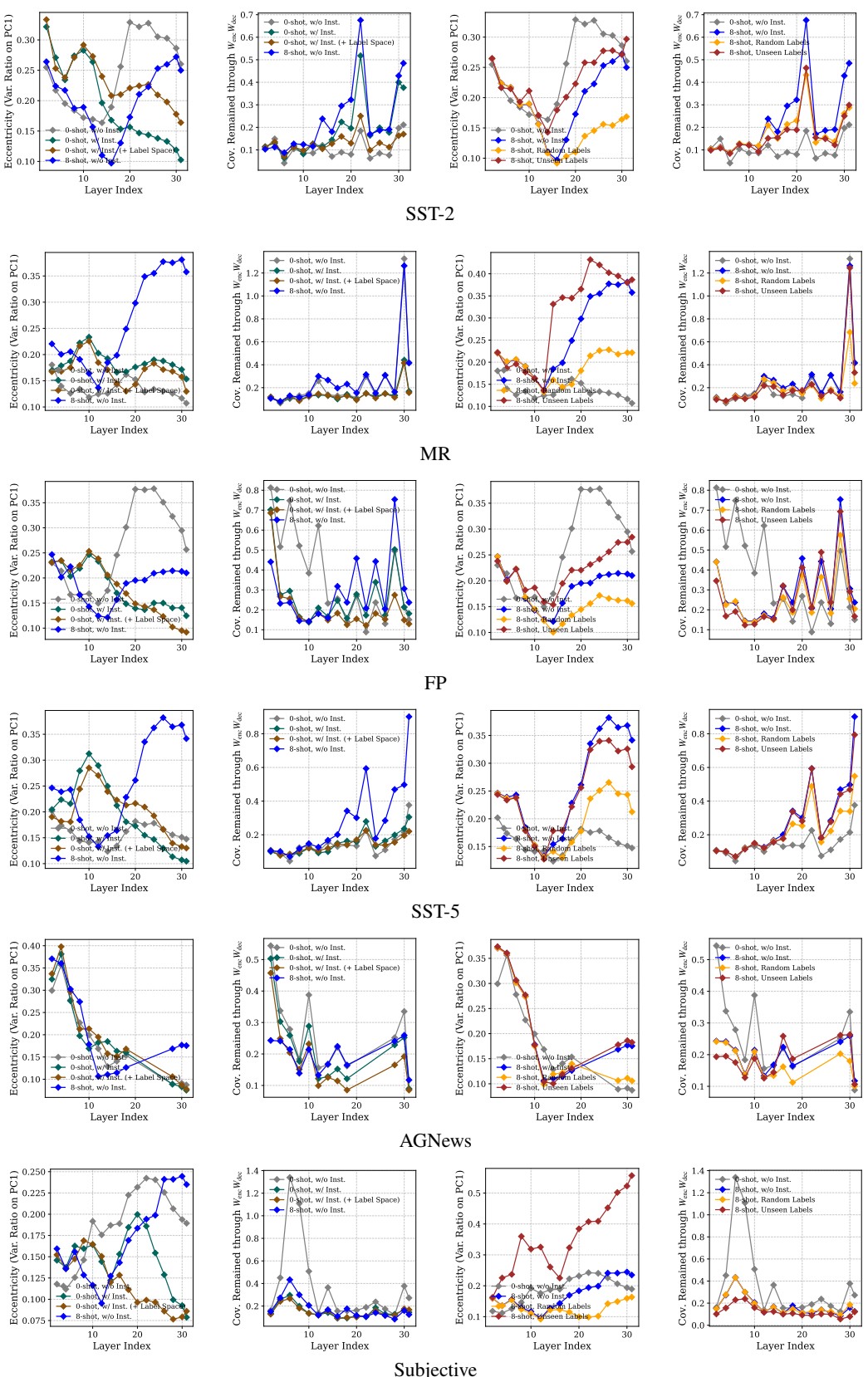

Figure 54: (Left 2) augmentation results for Fig. 5, (right 2) for Fig. 6 on Llama 3-8B.

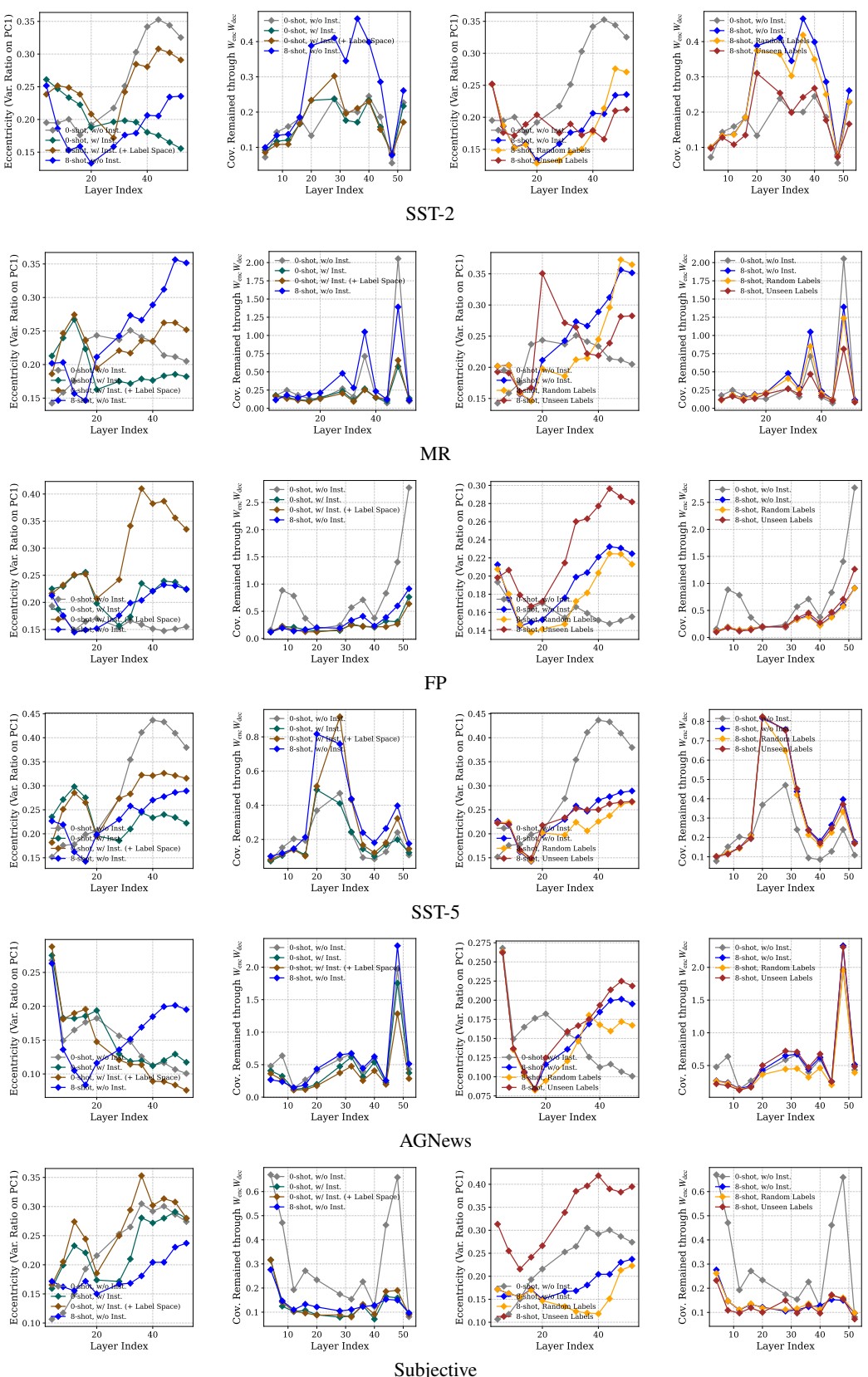

Figure 55: (Left 2) augmentation results for Fig. 5, (right 2) for Fig. 6 on Llama 3-13B.

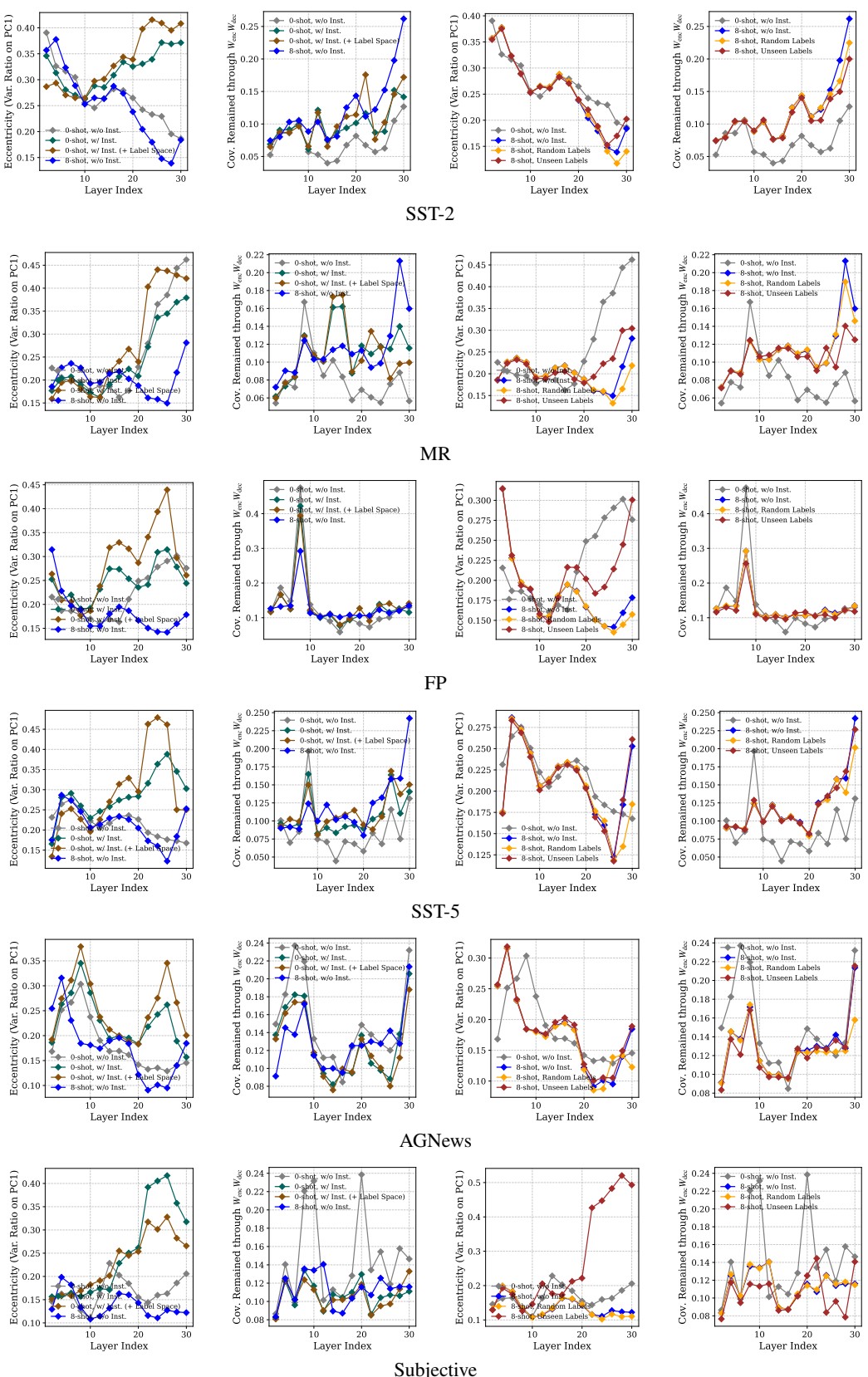

Figure 56: (Left 2) augmentation results for Fig. 5, (right 2) for Fig. 6 on Qwen 2.5-3B.

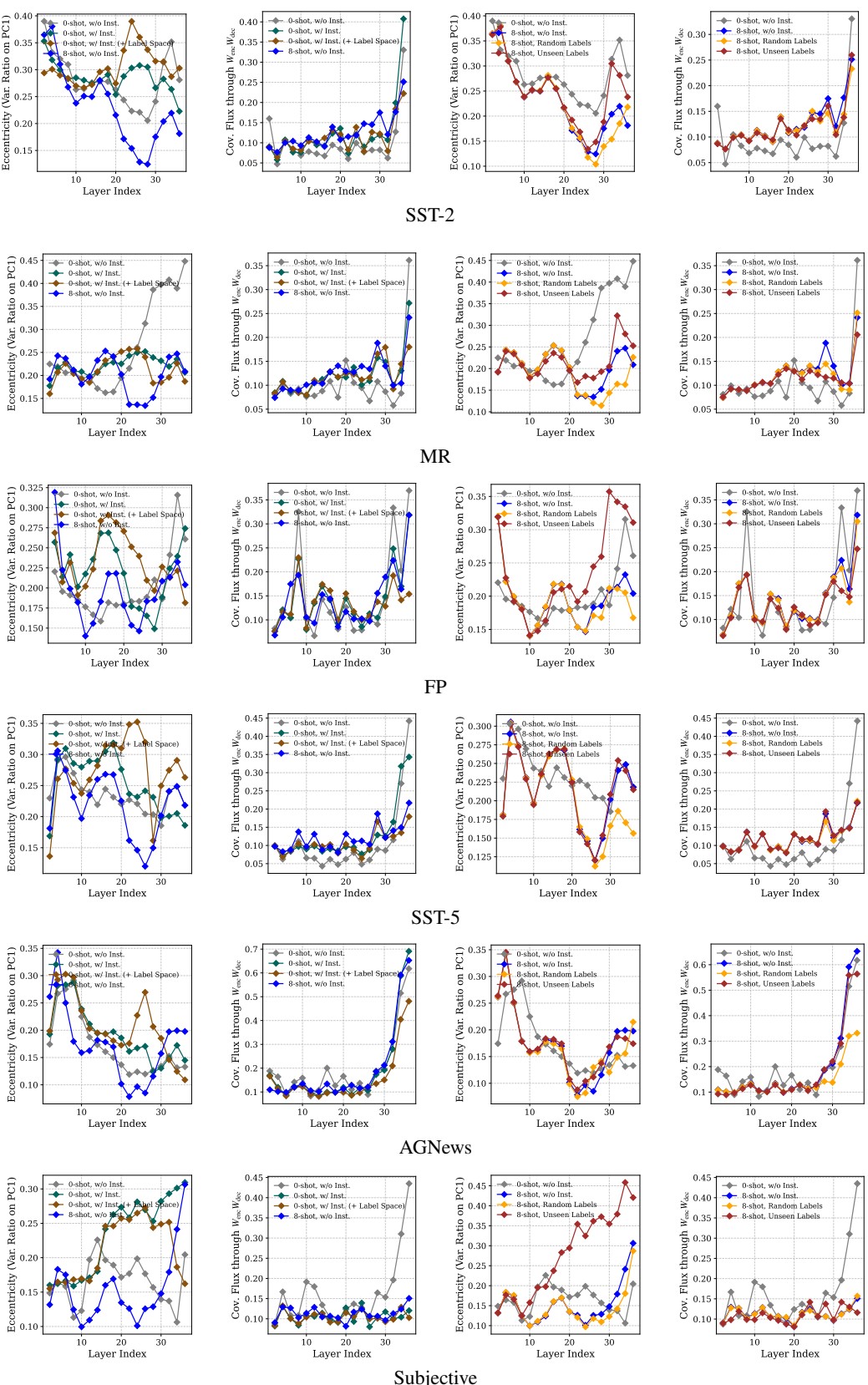

Figure 57: (Left 2) augmentation results for Fig. 5, (right 2) for Fig. 6 on Qwen 2.5-3B Instruct.

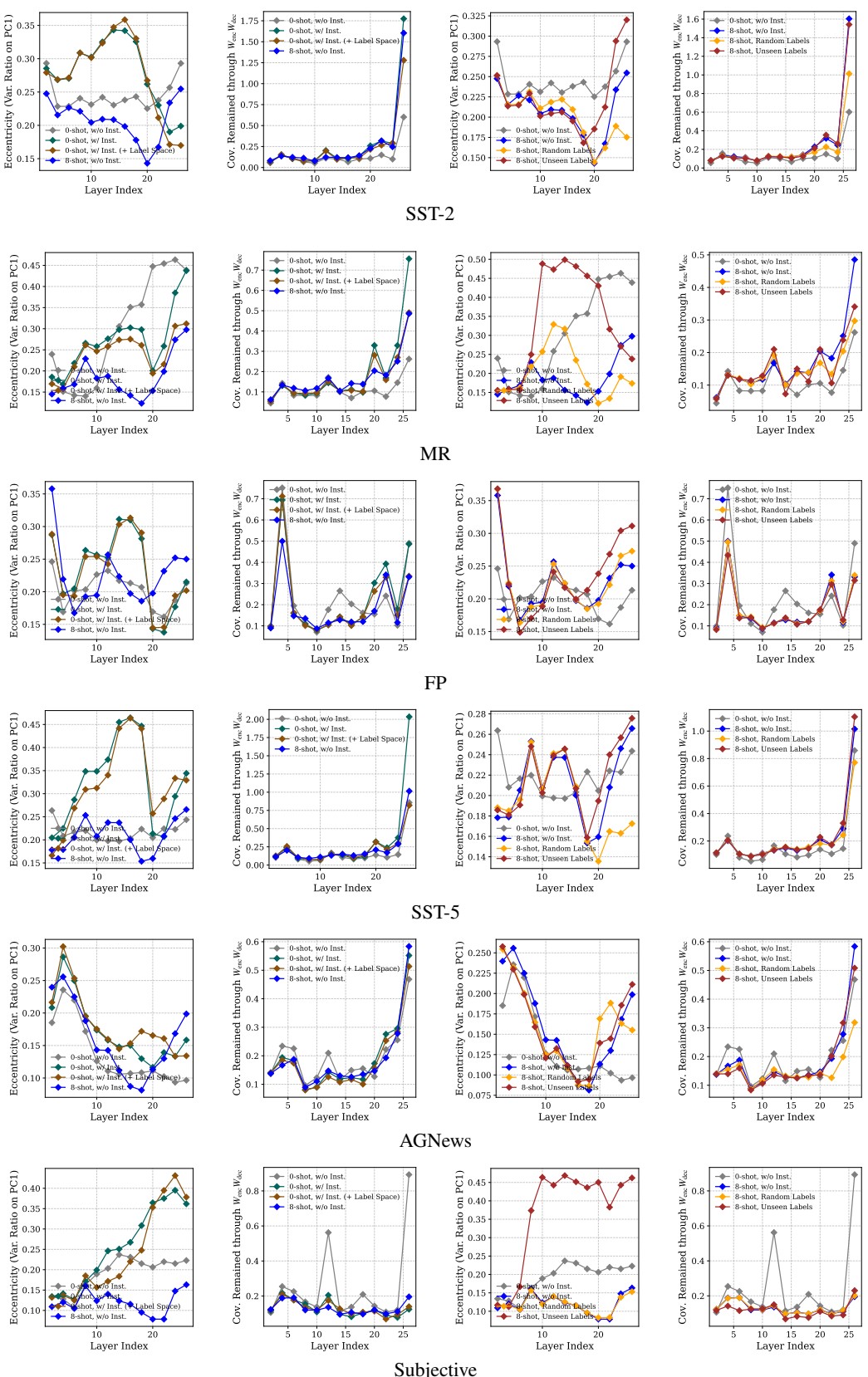

Figure 58: (Left 2) augmentation results for Fig. 5, (right 2) for Fig. 6 on Qwen 2.5-7B.

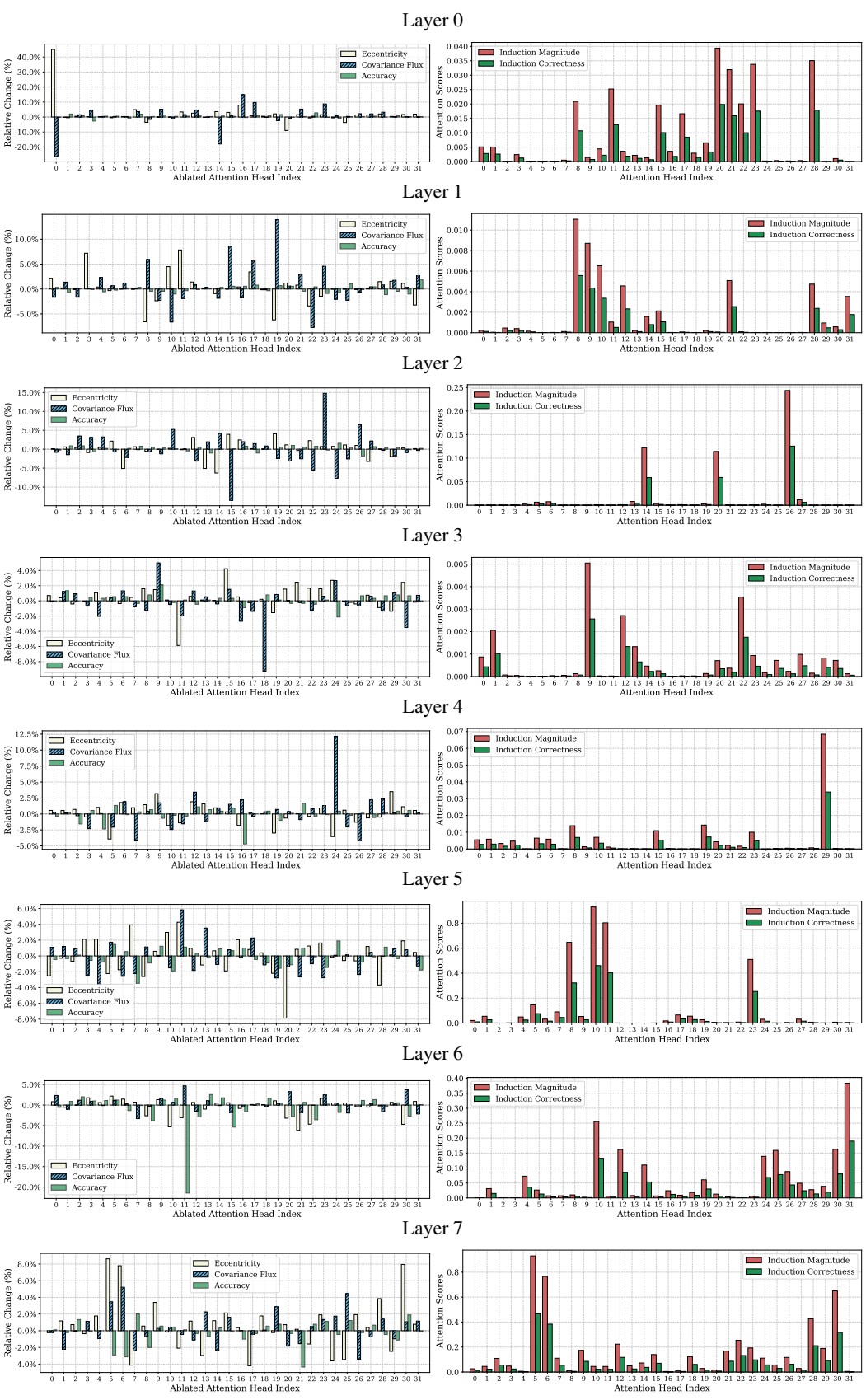

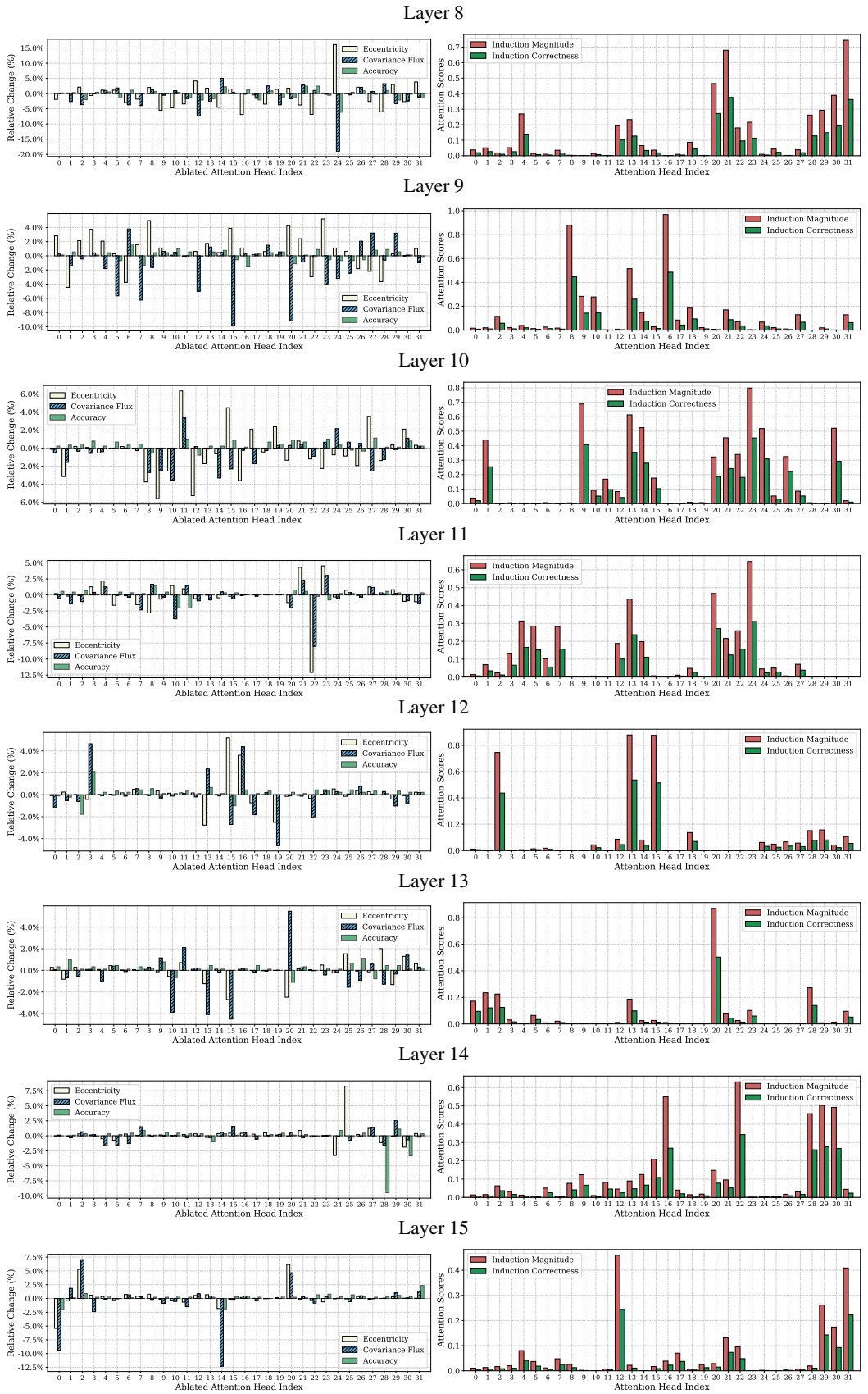

Figure 59: (Left) augmentation results for Fig. 7, (right) induction score of each attention head on Llama 3.2-1B, SST-2.

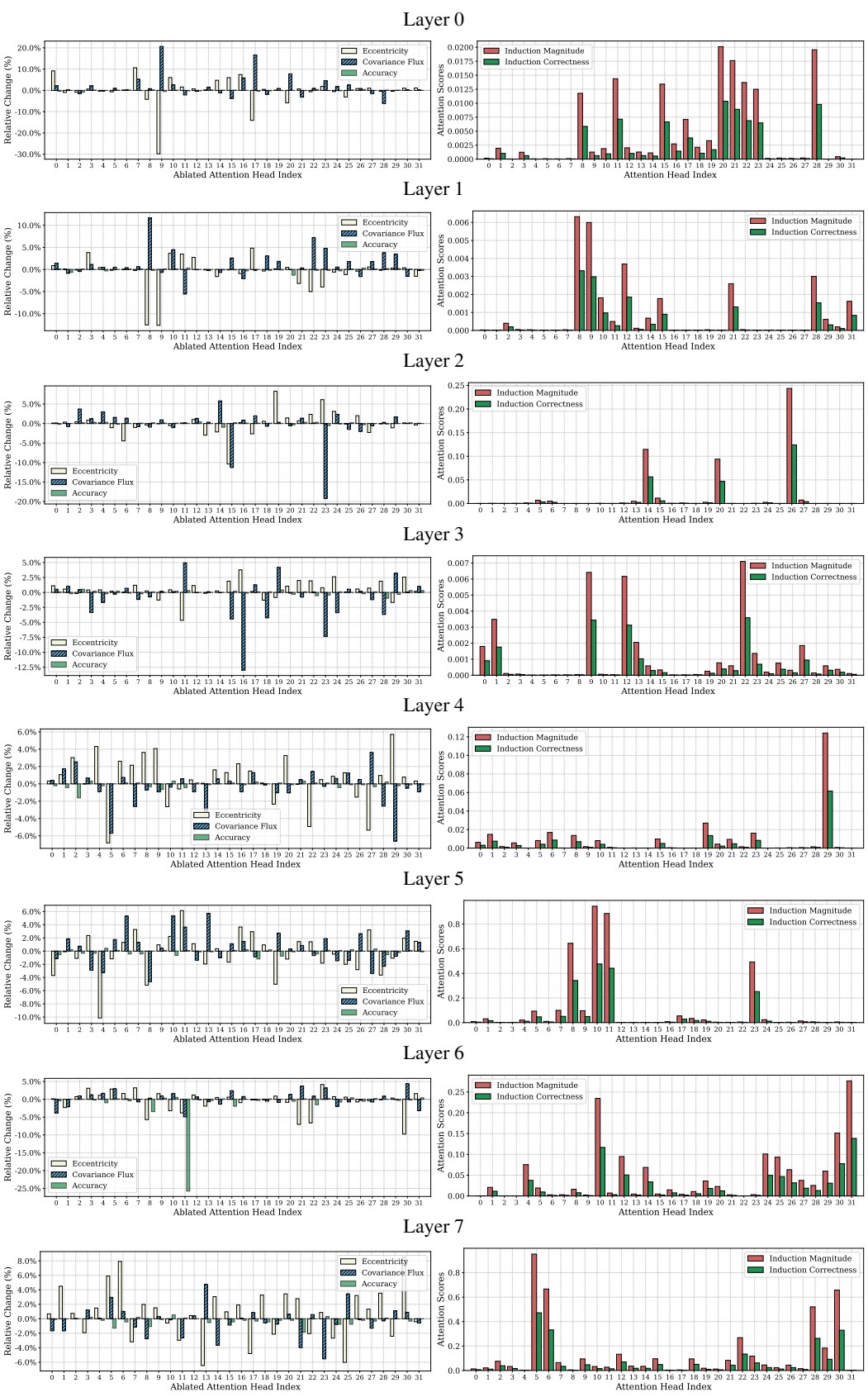

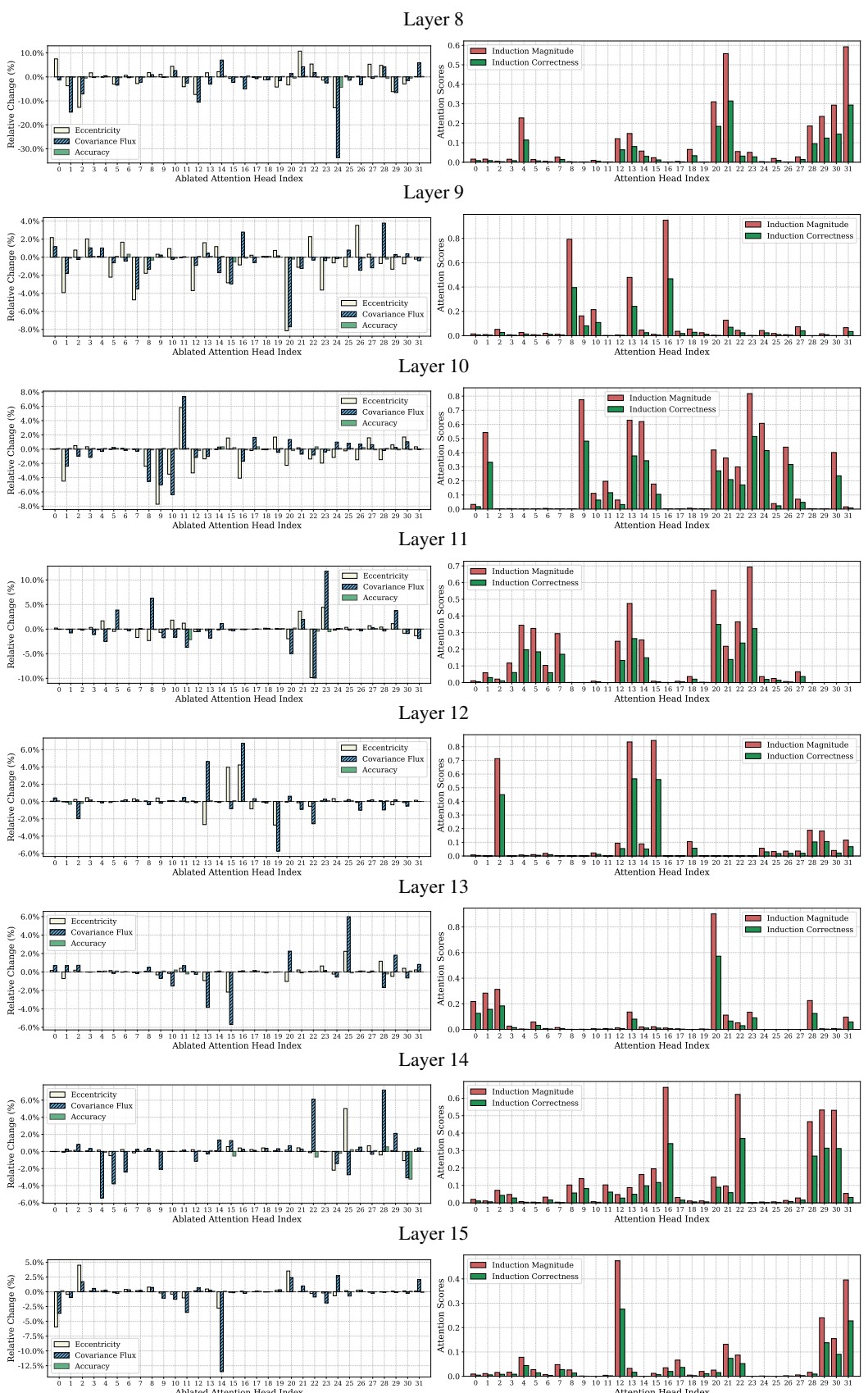

Figure 60: (Left) augmentation results for Fig. 7, (right) induction score of each attention head on Llama 3.2-1B, MR.

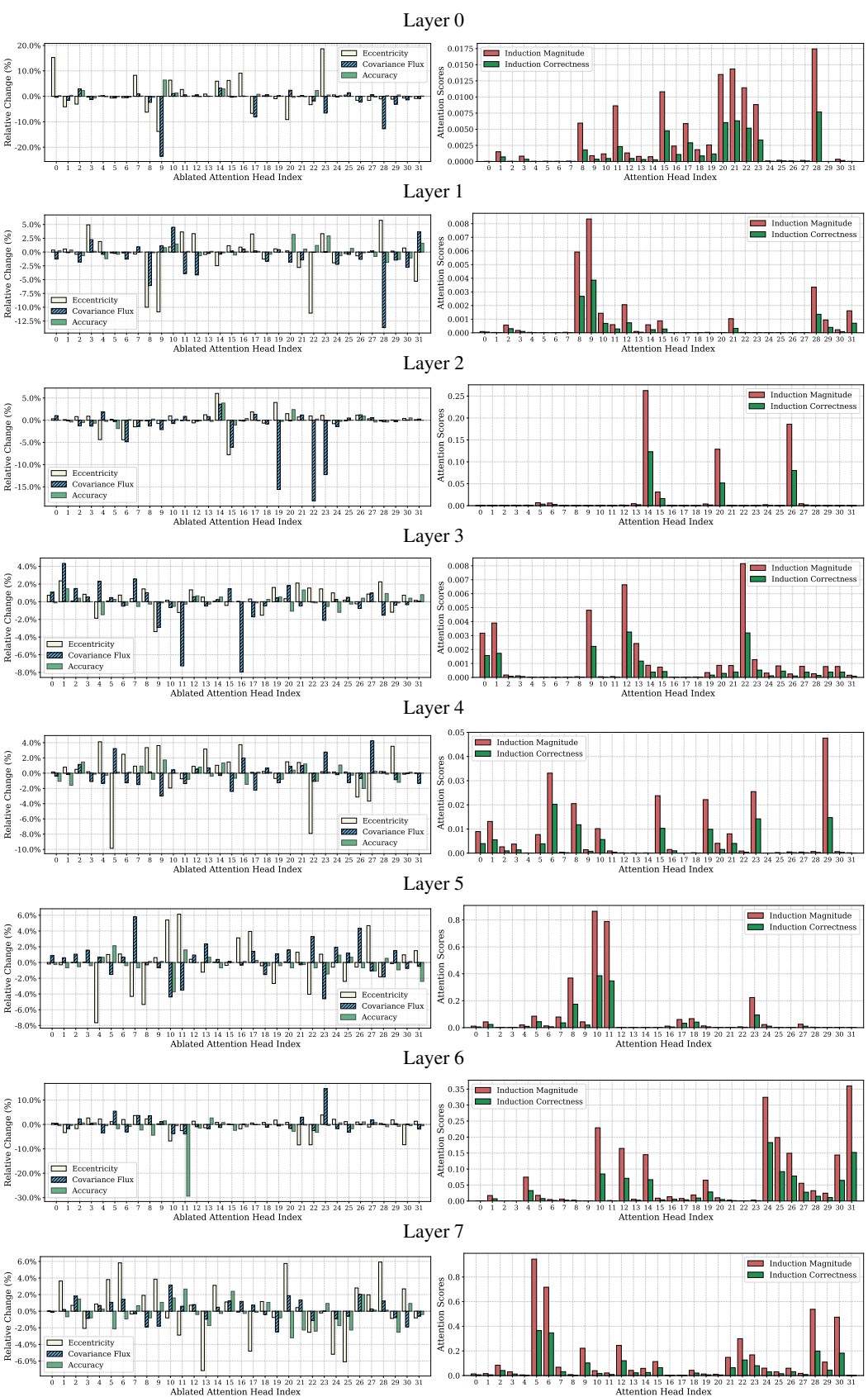

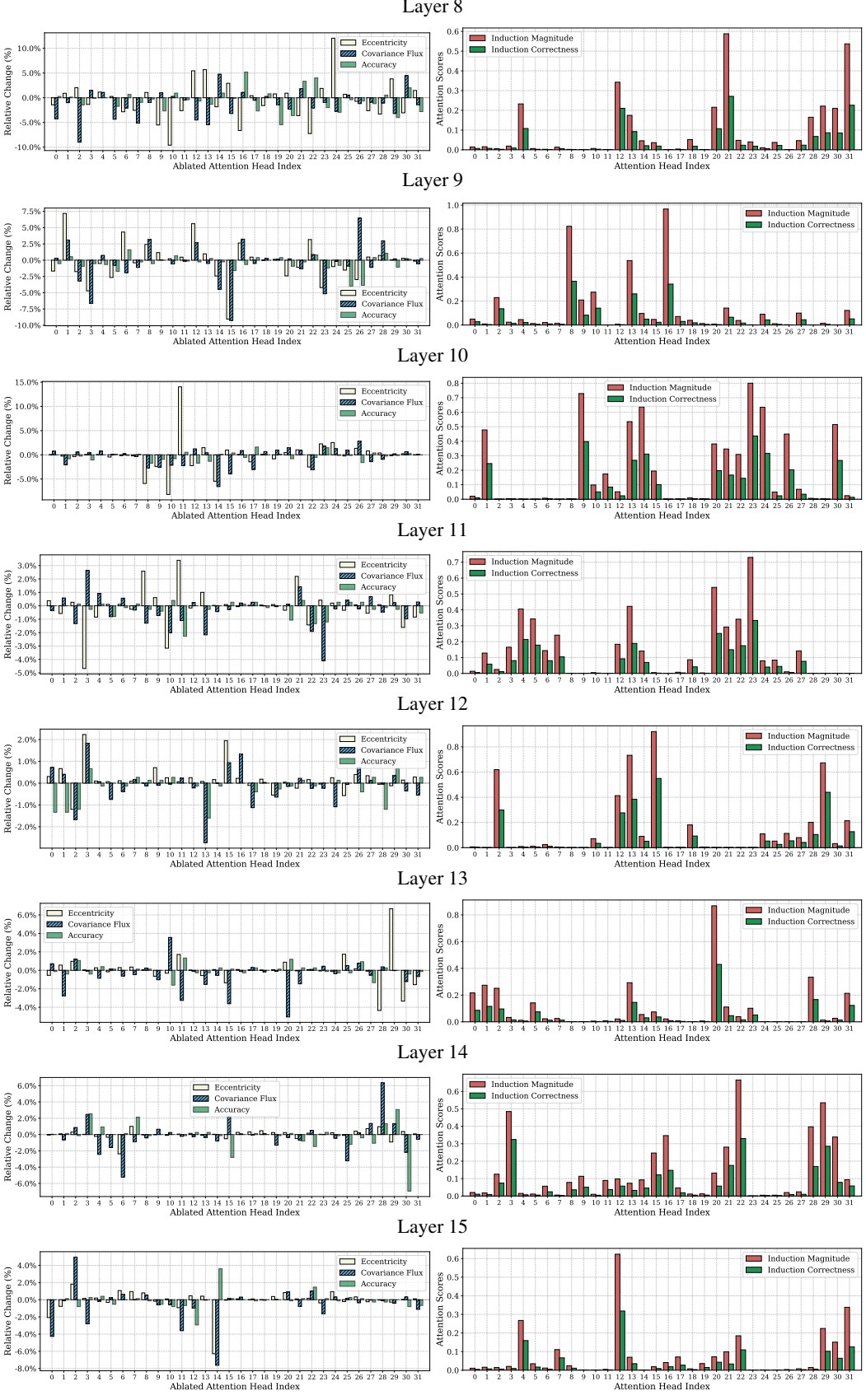

Figure 61: (Left) augmentation results for Fig. 7, (right) induction score of each attention head on Llama 3.2-1B, FP.

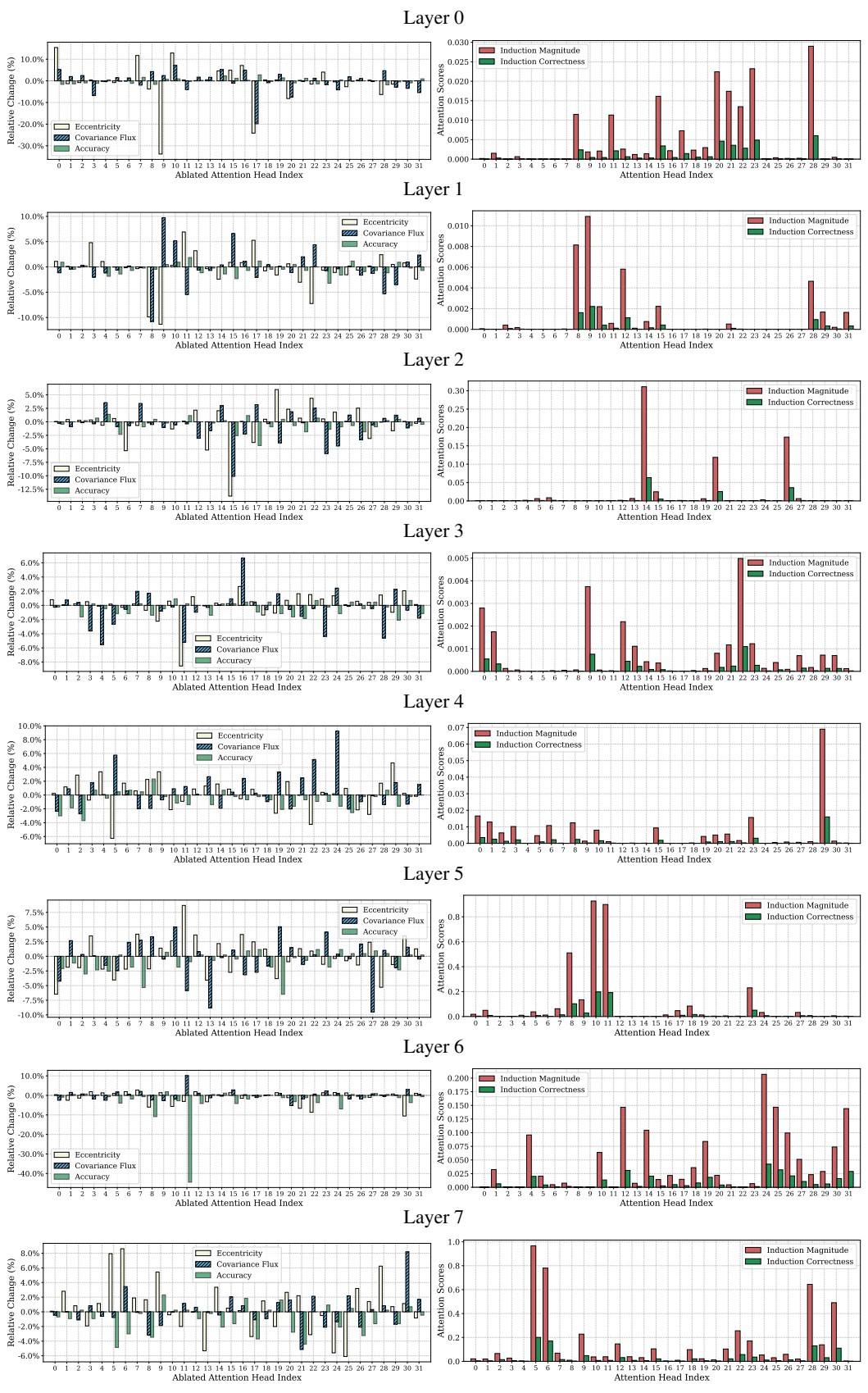

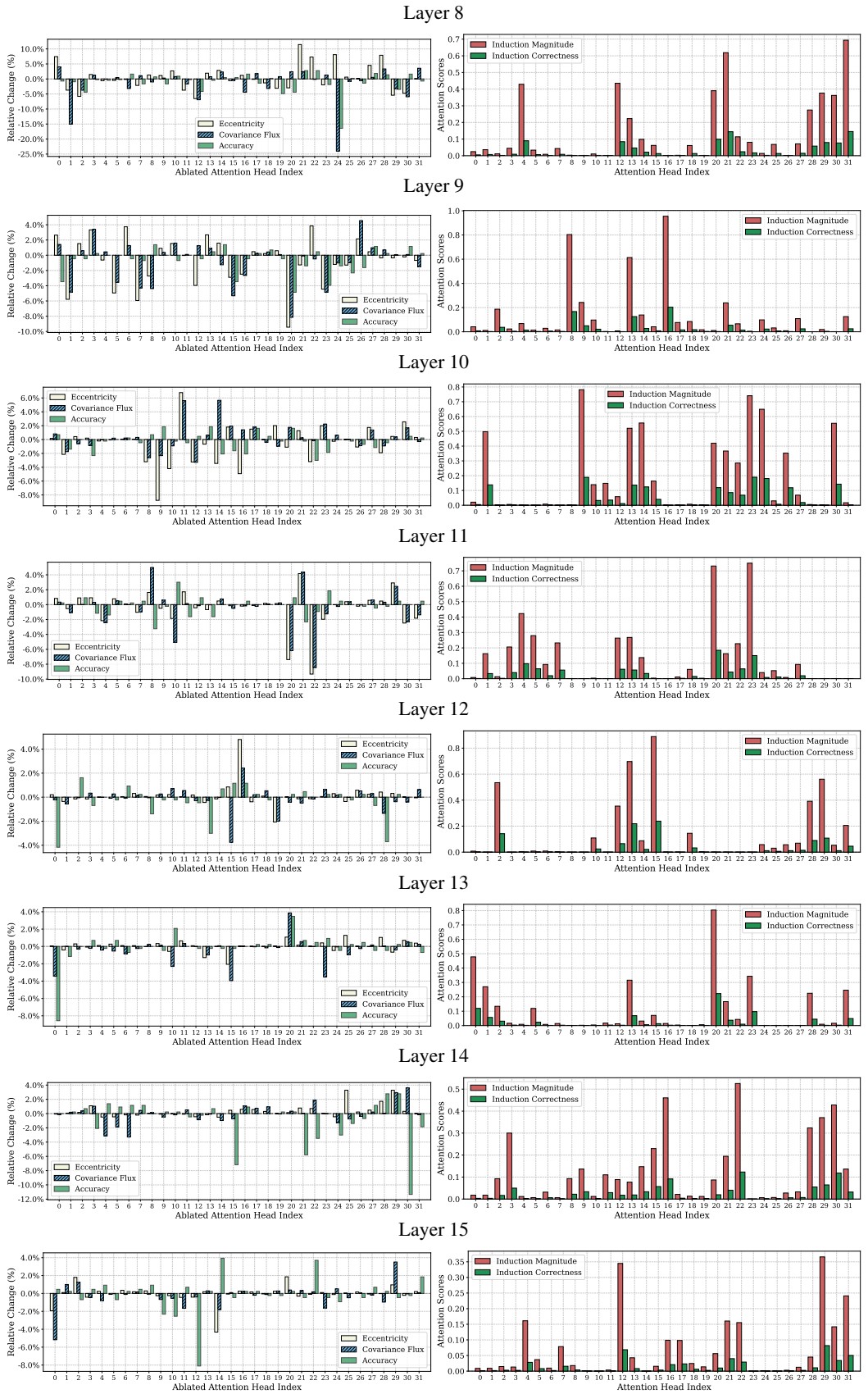

Figure 62: (Left) augmentation results for Fig. 7, (right) induction score of each attention head on Llama 3.2-1B, SST-5.

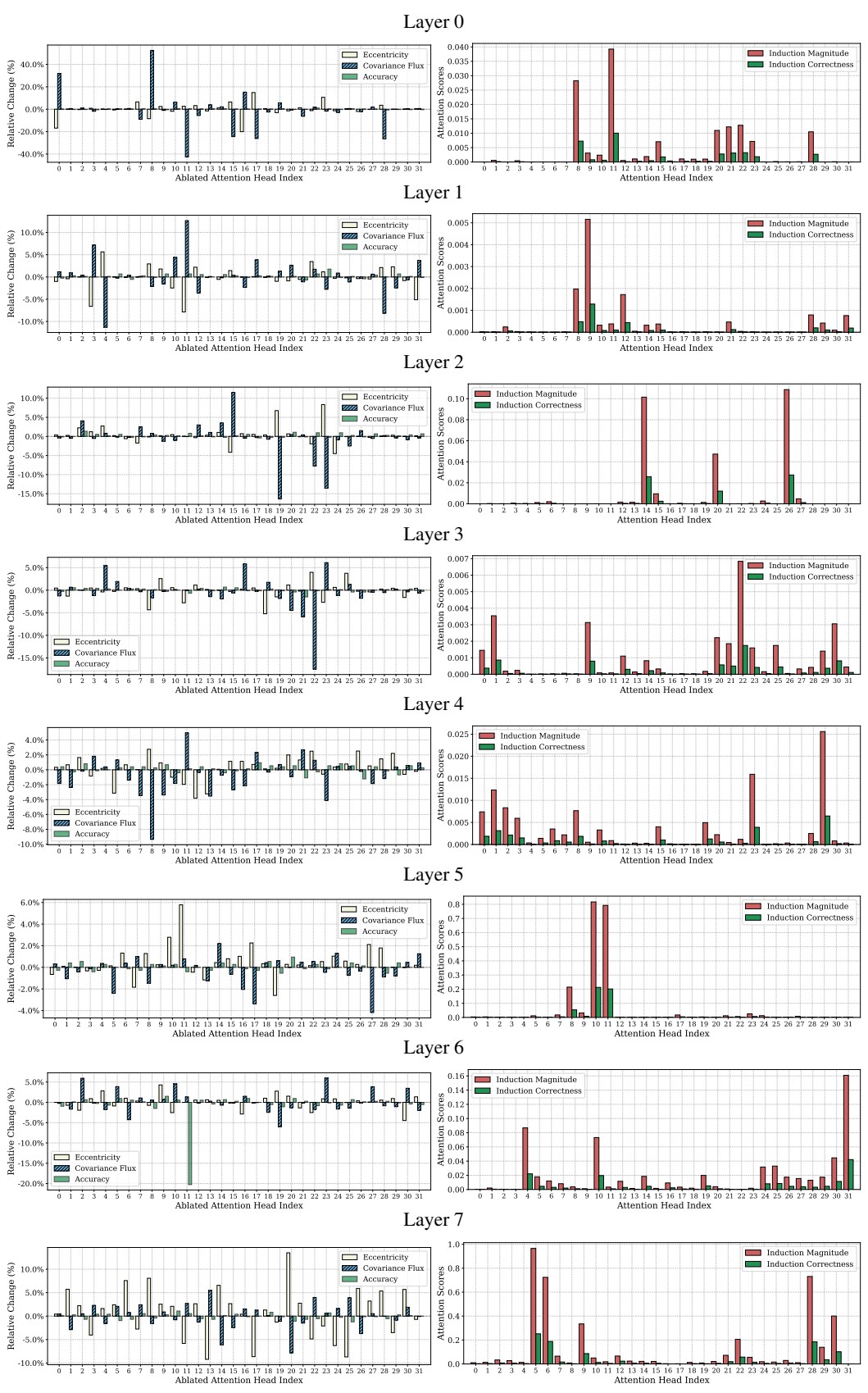

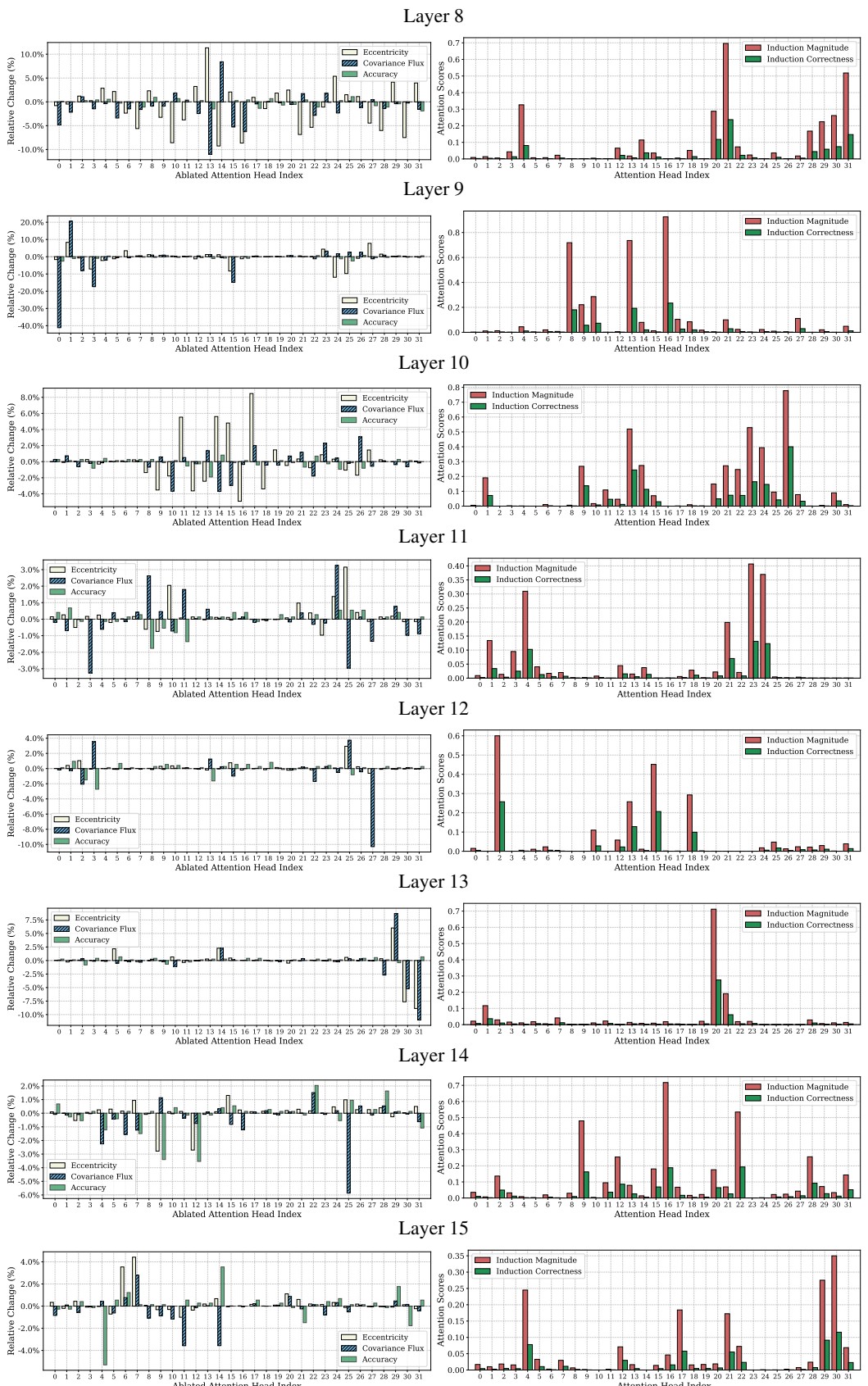

Figure 63: (Left) augmentation results for Fig. 7, (right) induction score of each attention head on Llama 3.2-1B, AGNews.

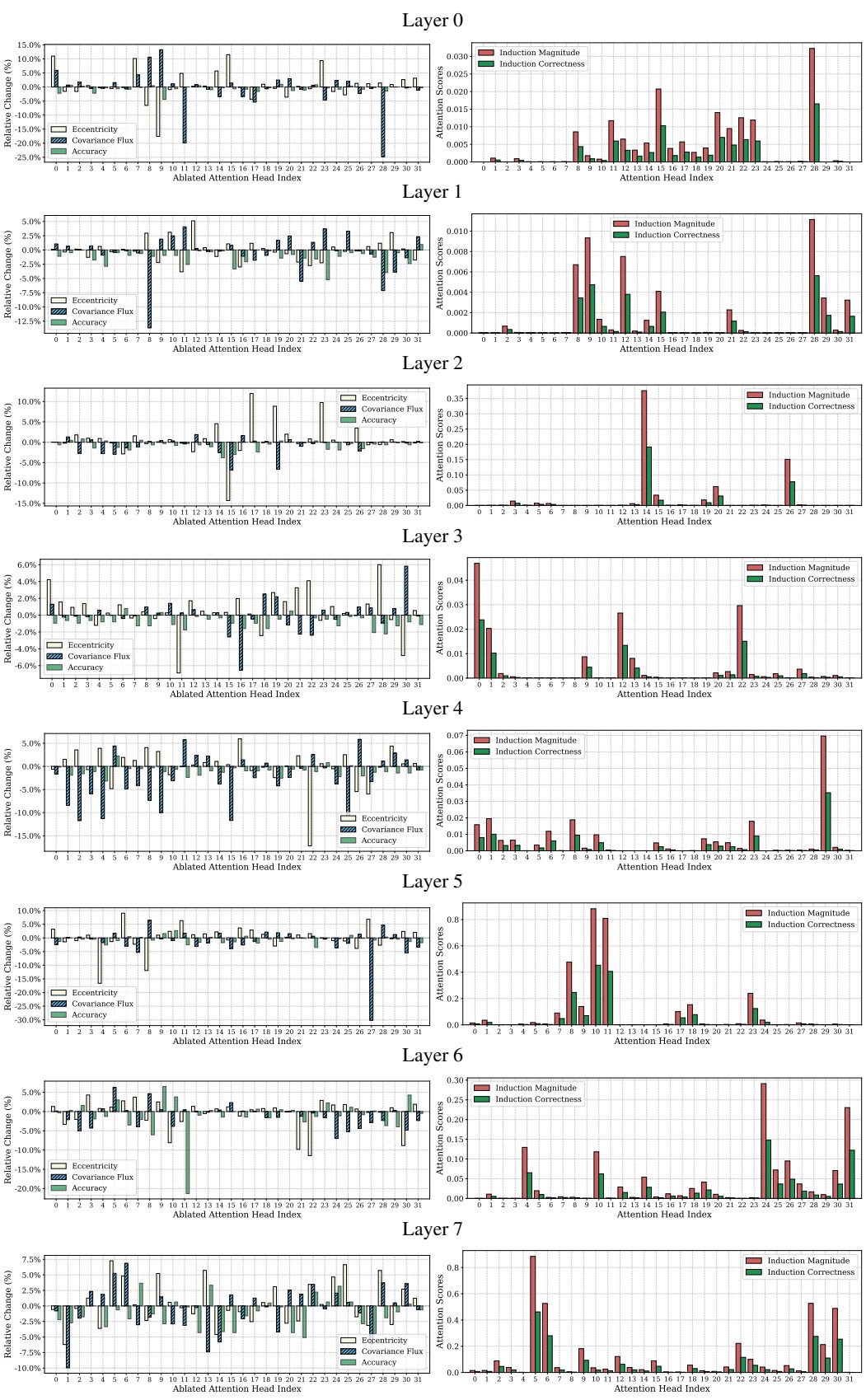

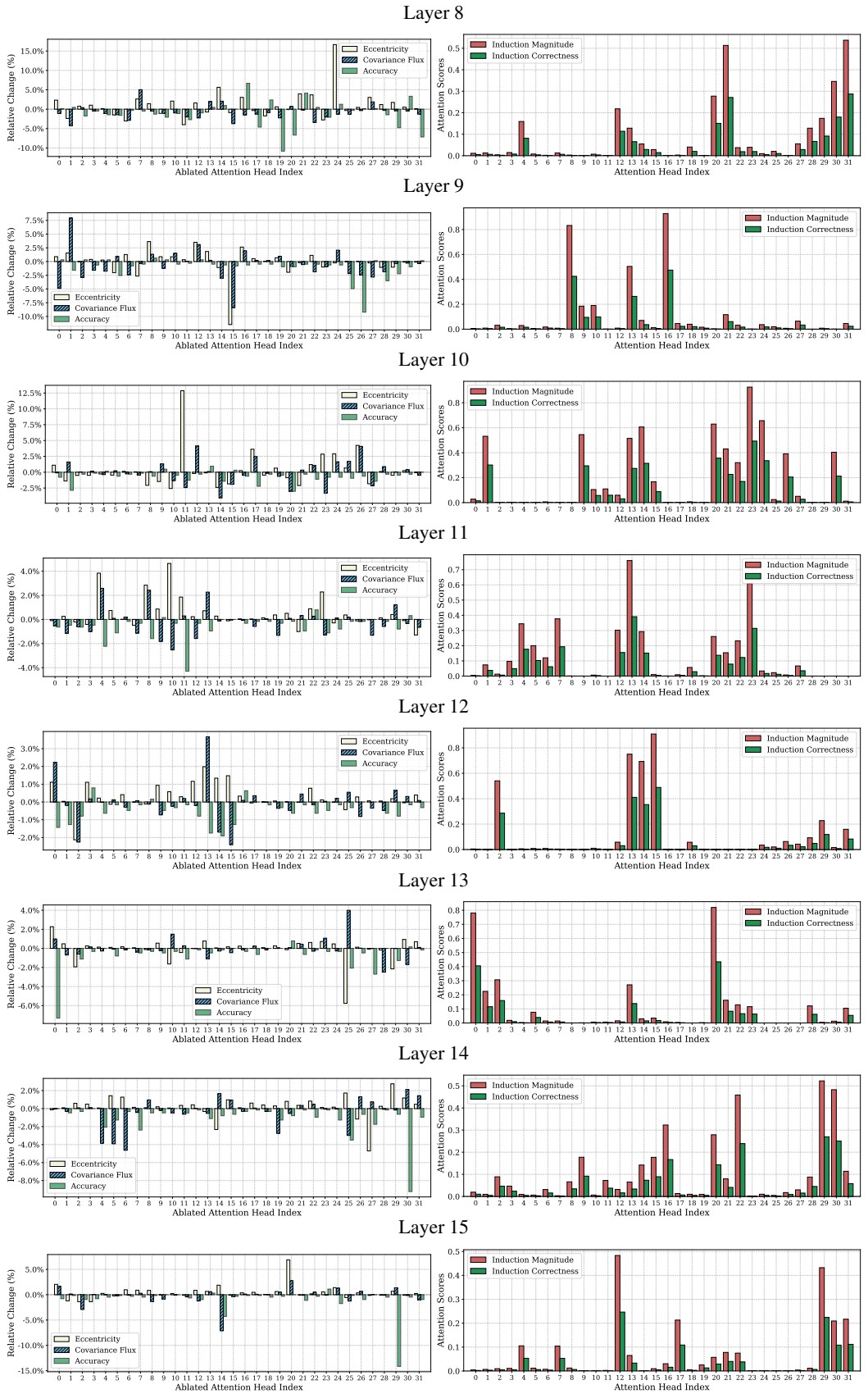

Figure 64: (Left) augmentation results for Fig. 7, (right) induction score of each attention head on Llama 3.2-1B, Subjective.

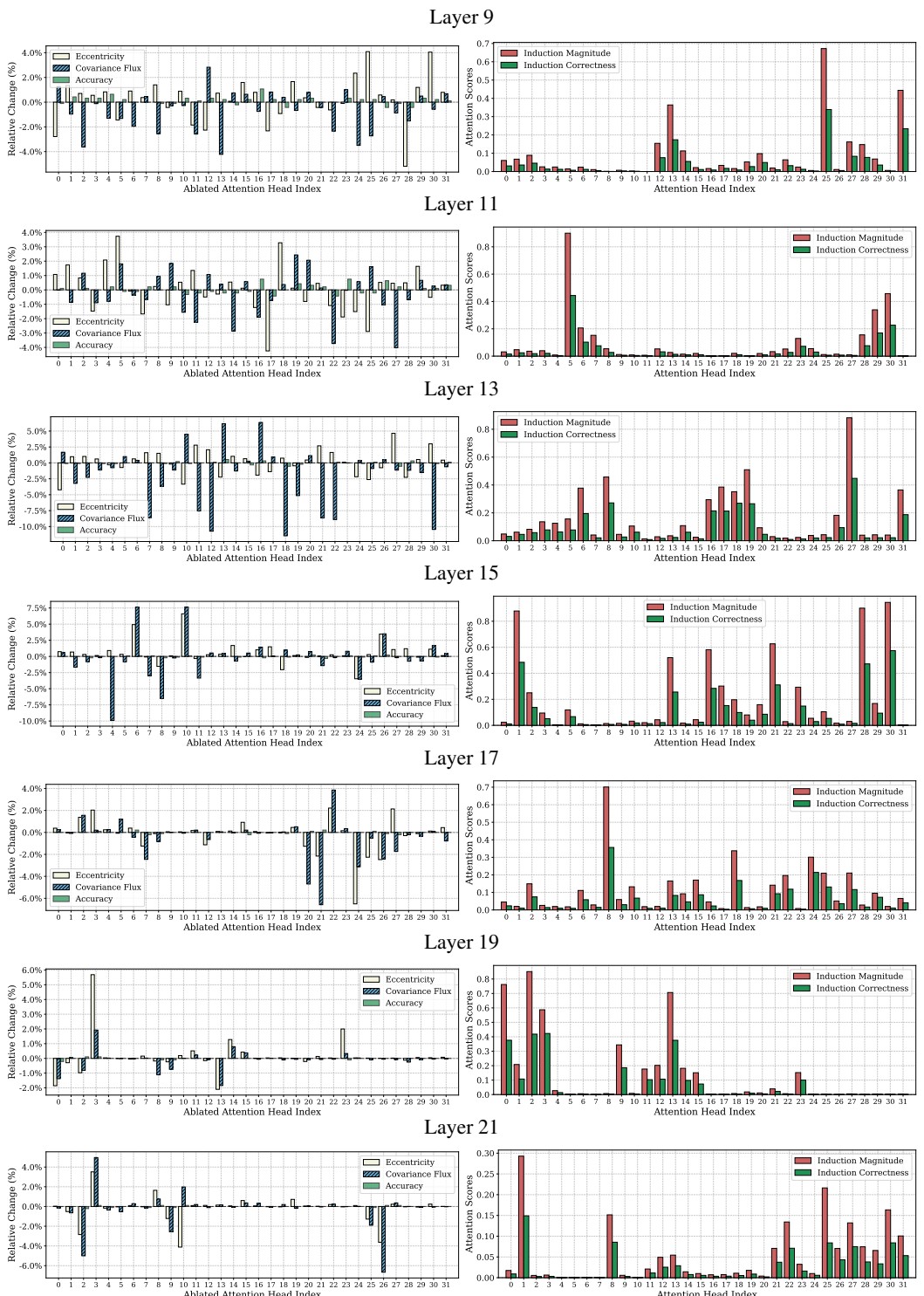

Figure 65: (Left) augmentation results for Fig. 7, (right) induction score of each attention head on Llama 3-8B, SST-2.

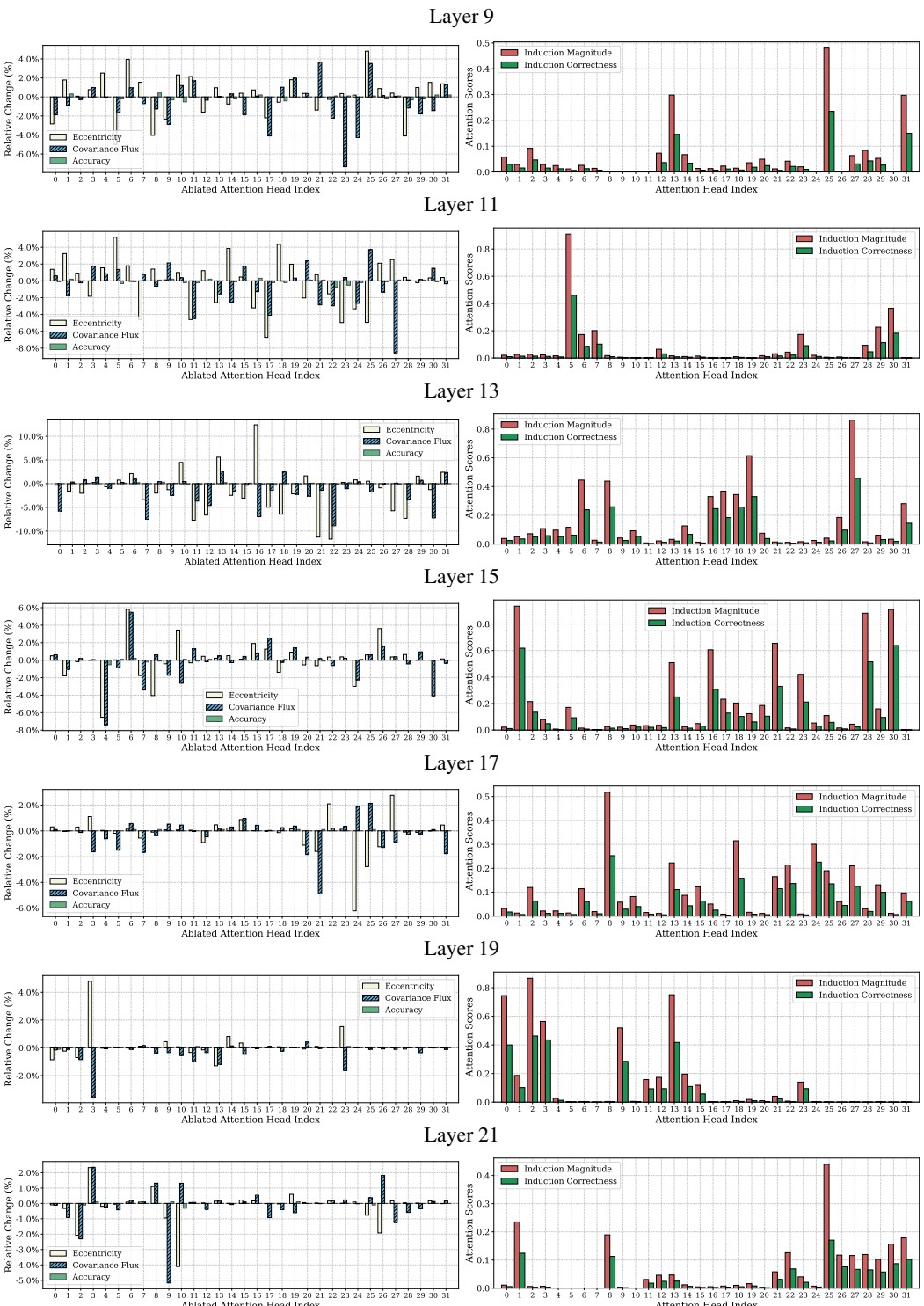

Figure 66: (Left) augmentation results for Fig. 7, (right) induction score of each attention head on Llama 3-8B, MR.

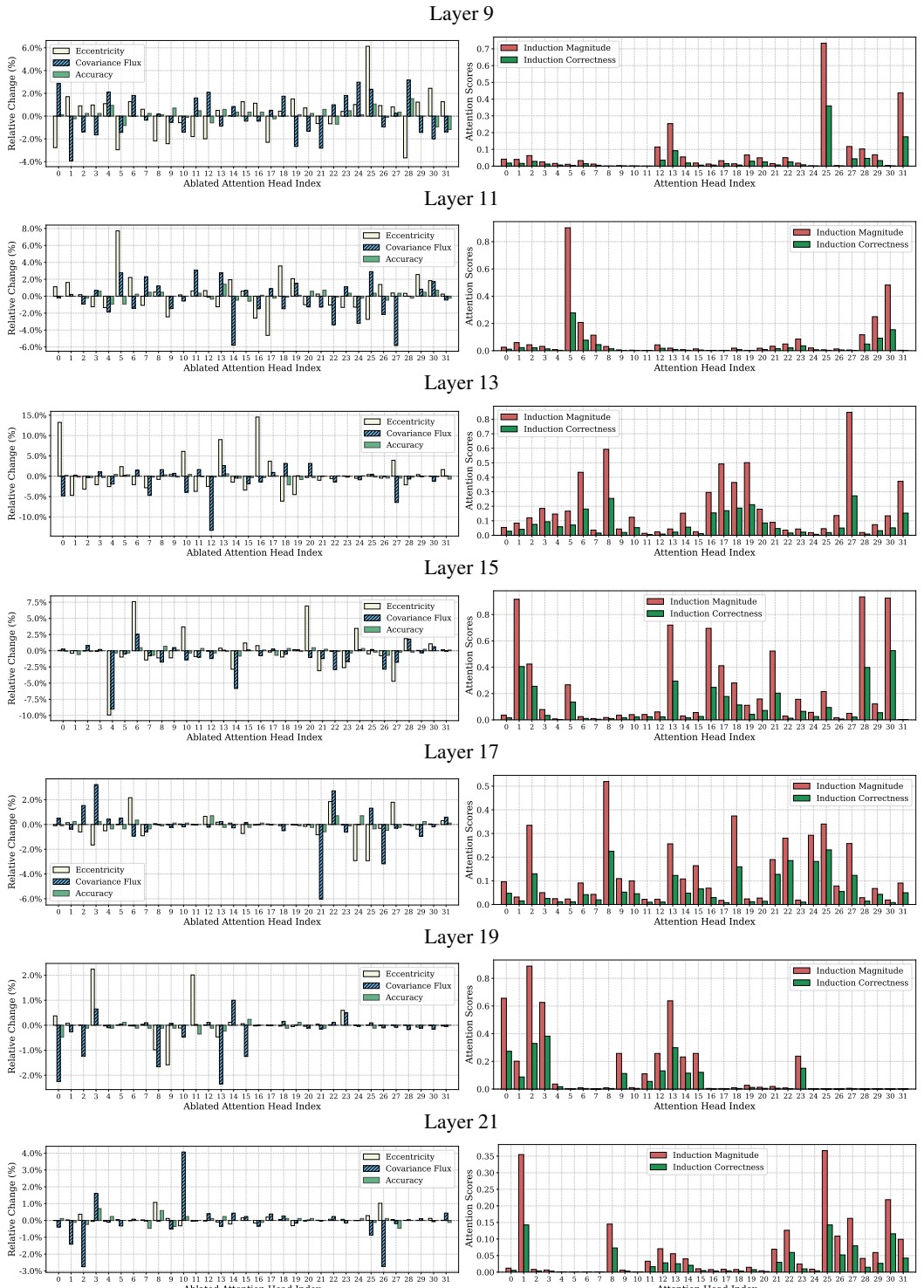

Figure 67: (Left) augmentation results for Fig. 7, (right) induction score of each attention head on Llama 3-8B, FP.

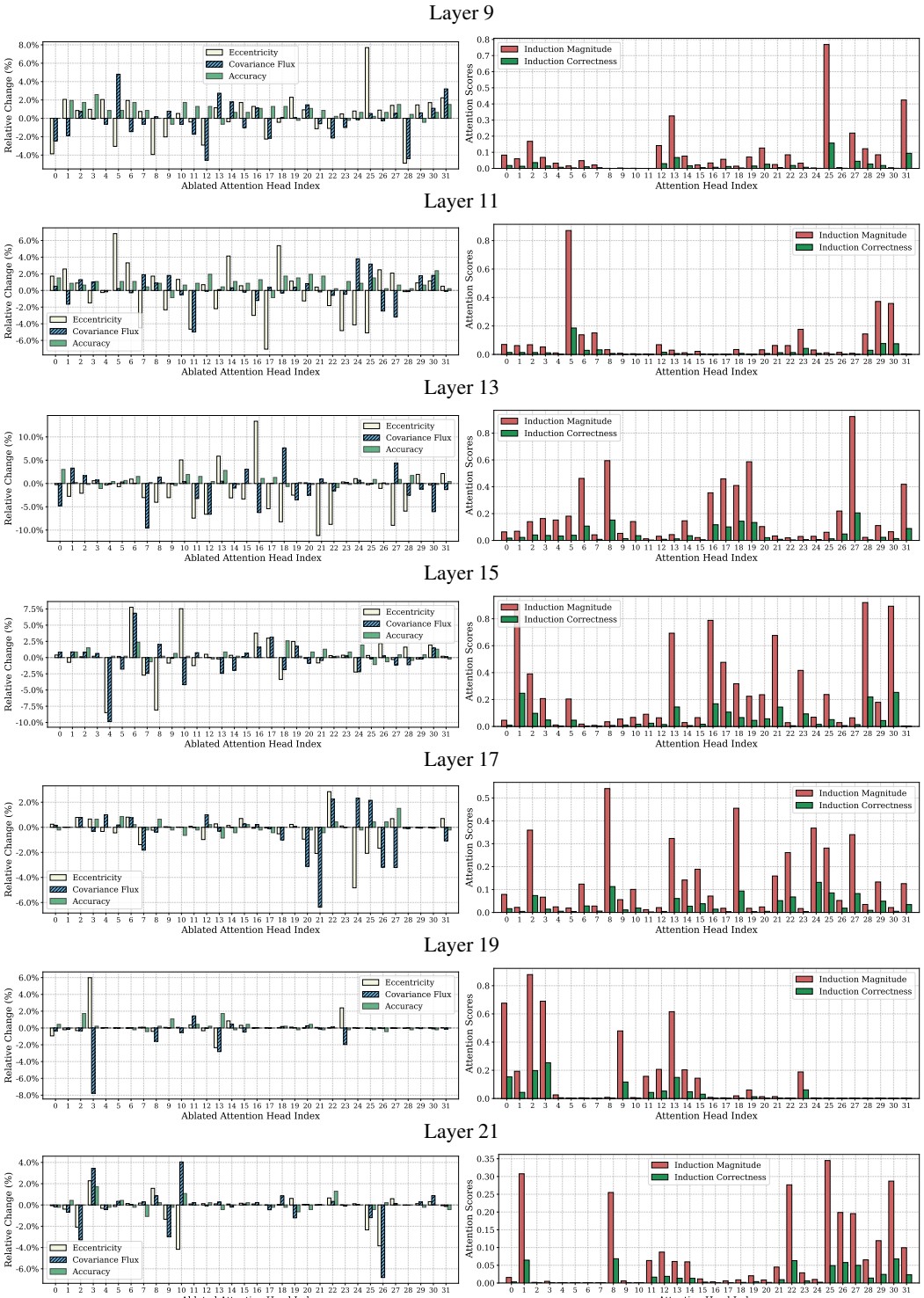

Figure 68: (Left) augmentation results for Fig. 7, (right) induction score of each attention head on Llama 3-8B, SST-5.

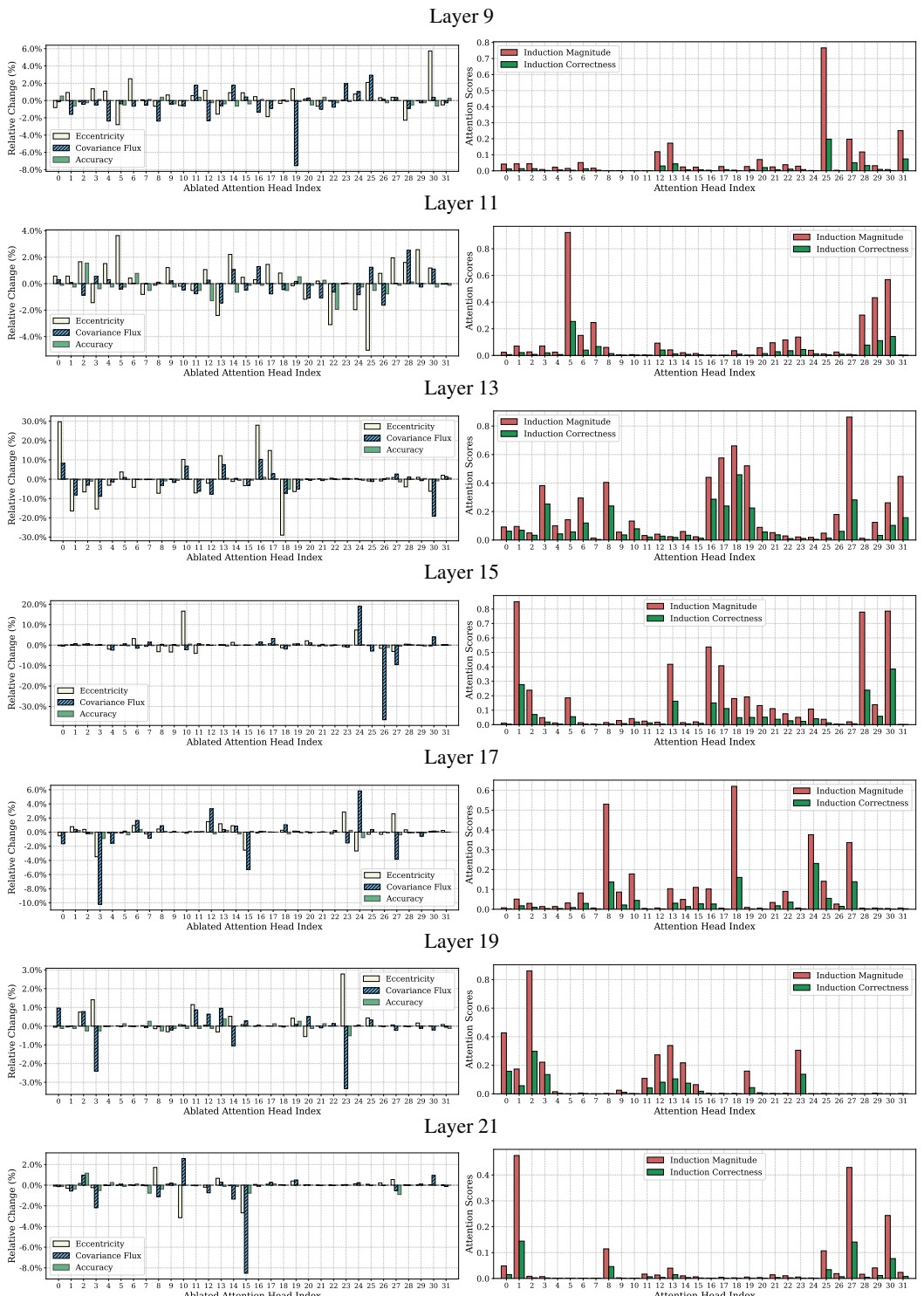

Figure 69: (Left) augmentation results for Fig. 7, (right) induction score of each attention head on Llama 3-8B, AGNews.

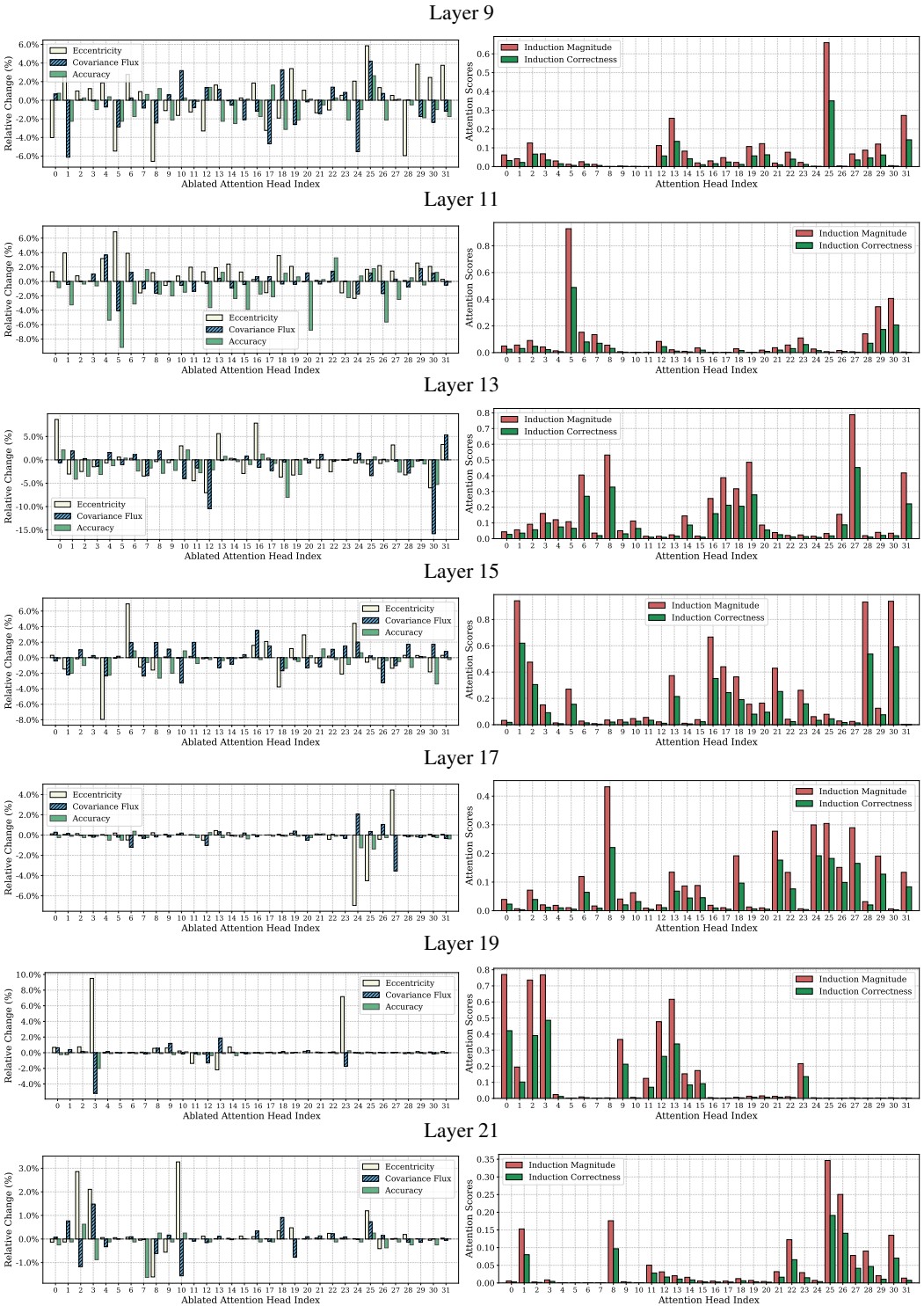

Figure 70: (Left) augmentation results for Fig. 7, (right) induction score of each attention head on Llama 3-8B, Subjective.

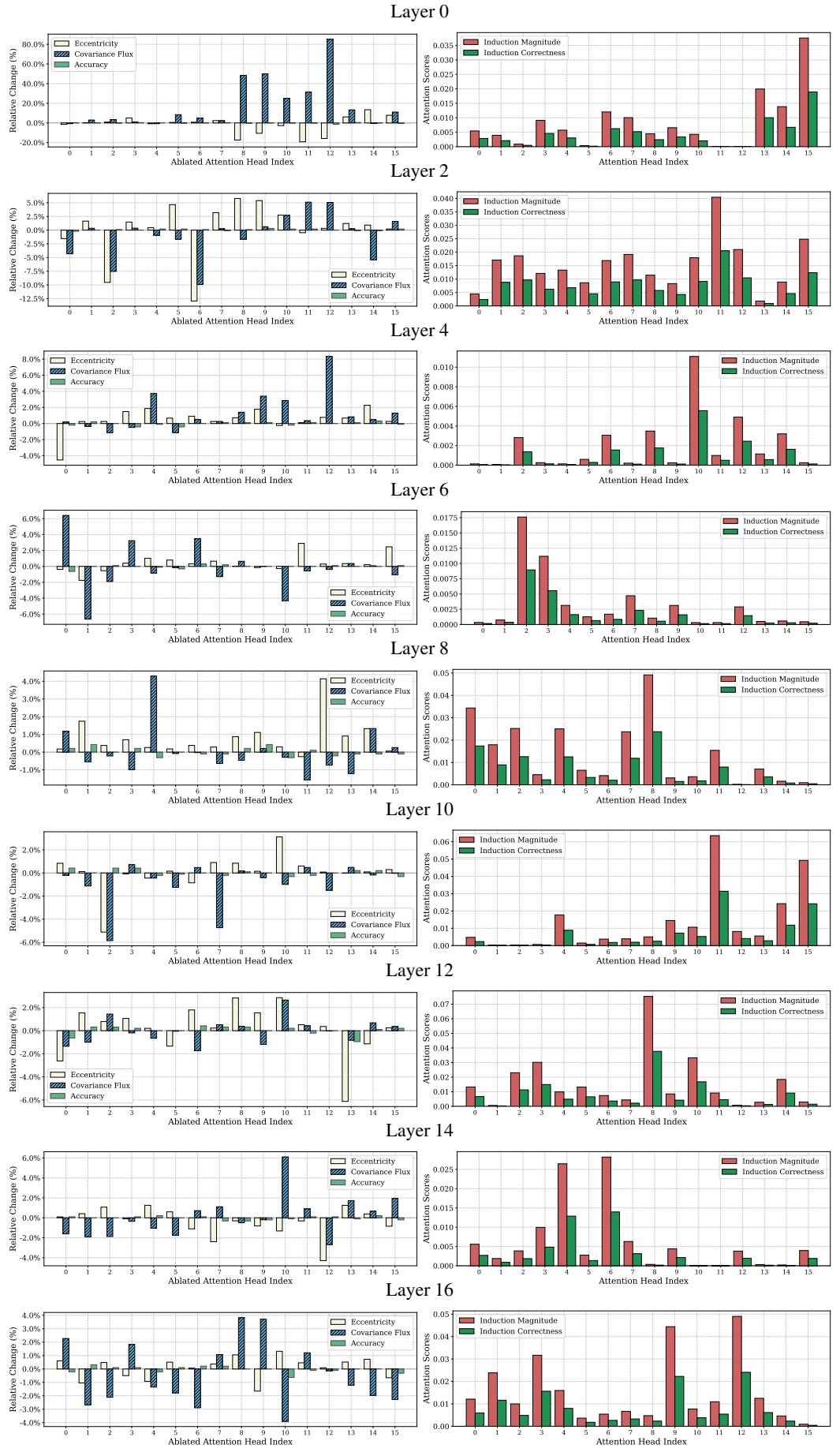

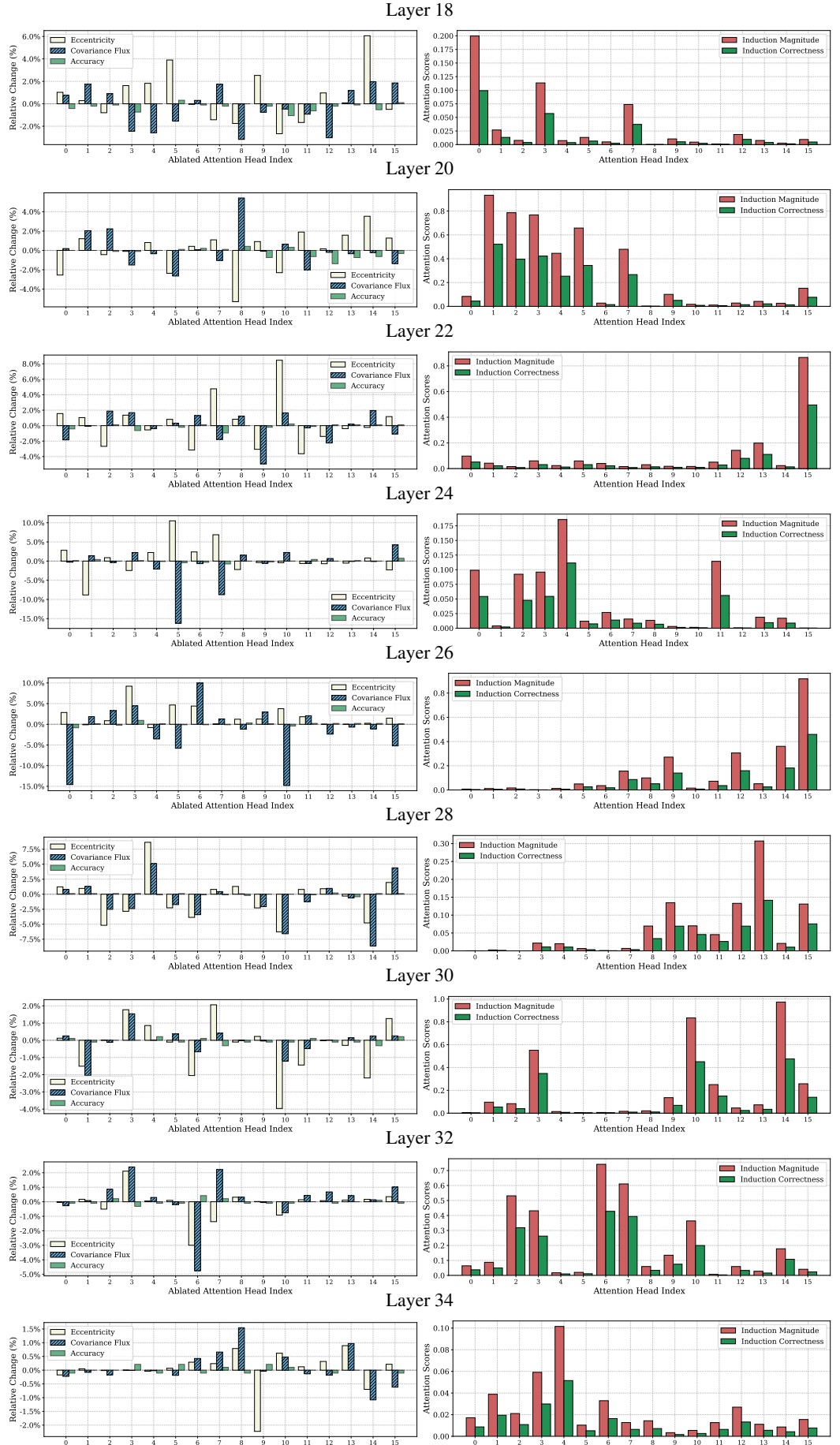

Figure 71: (Left) augmentation results for Fig. 7, (right) induction score of each attention head on Qwen 2.5-3B, SST-2.

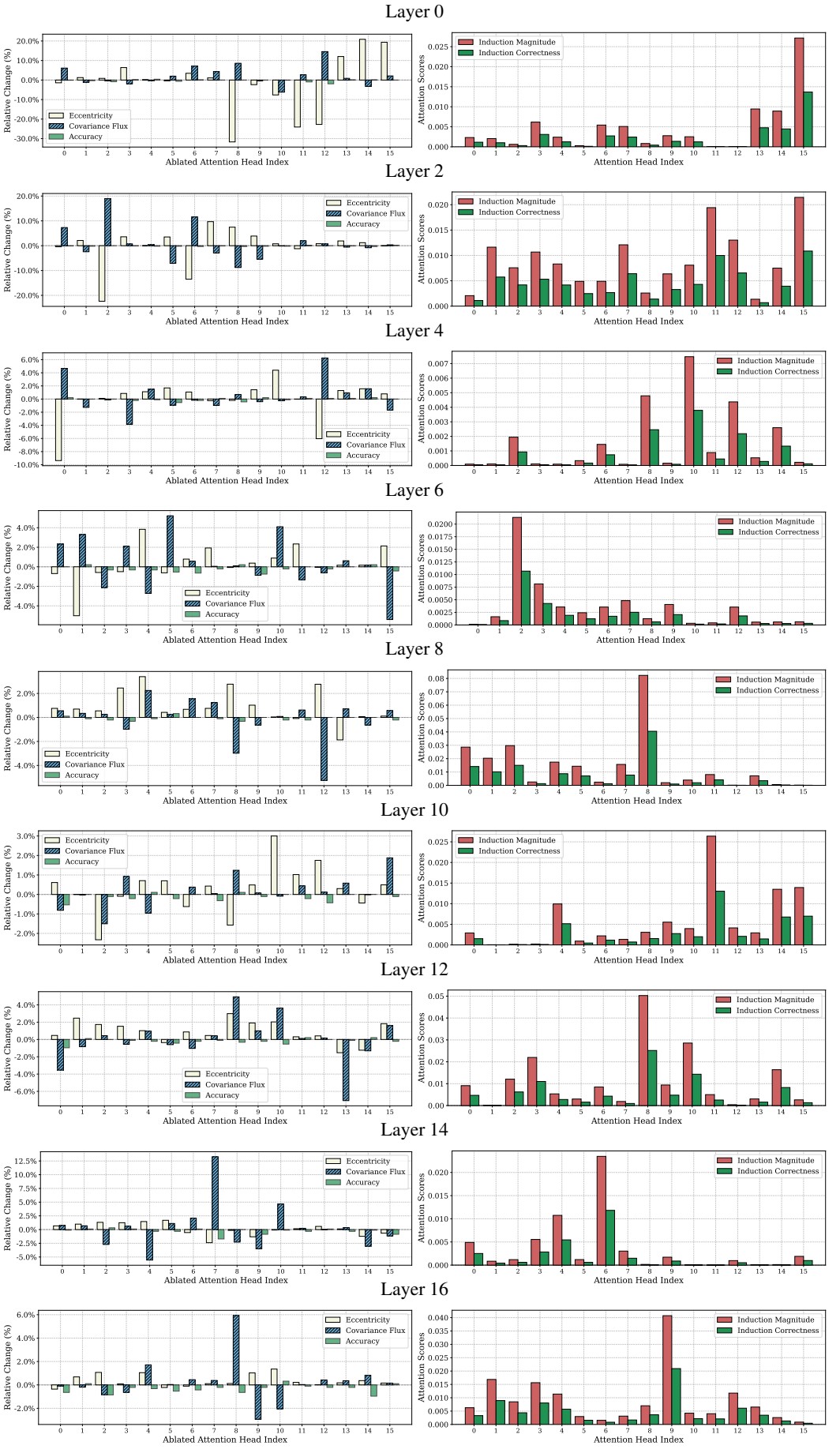

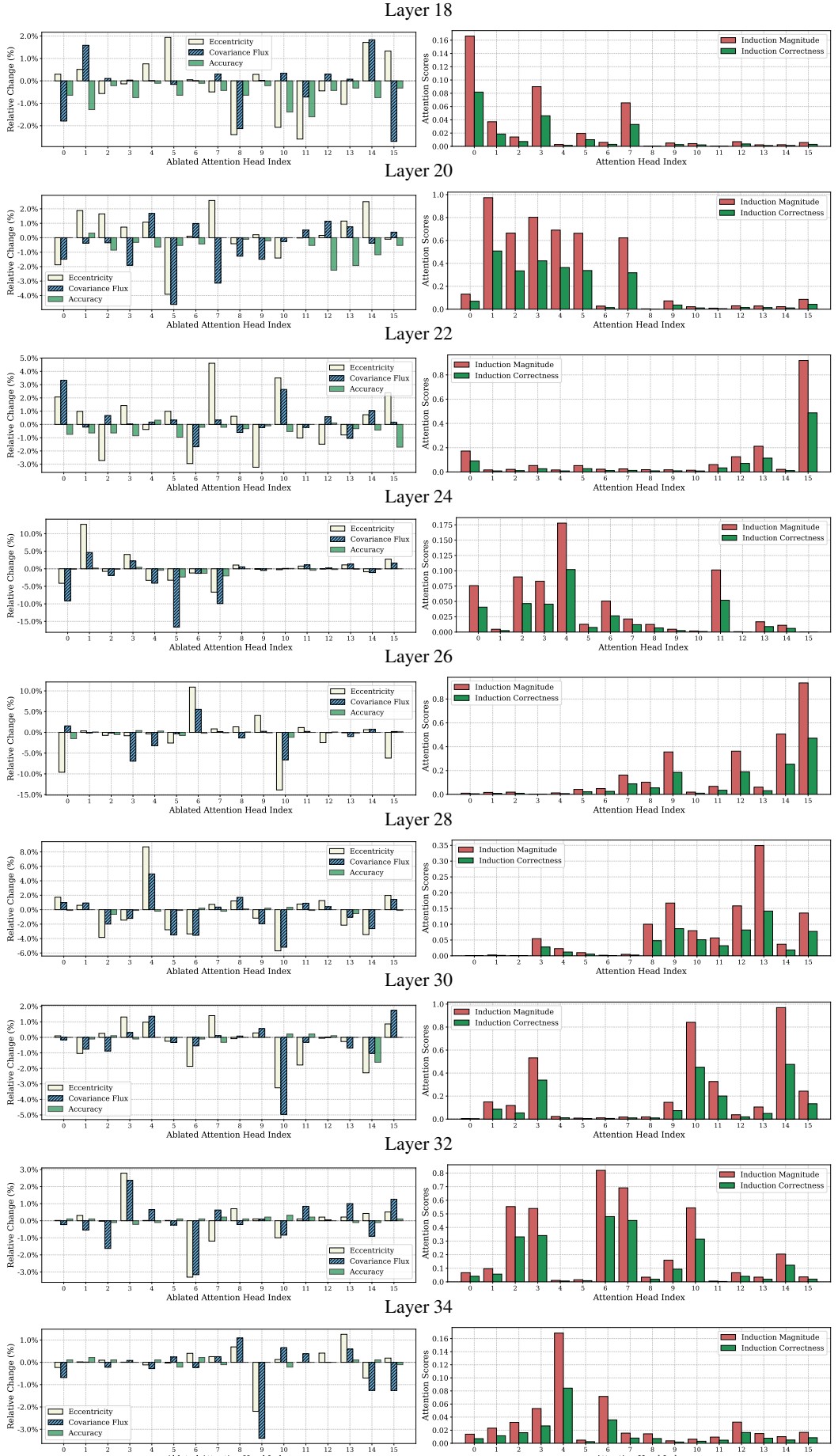

Figure 72: (Left) augmentation results for Fig. 7, (right) induction score of each attention head on Qwen 2.5-3B, MR.

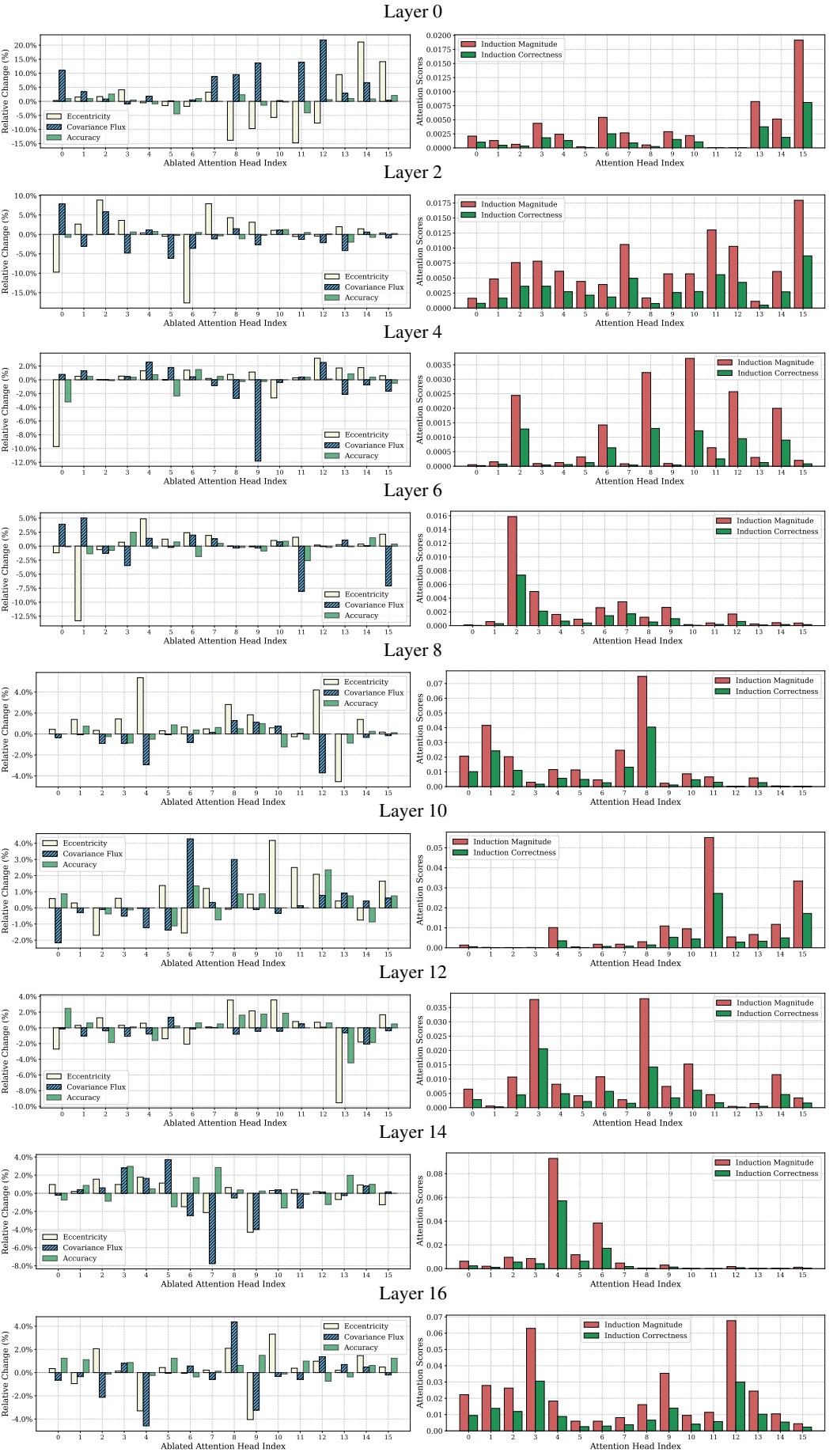

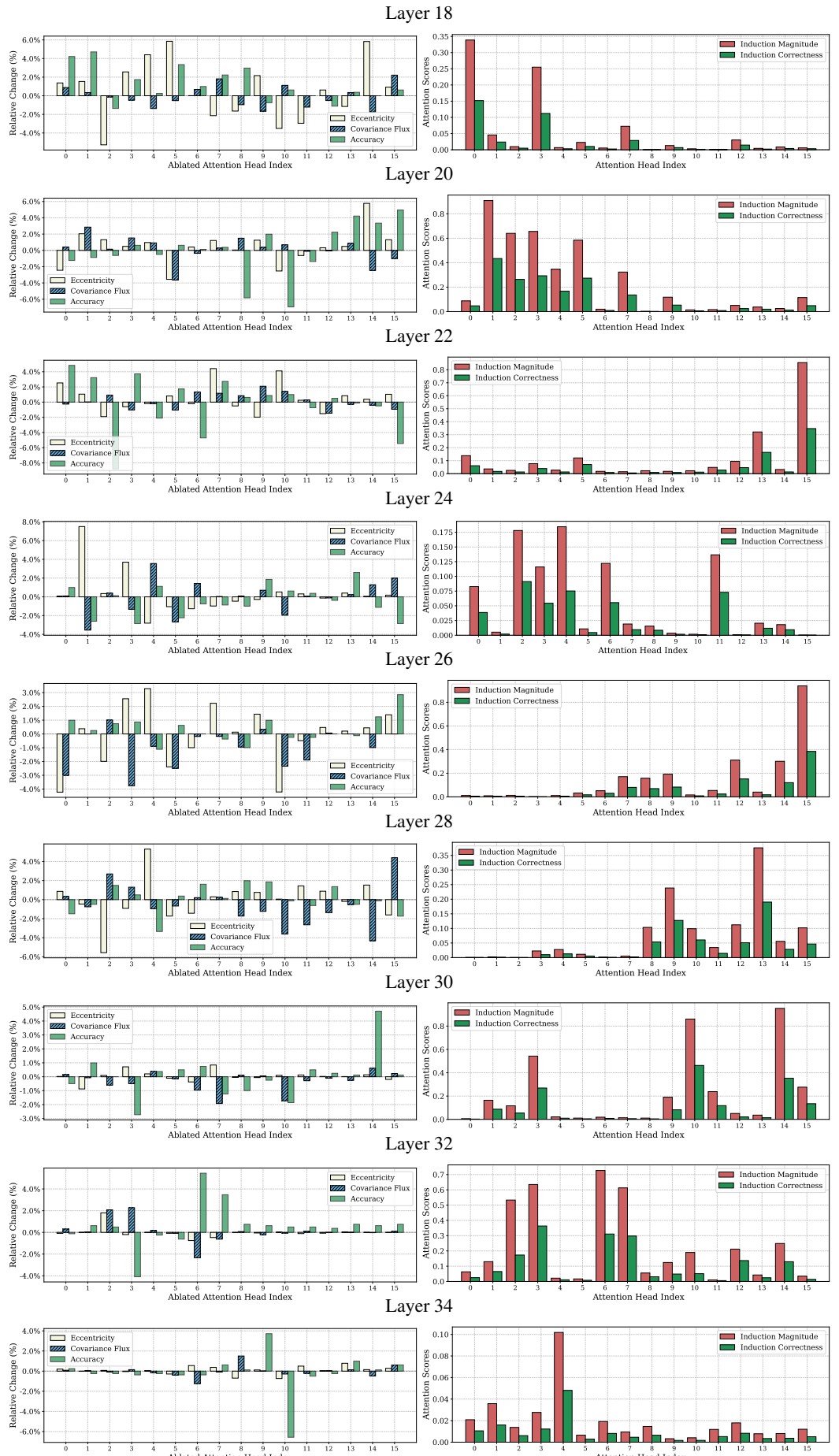

Figure 73: (Left) augmentation results for Fig. 7, (right) induction score of each attention head on Qwen 2.5-3B, FP.

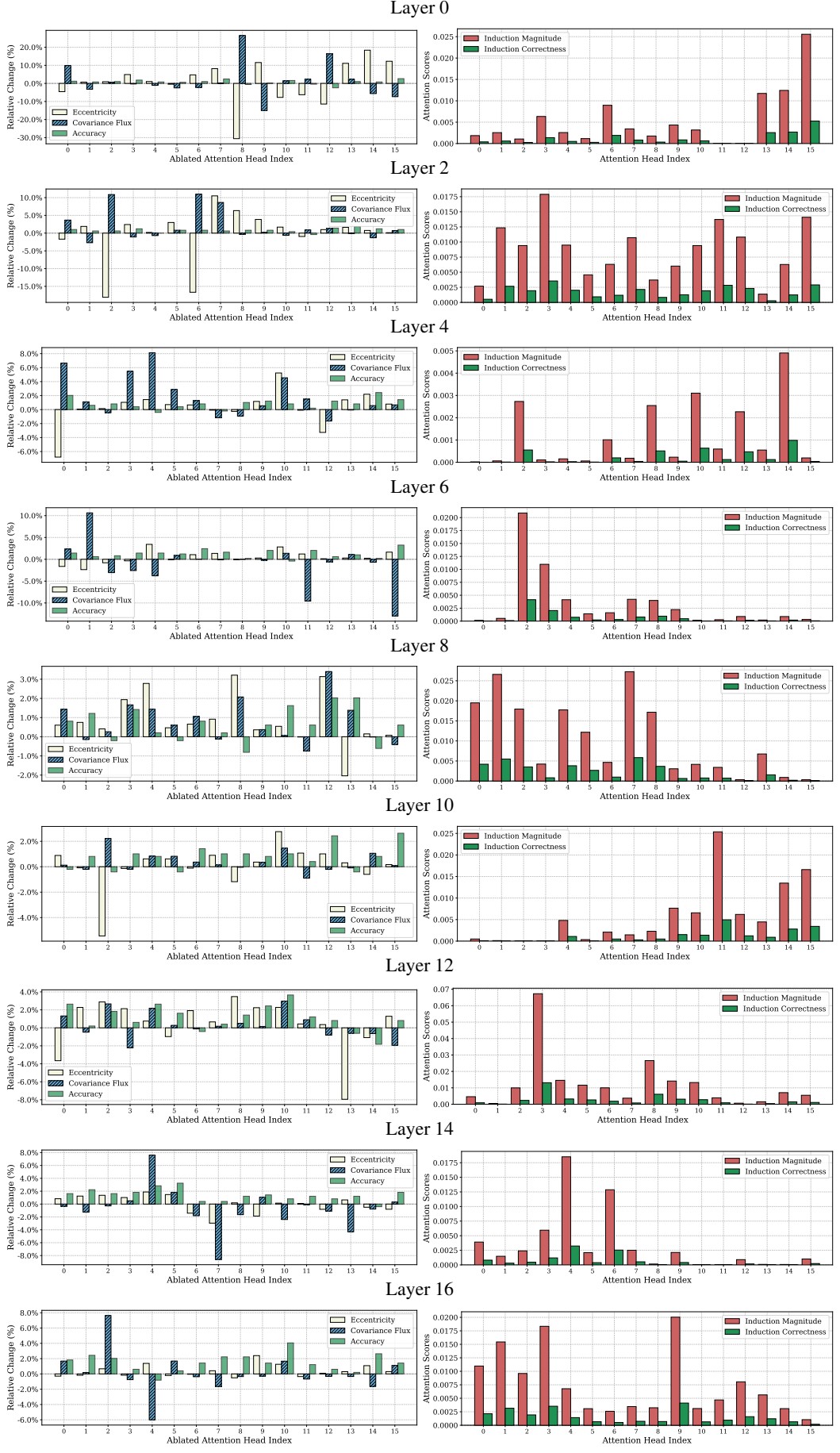

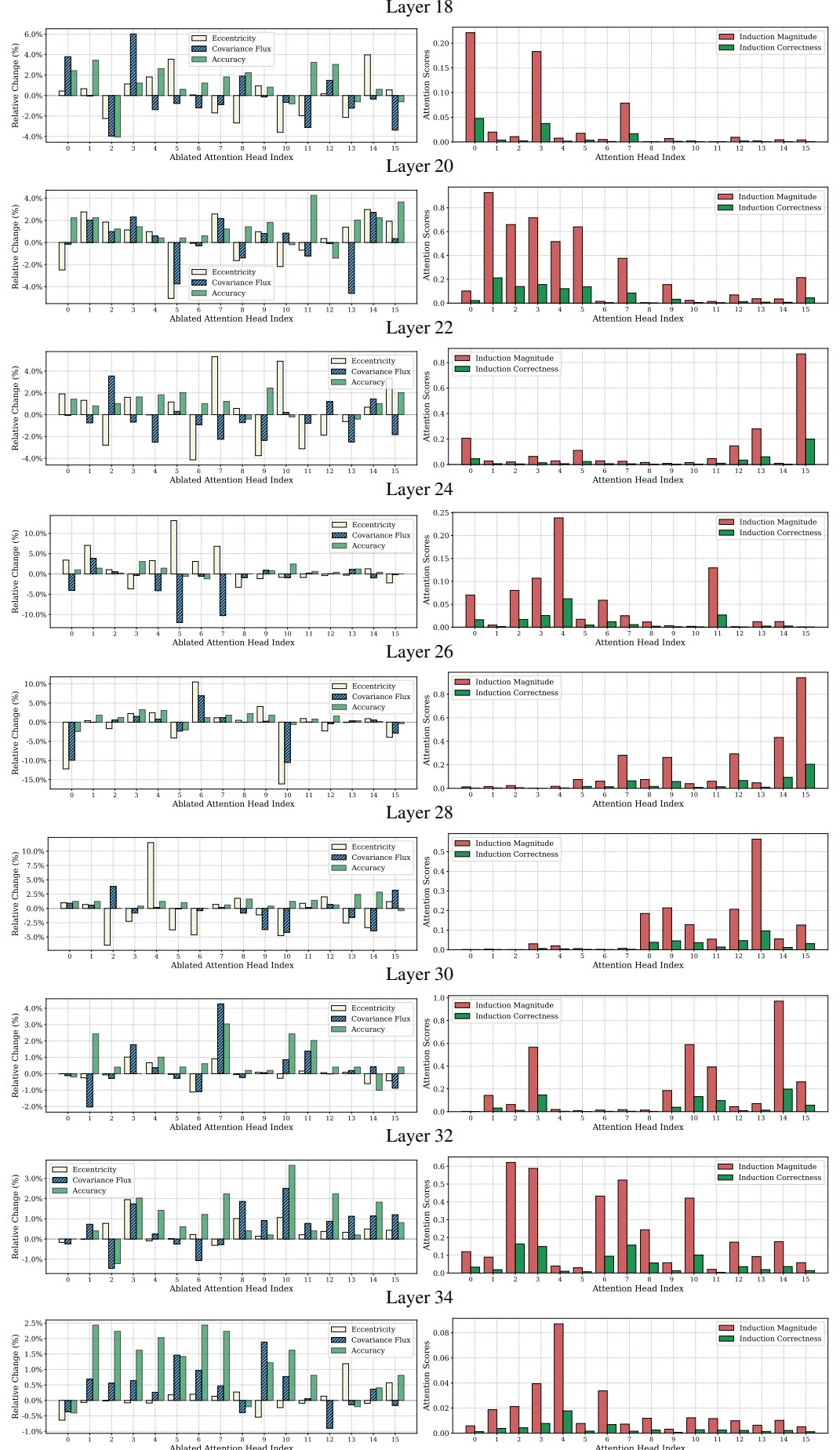

Figure 74: (Left) augmentation results for Fig. 7, (right) induction score of each attention head on Qwen 2.5-3B, SST-5.

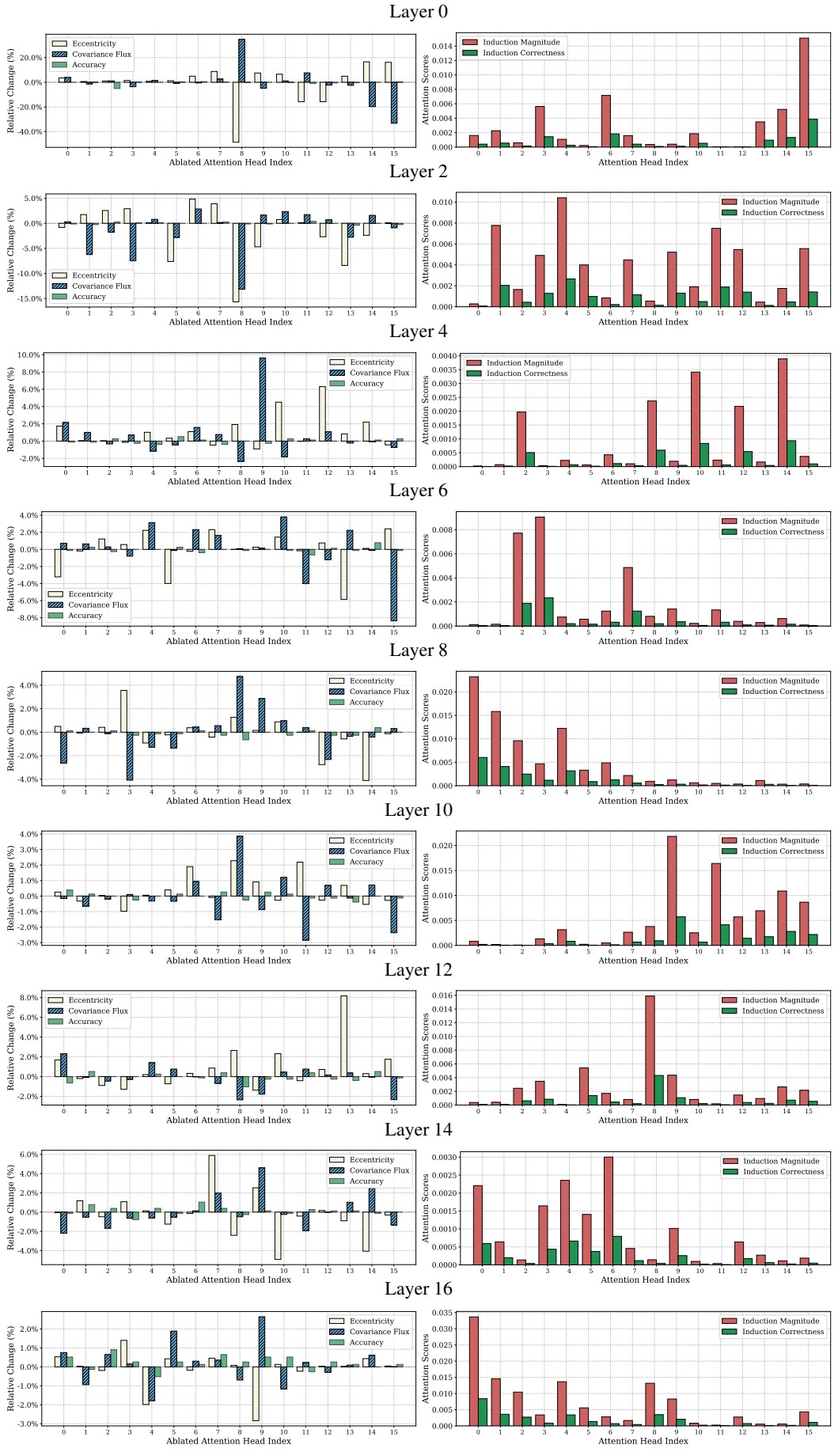

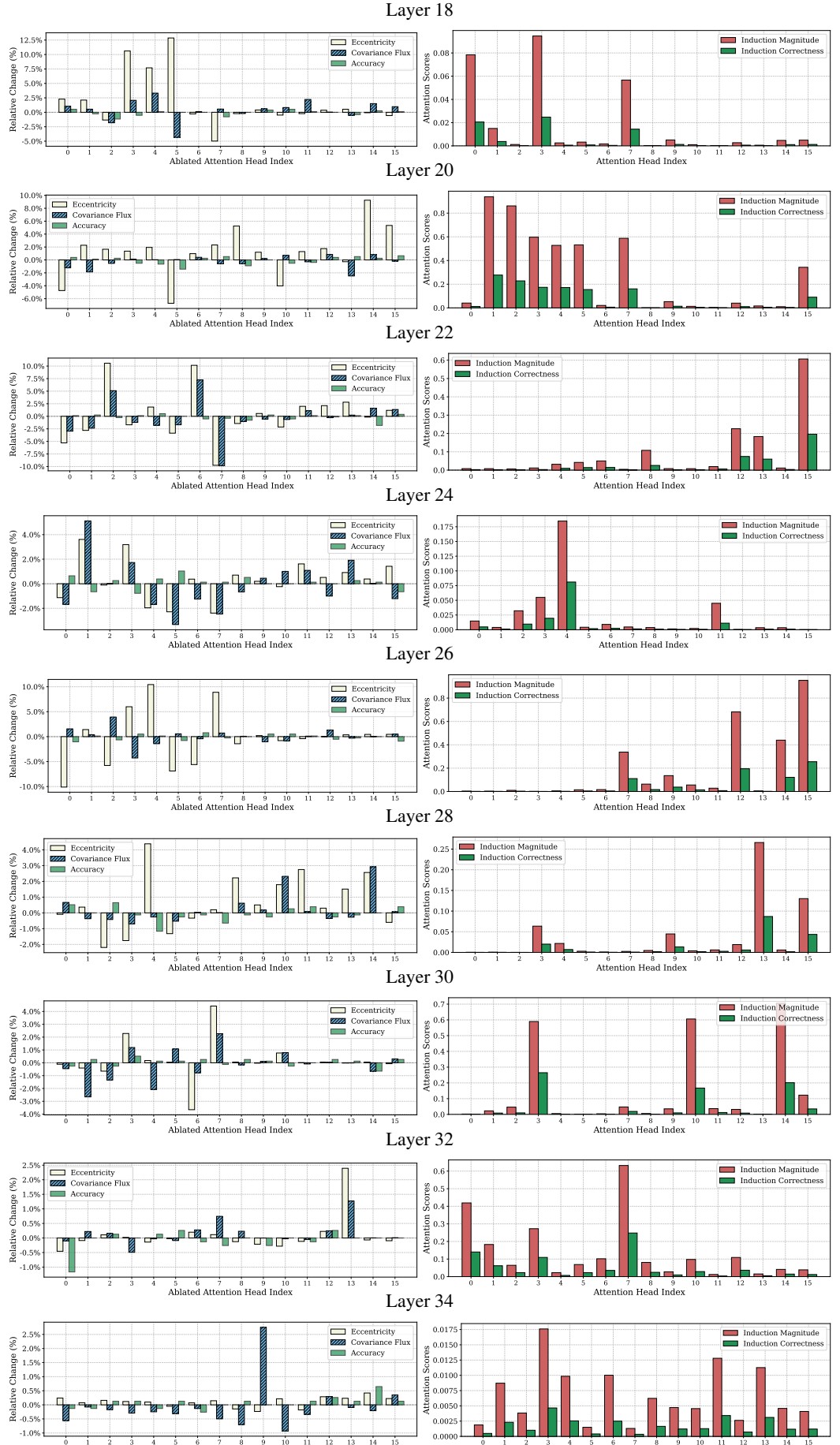

Figure 75: (Left) augmentation results for Fig. 7, (right) induction score of each attention head on Qwen 2.5-3B, AGNews.

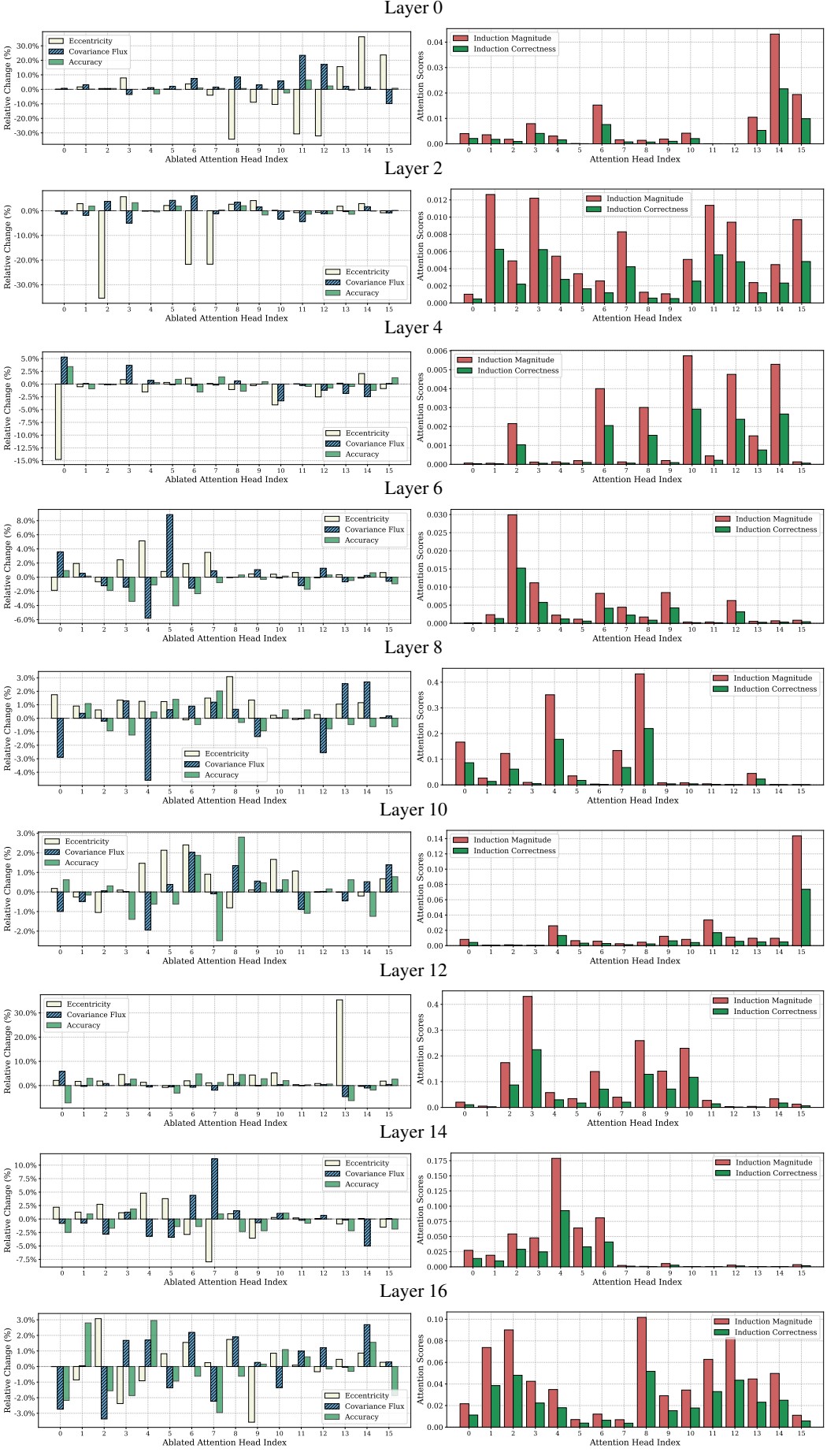

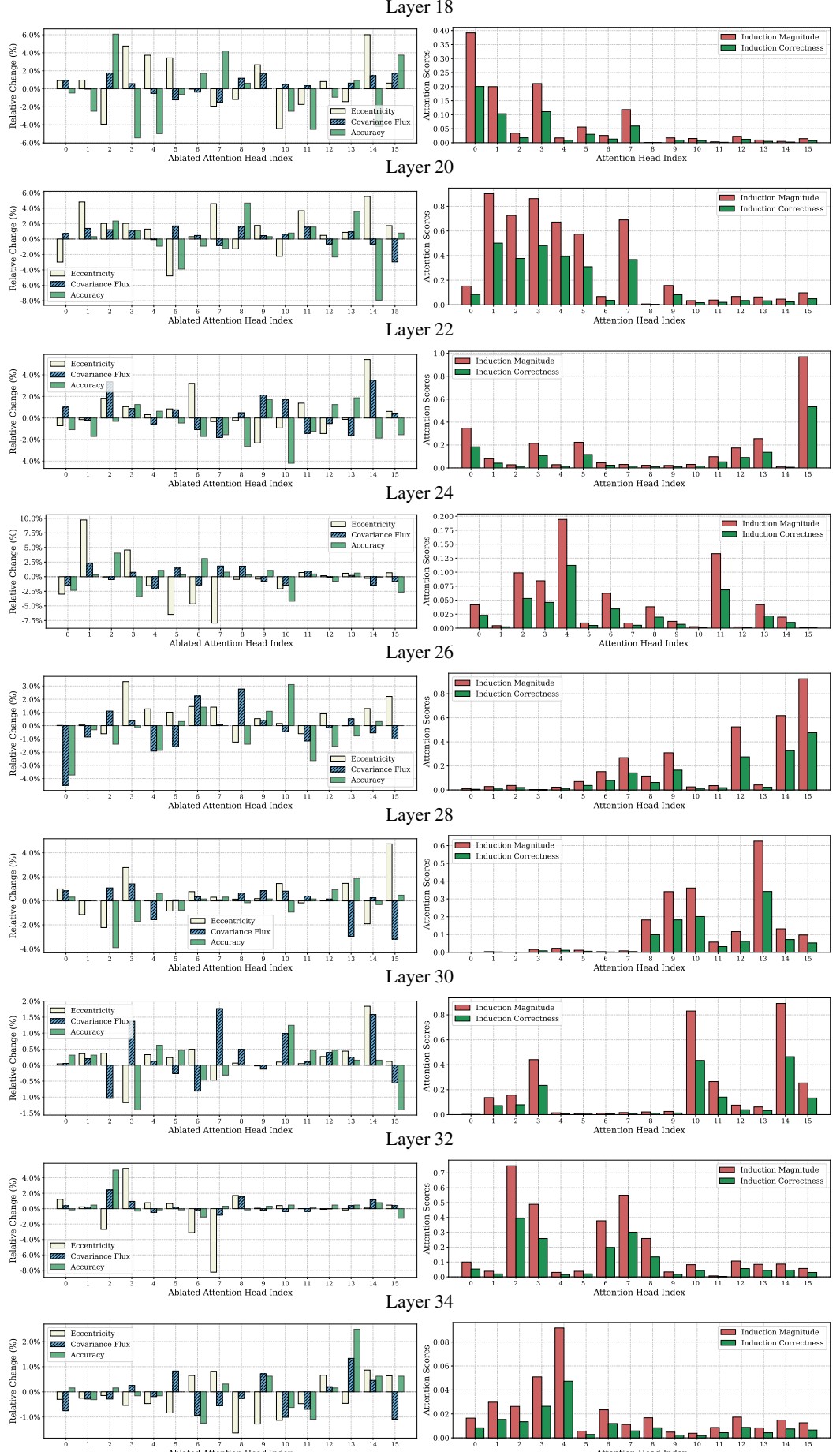

Figure 76: (Left) augmentation results for Fig. 7, (right) induction score of each attention head on Qwen 2.5-3B, Subjective.

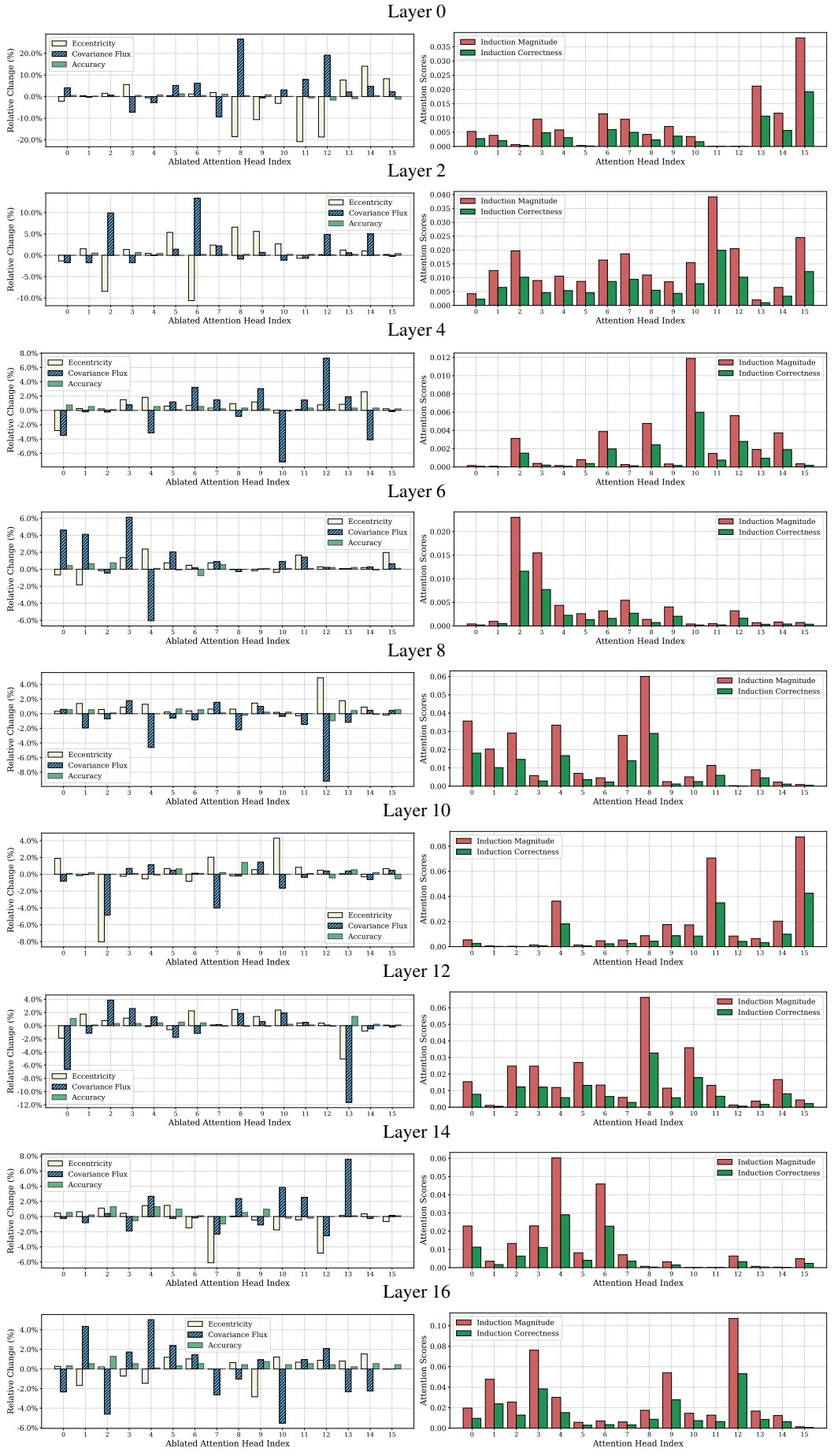

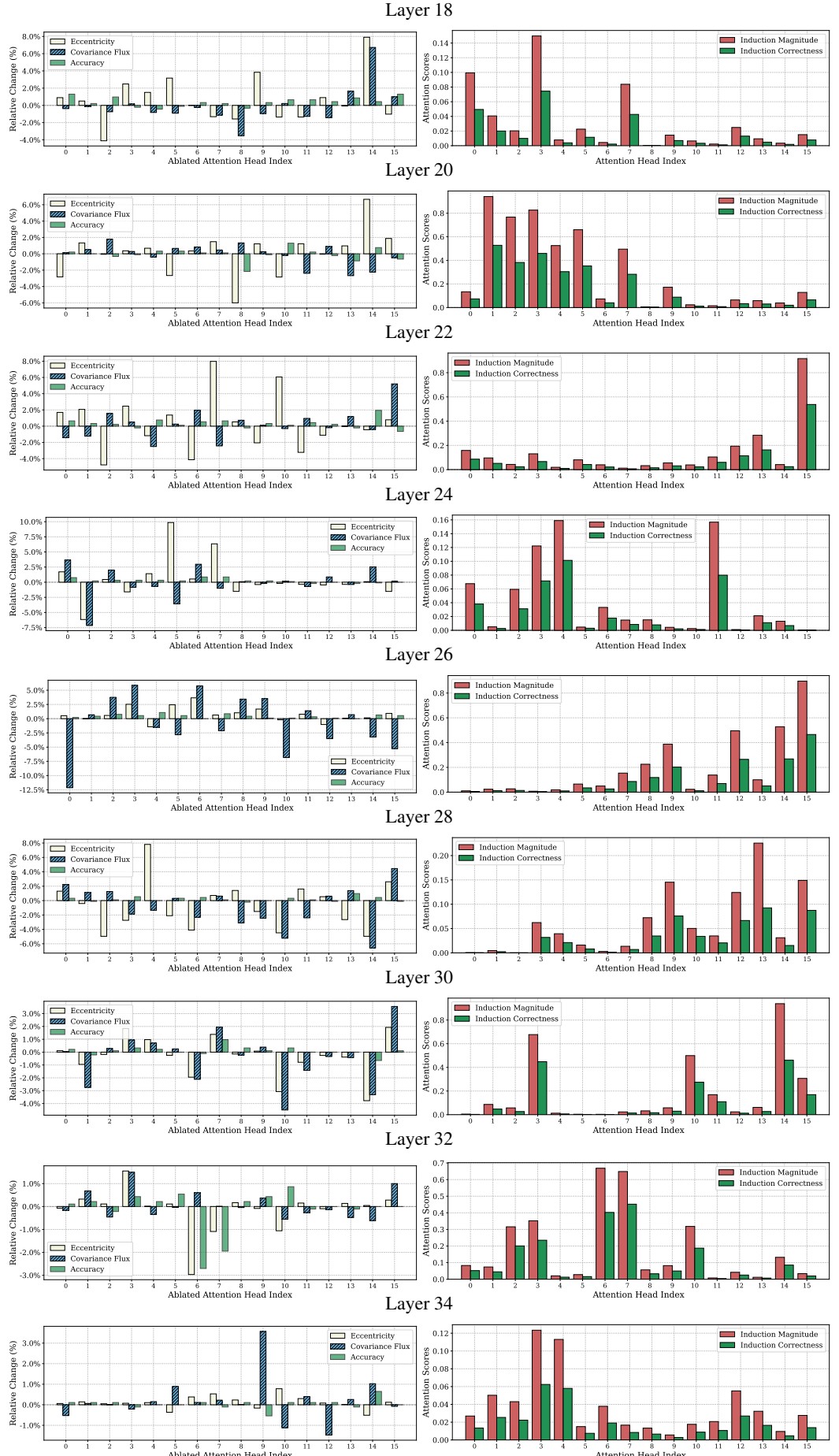

Figure 77: (Left) augmentation results for Fig. 7, (right) induction score of each attention head on Qwen 2.5-3B Instruct, SST-2.

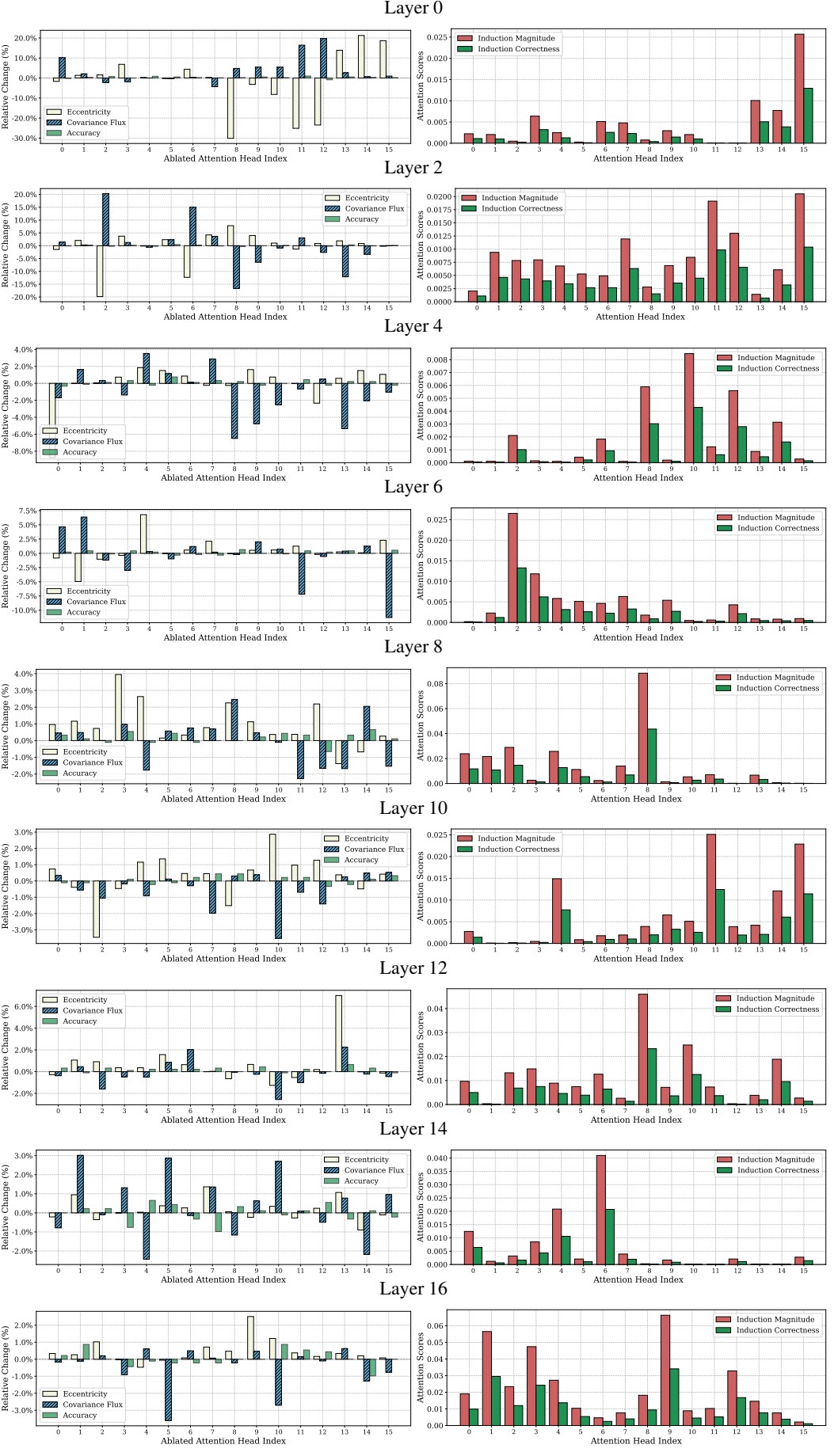

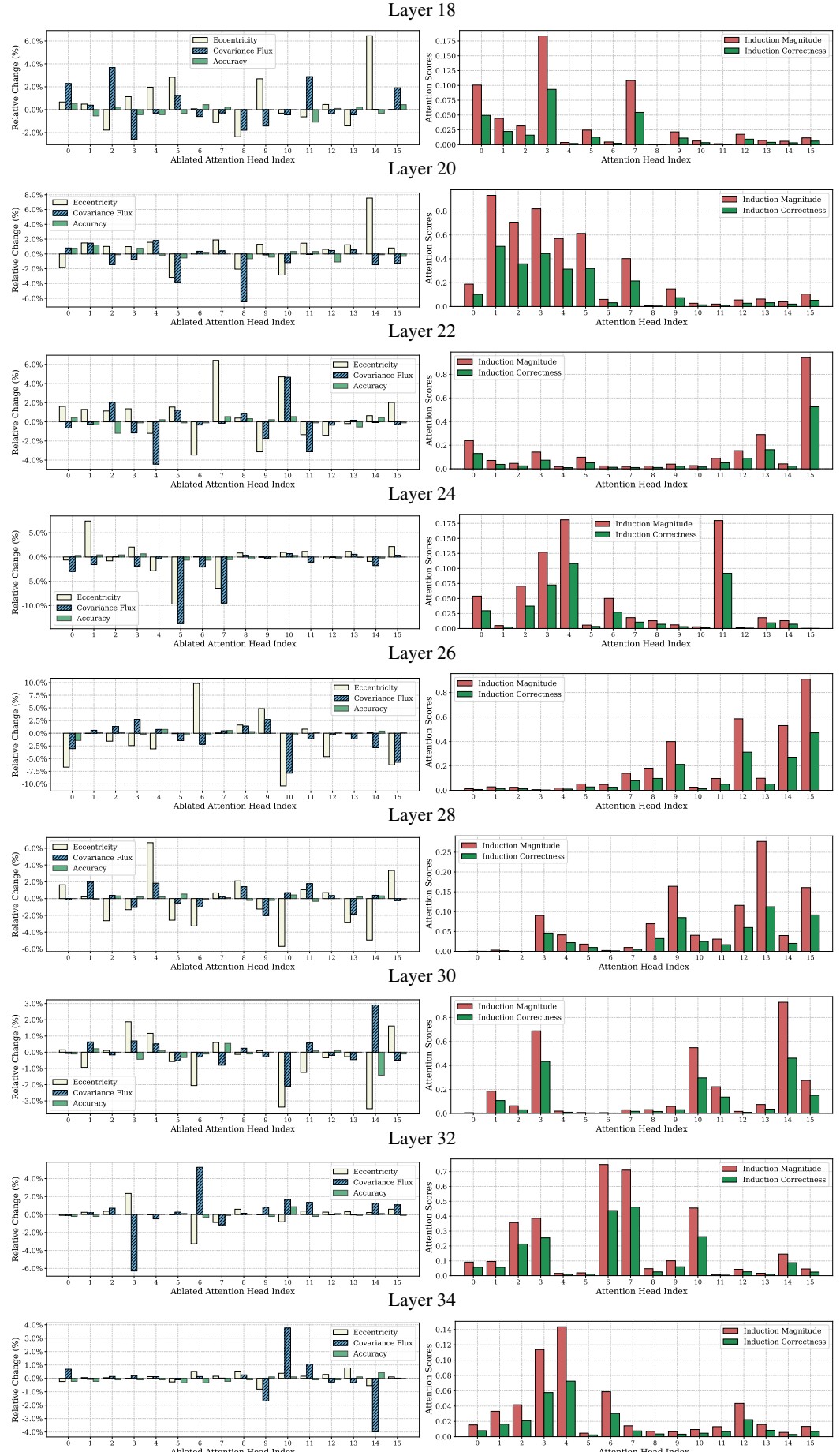

Figure 78: (Left) augmentation results for Fig. 7, (right) induction score of each attention head on Qwen 2.5-3B Instruct, MR.

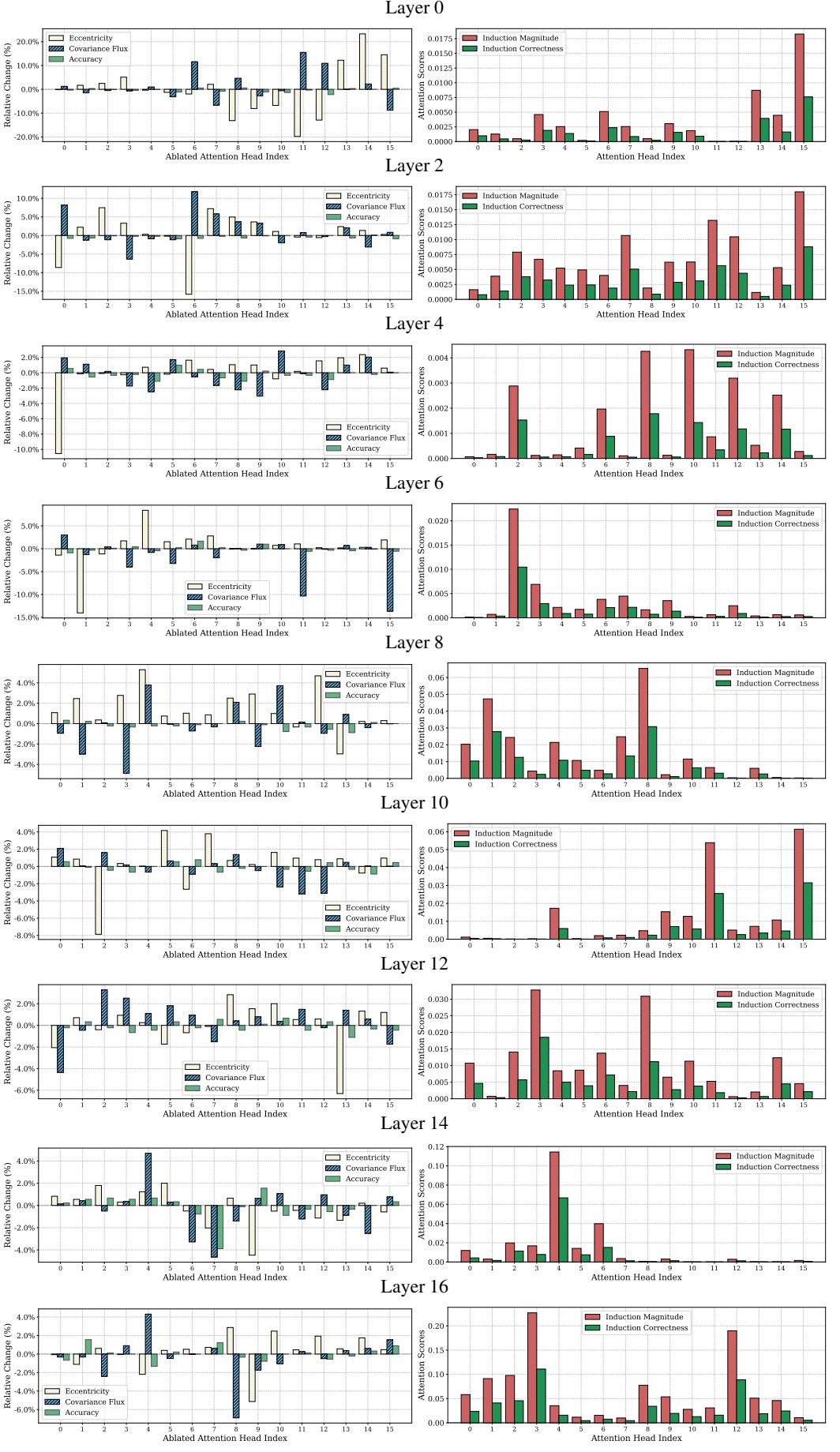

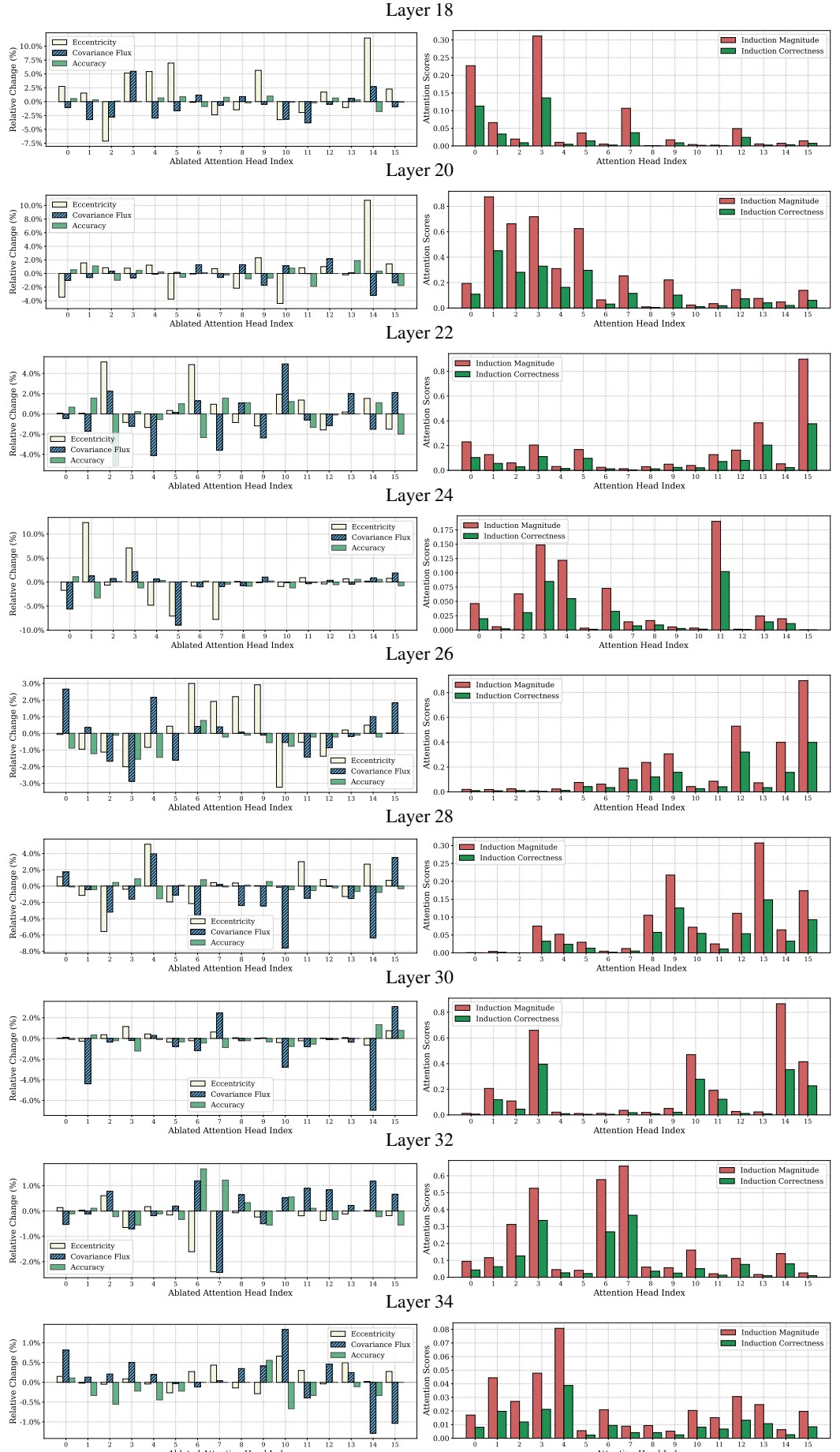

Figure 79: (Left) augmentation results for Fig. 7, (right) induction score of each attention head on Qwen 2.5-3B Instruct, FP.

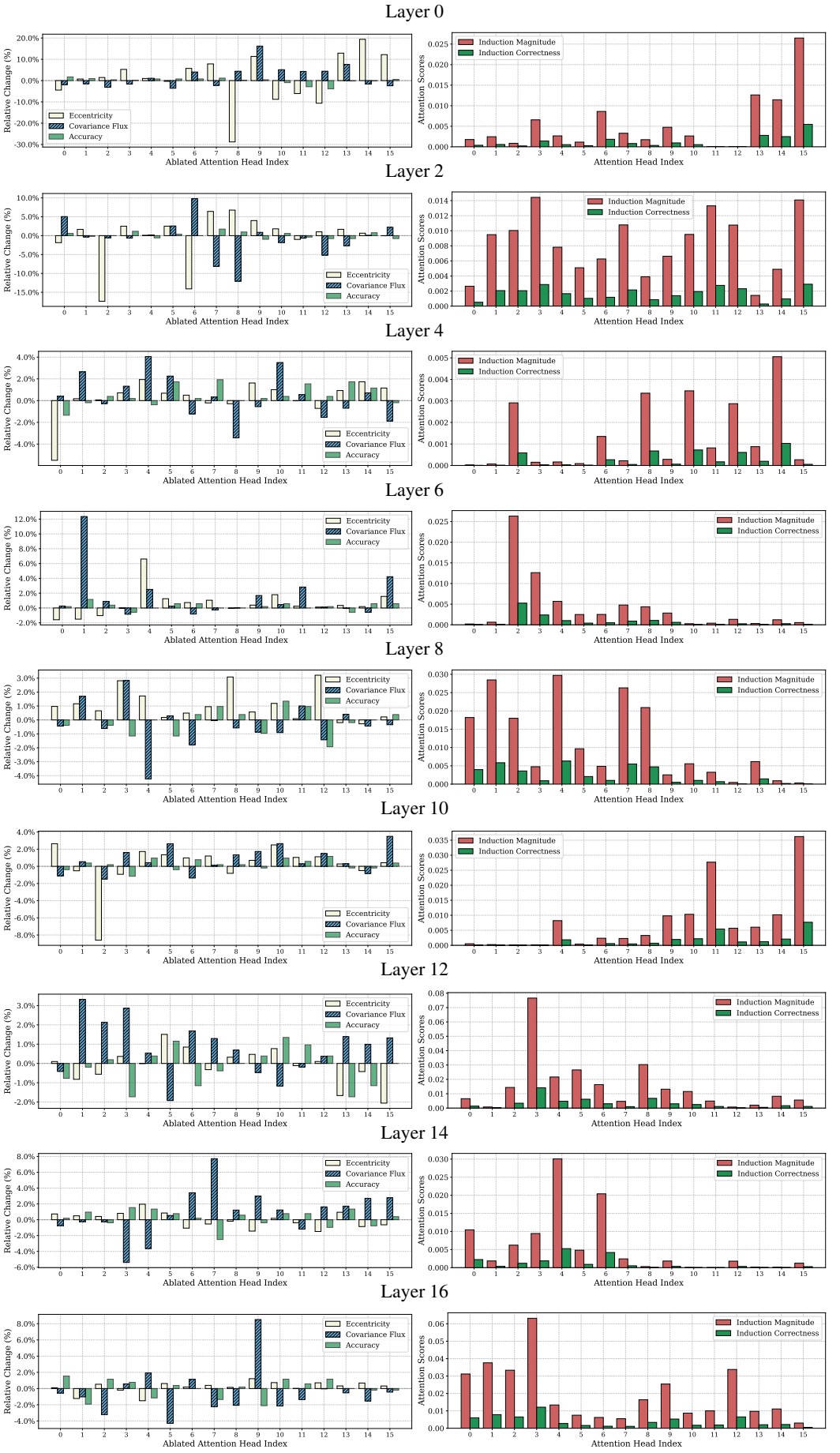

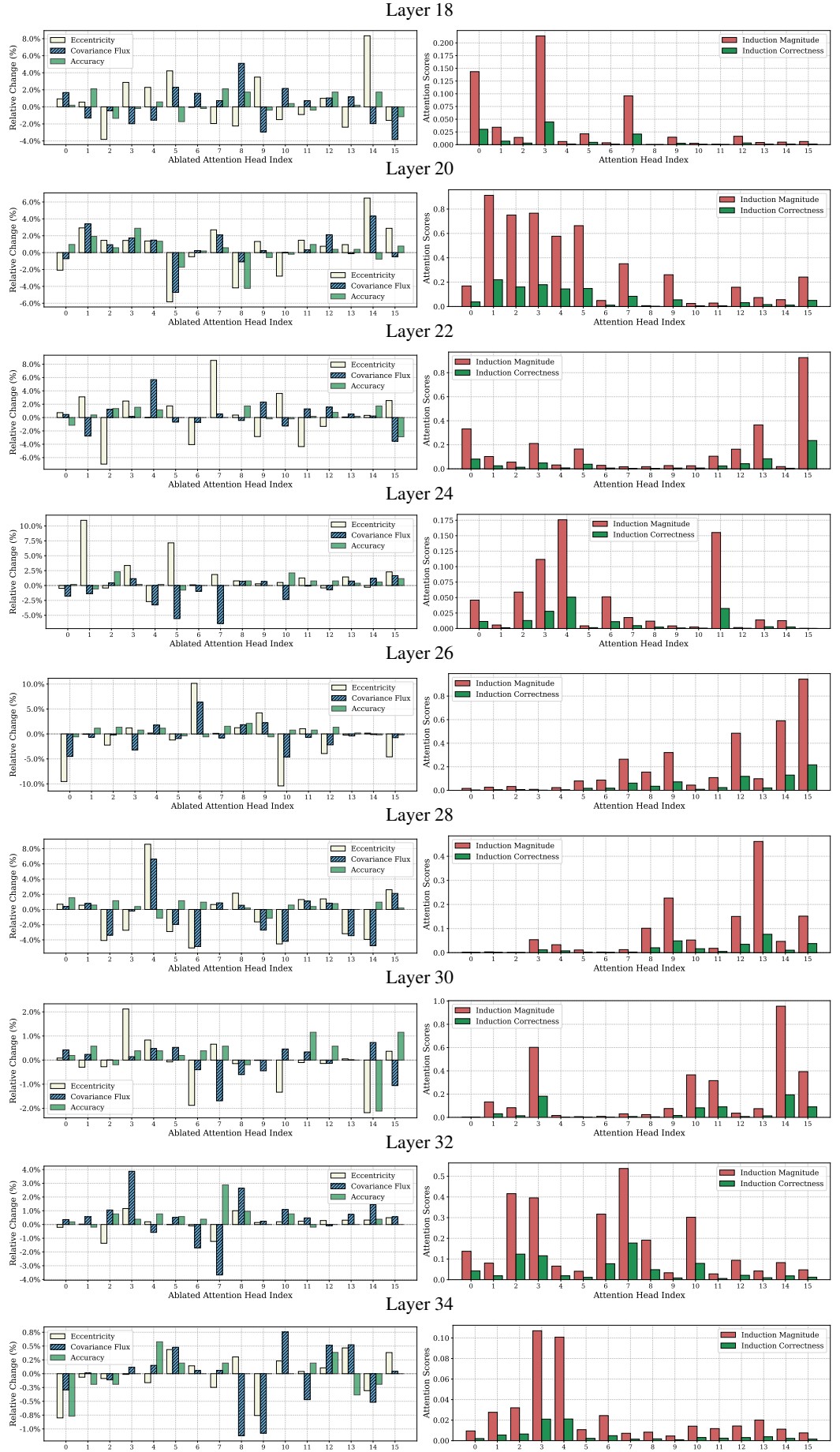

Figure 80: (Left) augmentation results for Fig. 7, (right) induction score of each attention head on Qwen 2.5-3B Instruct, SST-5.

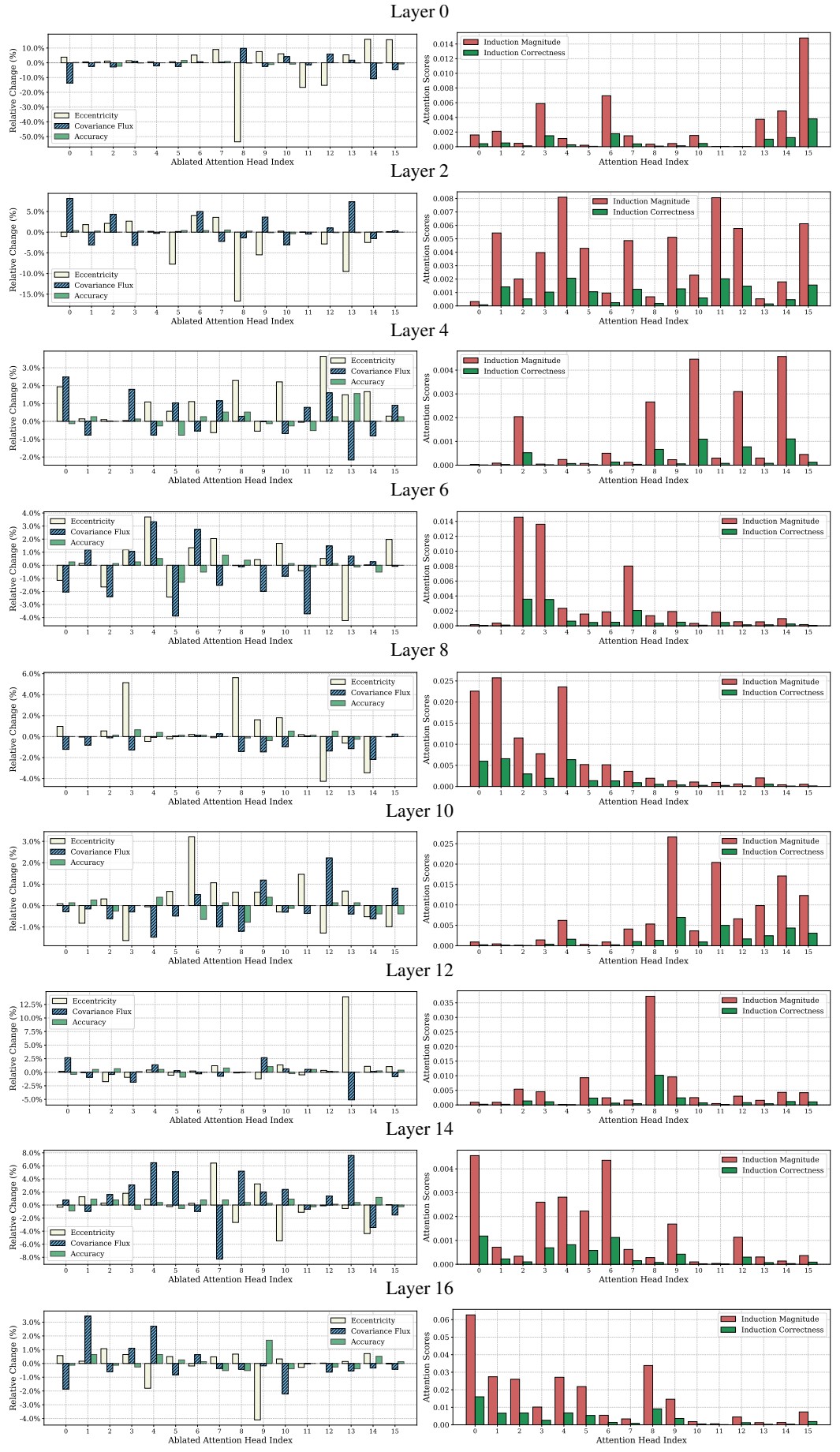

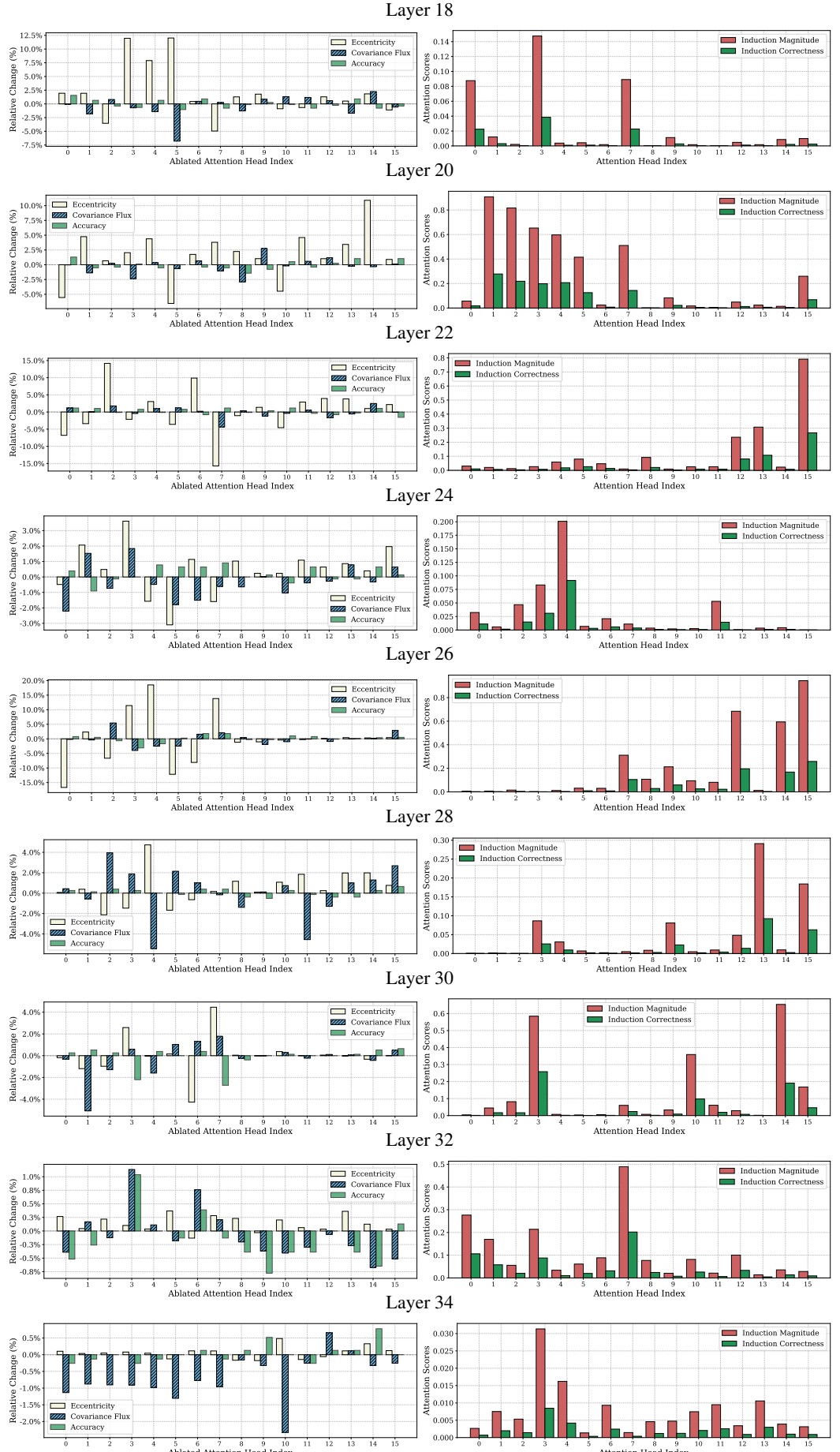

Figure 81: (Left) augmentation results for Fig. 7, (right) induction score of each attention head on Qwen 2.5-3B Instruct, AGNews.

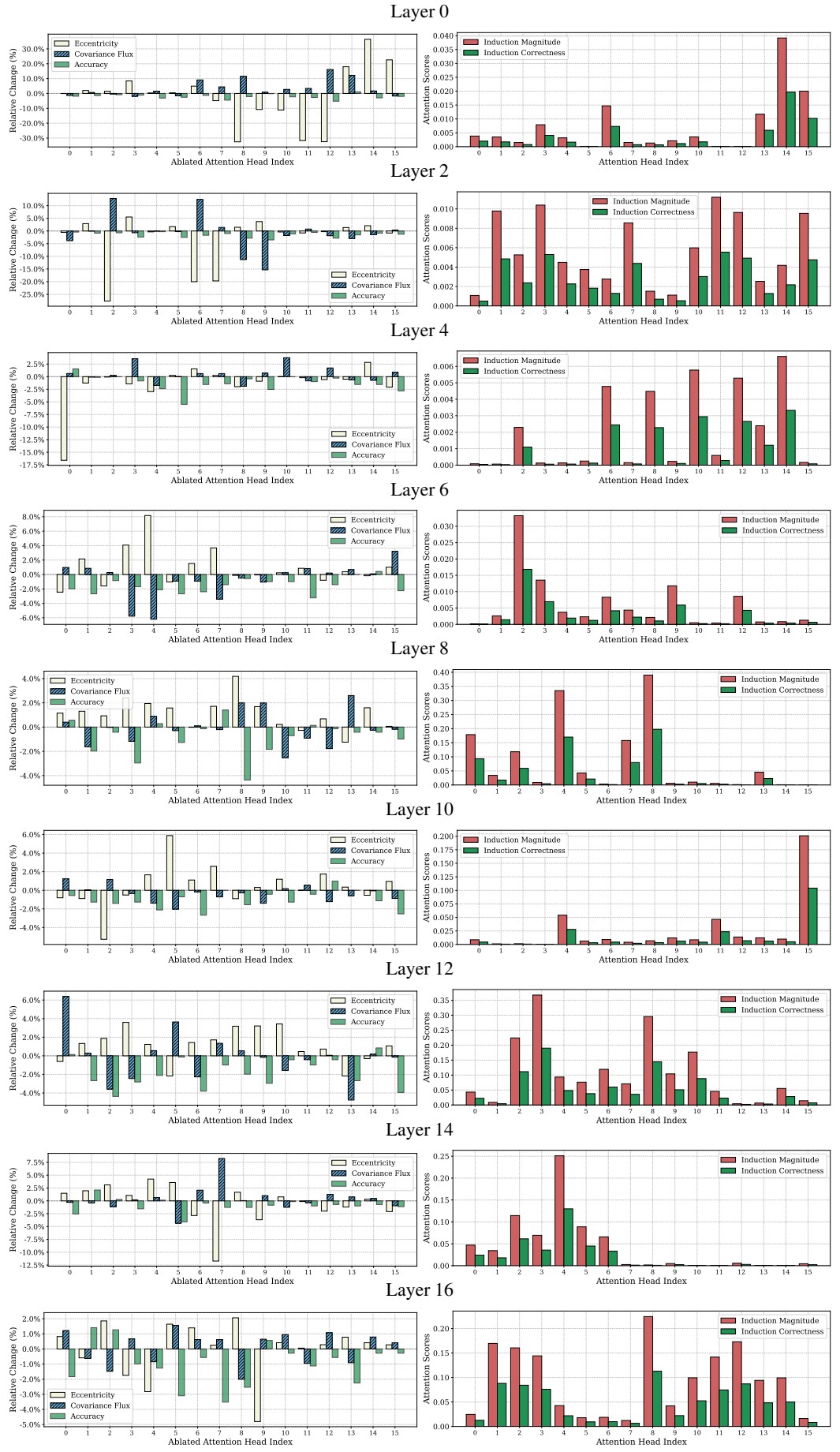

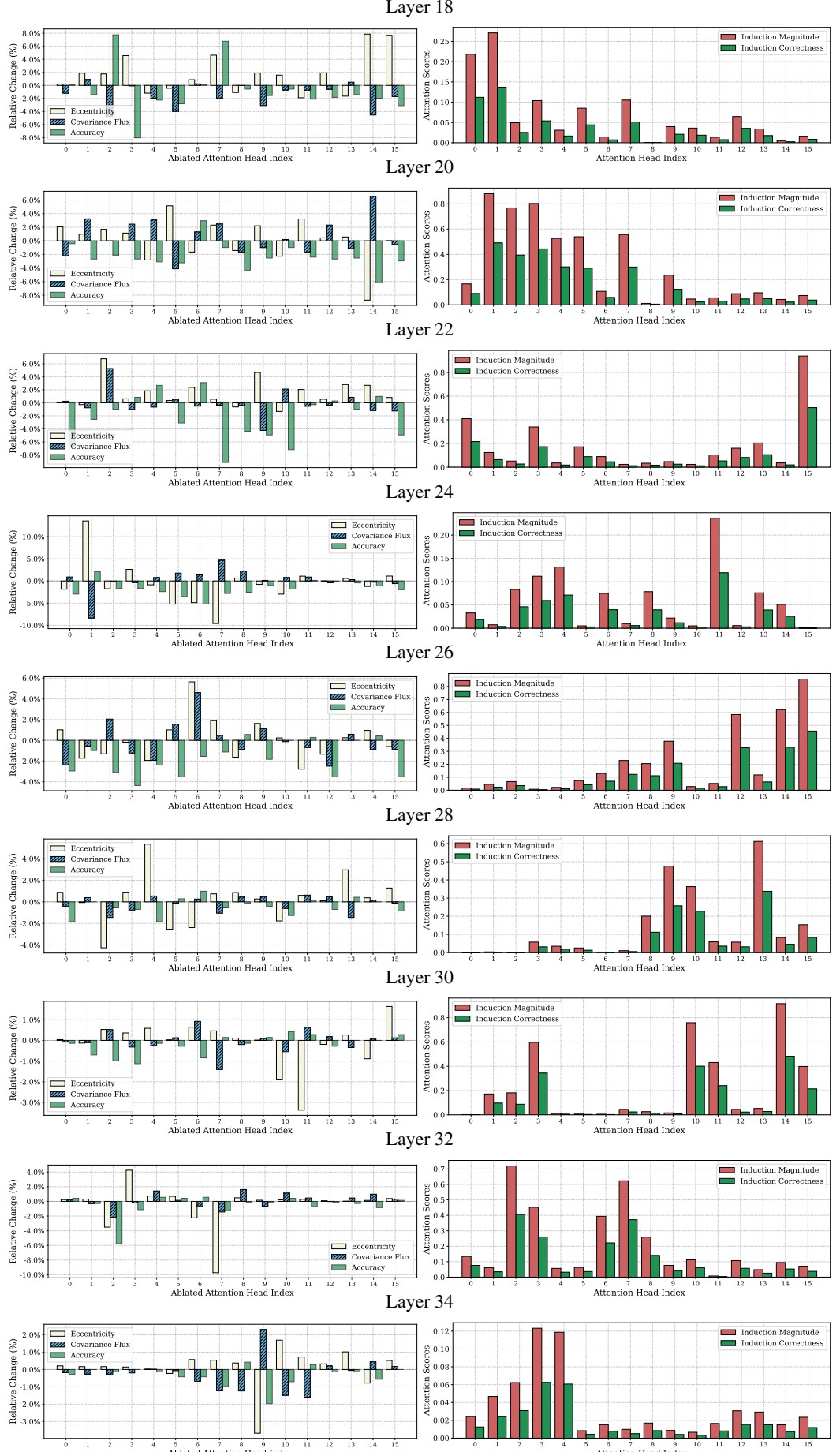

Figure 82: (Left) augmentation results for Fig. 7, (right) induction score of each attention head on Qwen 2.5-3B Instruct, Subjective.

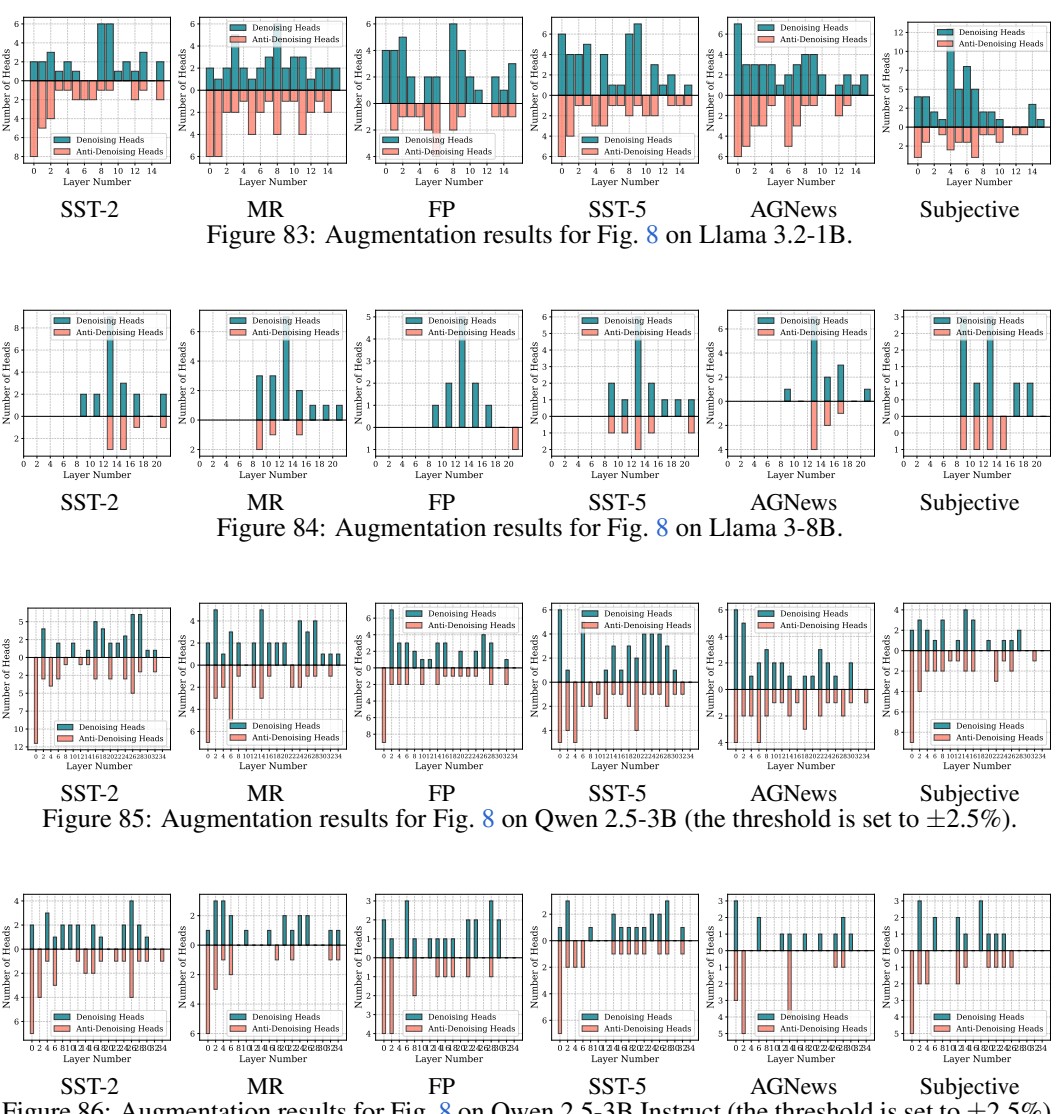

Figure 83: Augmentation results for Fig. 8 on Llama 3.2-1B.

Figure 84: Augmentation results for Fig. 8 on Llama 3-8B.

Figure 85: Augmentation results for Fig. 8 on Qwen 2.5-3B (the threshold is set to $\pm 2.5\%$).

Figure 86: Augmentation results for Fig. 8 on Qwen 2.5-3B Instruct (the threshold is set to $\pm 2.5\%$).

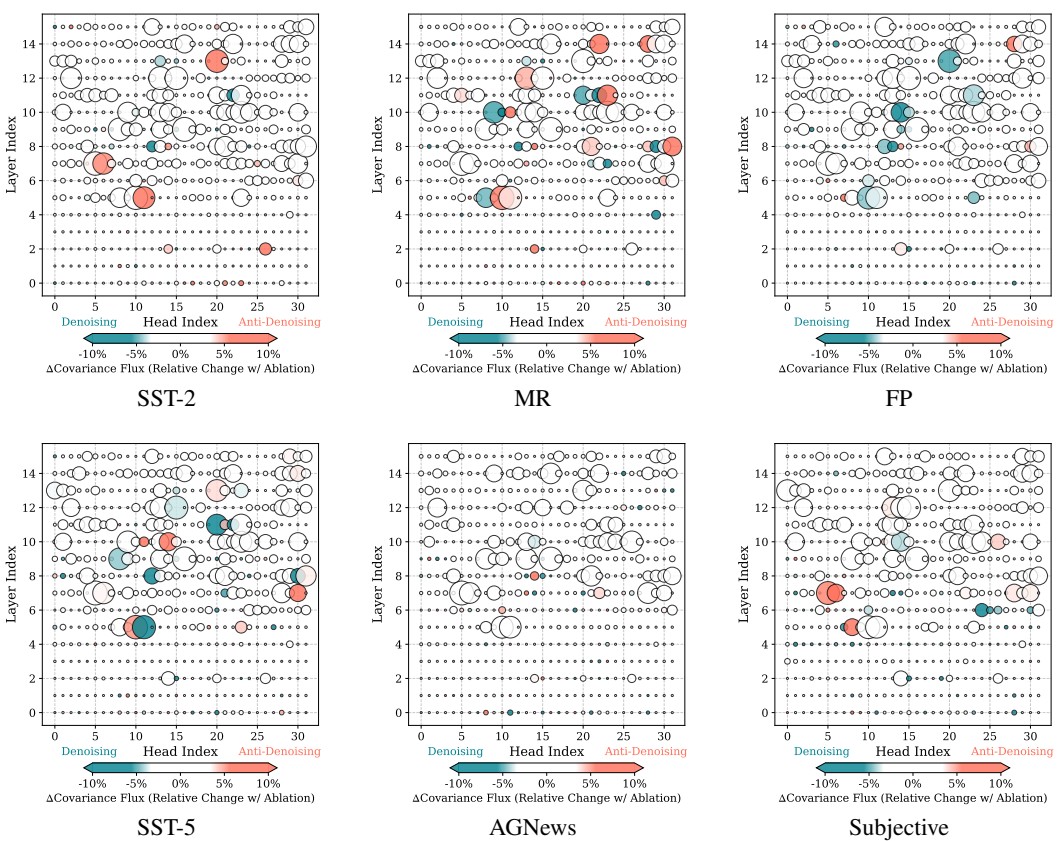

Figure 87: Augmentation results for Fig. 9 on Llama 3.2-1B.

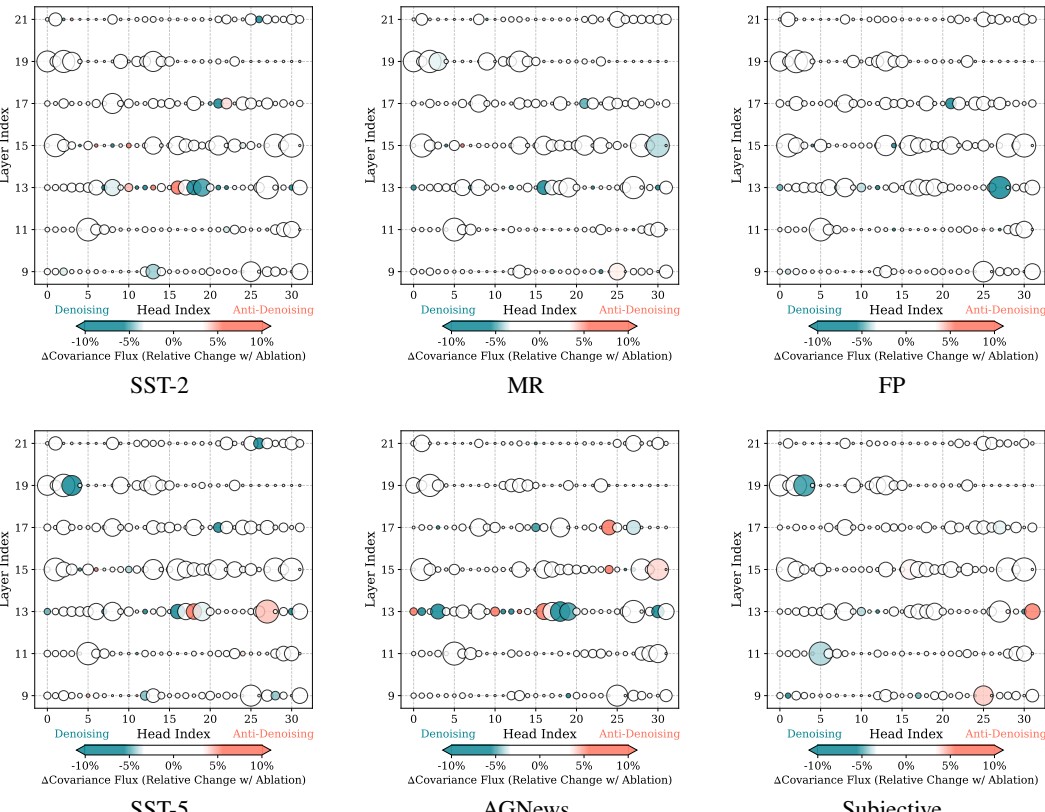

Figure 88: Augmentation results for Fig. 9 on Llama 3-8B.

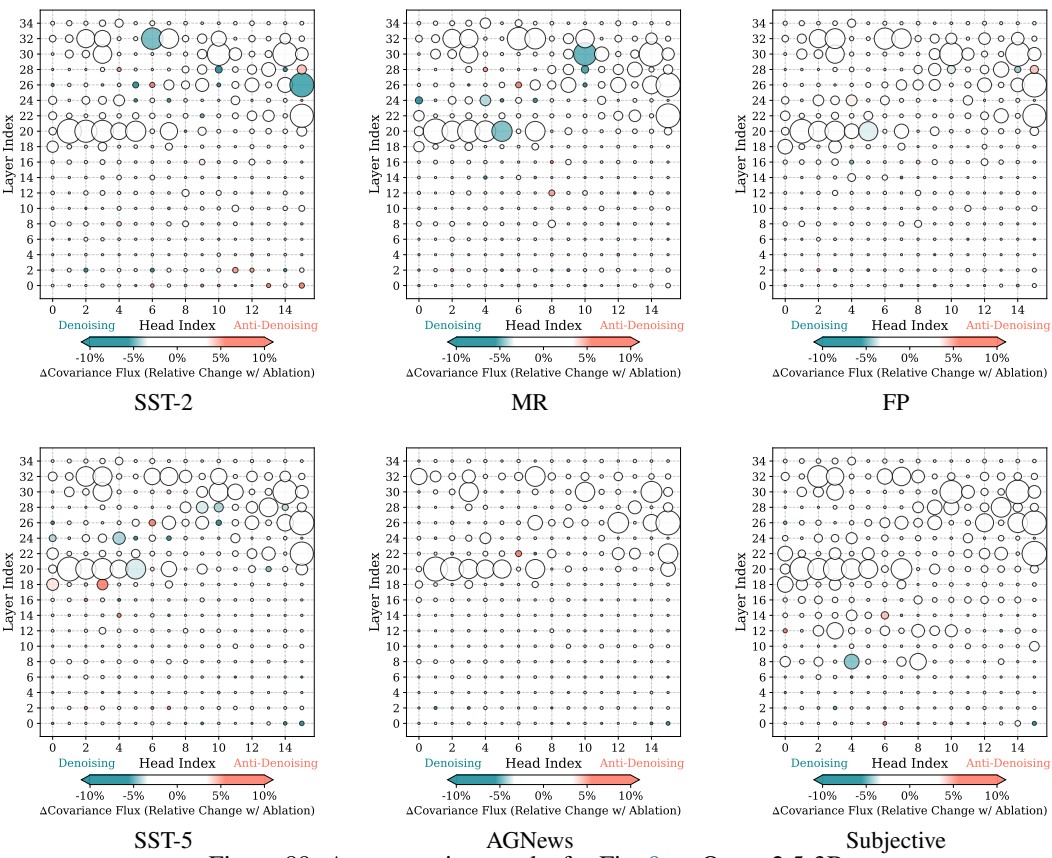

Figure 89: Augmentation results for Fig. 9 on Qwen 2.5-3B.

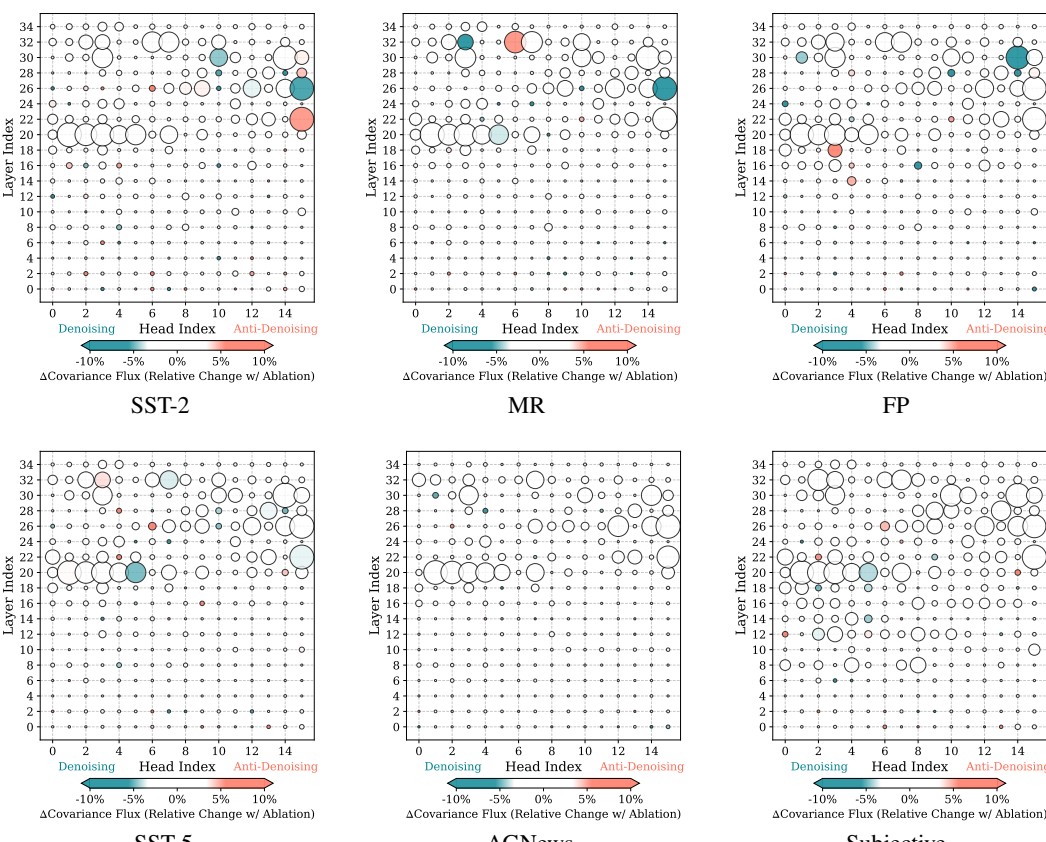

Figure 90: Augmentation results for Fig. 9 on Qwen 2.5-3B Instruct.

