# OpenReview forum: "Mechanism of Task-oriented Information Removal in In-context Learning"
_ICLR.cc/2026/Conference — ICLR 2026 Poster_

### Official Review · Reviewer_sEGe · 2025-10-31

**Soundness:** 3
**Presentation:** 3
**Contribution:** 2
**Rating:** 4
**Confidence:** 4

**Summary:**

The paper argues that few-shot in-context learning (ICL) works by selectively removing task-irrelevant information from query representations, steering the model toward the intended task. In zero-shot settings, injecting a low-rank "task-verbalization" filter that projects onto a Task-Verbalization Subspace (TVS) dramatically improves accuracy even while preserving only ~0.7% of dimensions, while in few-shot ICL the model spontaneously moves hidden states toward this TVS, as measured by geometric metrics that rise in middle-to-late layers. The paper identifies "Denoising Heads" (DHs)—attention heads whose ablation disrupts this removal operation—which are largely independent of induction heads and show local re-encoding patterns over query tokens; ablating DHs significantly reduces ICL accuracy and nearly collapses performance in unseen-label settings, demonstrating the causal importance of this mechanism. The authors note this mechanism applies to clustering-style classification tasks but not bijective tasks like translation, with experiments conducted across multiple text classification datasets and models including LLaMA and Qwen variants.

**Strengths:**

1. Task‑oriented information removal operationalized via a low‑rank subspace and two geometric metrics that correlate with accuracy and layer depth.

2. identification and causal ablations of Denoising Heads, with independence from induction heads and a sensible local re‑encoding attention pattern.

3. Explicit failure case on bijective tasks (translation/fact recall) and an explanation of why low‑rank removal should not help there.

4. Broad, careful experiments across six datasets and three models, with appendix replications and unseen‑label analyses that stress‑test induction‑only stories.

**Weaknesses:**

1. Theory is largely heuristic. The link between covariance and “information removal” is argued but not formally proved; metric choices and DH thresholds (±3.5%) are somewhat ad hoc.

2. Narrow task family. The focus is single‑label classification; reasoning, chain-of‑thought, or generation settings are not studied, and the paper itself notes the mechanism likely does not apply to bijective mappings.

3. Scale and generality. Only up to 8B models are used. It remains unclear whether DH distributions and metric trends stabilize for larger base or instruction‑tuned LMs.

**Questions:**

1. How sensitive are results to scanning all layers/heads? Could you provide one full‑layer run on a smaller model to confirm no hidden DH clusters?

2. Do your metrics or DH patterns say anything in multi‑class generative outputs (e.g., natural language rationales)? If not, what would be needed?

3. Any preliminary results on larger models (e.g., 13B/34B) or instruction‑tuned variants to test whether DHs persist or shift?

---

> ### Author Response · Authors · 2025-11-21
>
> Thank you for your insightful review of our paper. We greatly appreciate the thoroughness of your comments and the helpful suggestions, which will certainly strengthen our paper. Below, we provide our responses to the points you have kindly raised.
>
> ---
>
> **Weakness 1.1: The Link between Covariance and Information Removal**
>
> Thank you for pointing this out. The idea that “covariance is an information measure” is one of the basic intuitions in information theory, and it is the foundation for utilising covariance to quantify information. To make this explicit, we added a proof of this intuition in the revised version. Specifically, we show that the eigenvalues of the covariance matrix of the hidden-state set are the entropy components along each principal direction. **Please see Revision Note 4 for details.**
>
> **Weakness 1.2: Fixed thresholds**
>
> We understand your concern. However, as mentioned in the paper, our goal is not to identify all DHs, but to use them as a handle for testing the causal relationship between information removal and ICL performance. Therefore, we only need to ensure that the chosen threshold is broadly permissive, meaning that any mild threshold can demonstrate the significance of DH ablation relative to random ablation.
>
> To address this point, we added an experiment showing that the choice of threshold is very flexible: the results remain robust across a mild range of −20% to 0%. **Please refer to Revision Note 3.**
>
> **Weakness 2.1; Question 2: Generative Settings**
>
> We appreciate your perspective. We have added a discussion of the generative setting in the paper, **please refer to Revision Note 6**.
>
> Specifically, in the translation task (noting that translation is not a bijective task), we observed an increase in eccentricity consistent with classification tasks. However, as we acknowledge in the revision, training a linear filter on 0-shot data to examine covariance flux and identify denoising heads is difficult, and likely requires a nonlinear filter. Therefore, this remains a preliminary observation, indicating that our framework is still effective in generative settings.
>
> **Weakness 2.2: Bijective Tasks**
>
> Thank you for pointing this out. However, we would like to offer a different view. As discussed in the paper, prior work has shown that such bijective tasks are special cases [1], so separating them from our framework is acceptable. We consider that the ICL mechanism for bijective tasks is more likely based on extracting signal vectors from the ICL demonstrations to recall “knowledge” within the MLP module [2], which is entirely different from the mechanism we study.
>
> [1]. Y Dong, et al., Understanding Task Vectors in In-Context Learning: Emergence, Functionality, and Limitations.
> [2]. D Dai, et al., Knowledge Neurons in Pretrained Transformers. ACL 2022.
>
> **Weakness 3 / Question 3: Experiment on More Models**
>
> Thank you for pointing this out. We have conducted experiments on the 3B Instruct and 13B models, **please refer to Revision Note 8.**
>
> Interestingly, on instruct-tuned models, DHs show a weaker effect in the Seen Label setting, but still perform significantly in the Unseen Label setting. A possible reason is that Instruct tuning may have strengthened the induction circuits, giving them sufficiently large effects that overshadow the DHs in the Seen Label scenario. However, due to their inherent limitation (i.e., they can only copy labels), induction circuits still fail to contribute significantly in the Unseen Label scenario, allowing DHs to dominate.
>
> **Question 1: Full Scan**
>
> Note that we originally conducted a full scan on Llama 3.2-1B. To make this clear, we added a description of the scanned layers in Table 2. **Please refer to Revision Note “Other minor revisions”.**
>
> ---
>
> Once again, thank you for the time you have devoted to our paper and for your excellent review. It has greatly improved the quality of our work. We look forward to further discussion with you. If you feel that your concerns have been satisfactorily resolved, we would be grateful if you could consider this when updating your evaluation.

---

> > ### Comment · Reviewer_sEGe · 2025-11-24
> > **Most of my concerns are addressed**
> >
> > Thank you for the detailed response. The revisions have addressed most of my concerns.
> >
> > The added proof linking covariance to entropy components strengthens the theoretical grounding considerably. And the threshold robustness experiments shows that the proposed method is not sensitive to hyperparameters.
> >
> > The generative experiments addressed my major concern, showing that proposed TVS hypothesis can be generalized to boarder applications beyond classification tasks.
> >
> > Additional  experiments on 13B and instruction-tuning models are also interesting and helpful. I'd love to see more discussions on the differences of DH on the base models vs instruction-tuning models.
> >
> > I'm satisfied with the revisions and have raised my score to 8.

---

> > > ### Author Response · Authors · 2025-11-25
> > >
> > > Thank you very much for your thoughtful comments and recognition of the improvements in our revisions. We sincerely appreciate your constructive suggestions, which have greatly enhanced this paper. Your positive feedback and support have greatly encouraged us. Thank you again for your time and contribution to our work, we will thank all the reviewers in our acknowledgements.

---

### Official Review · Reviewer_5eyE · 2025-11-01

**Soundness:** 4
**Presentation:** 2
**Contribution:** 4
**Rating:** 10
**Confidence:** 4

**Summary:**

This paper investigates the internal mechanism of In-Context Learning (ICL) in large language models (LLMs) from a new perspective — task-oriented information removal. Instead of viewing ICL as pattern imitation or label copying via Induction Heads, the authors argue that few-shot demonstrations act as filters that suppress task-irrelevant information in the hidden representations.

To support this view, they introduce two novel probing metrics: Covariance Flux through the Task-Verbalization Subspace (TVS) and Eccentricity, which quantify how hidden-state variance aligns with task-relevant directions and how anisotropic (filtered) the representations become. Using these tools, they demonstrate that (1) LLMs spontaneously recover the TVS during ICL, (2) this process can be measured as an information-removal trajectory across layers, and (3) a new class of attention heads, Denoising Heads, drive this mechanism, overcoming the well-known limitation of Induction Heads when facing unseen labels.

**Strengths:**

1. **Novel probing metrics with clear interpretability.**
The introduction of Covariance Flux through TVS and Eccentricity provides a fresh, quantitative lens for studying internal representation dynamics of LLMs. These metrics go beyond uncovering new ICL mechanisms and identifying denoising heads, enabling a measurable understanding of how information filtering evolves within transformer layers.

2. **New mechanism of ICL: task-oriented information removal.**
This paper shifts the focus of ICL mechanism from token-copying to information filtering, which deepens our conceptual understanding of ICL.

3. **Identification of Denoising Heads addressing the Induction-Head limitation.**
The discovery of Denoising Heads is an insightful contribution. These heads operate locally and semantically, filtering query representations rather than copying lexical patterns, thereby explaining why LLMs can generalize to unseen labels -- something Induction Heads cannot do.

4. **Strong empirical grounding.**
The experiments are extensive and solid.

**Weaknesses:**

**Overly dense and somewhat disorganized presentation.**
While the findings are rich, the paper feels overly packed. The main sections attempt to cover TVS analysis, metric design, mechanistic ablations, and head identification all at once. The narrative flow occasionally sacrifices clarity for compactness, making it harder for readers to see the conceptual through-line.

**Questions:**

Suggestion:

 **Center the paper on the probing methodology.**

If the authors had focused the paper around the two new metrics (Covariance Flux and Eccentricity) as a general probing framework, the contribution would appear sharper and of broader relevance. The ICL mechanism and Denoising Head discovery could then be presented as compelling applications of this framework, rather than parallel findings competing for attention.

**A more layered structure** -- with detailed exposition of the probing metrics in the main text and the some extended empirical findings moved to the appendix -- would make the paper both clearer and more impactful. You can organize the paper similar to [1].

Minors:

1. In Figure 3, I understand that the marker represents the lower bound of information removal with a value between 0 and 1. However, please include at least one reference marker (for example, the marker size corresponding to no information removal) so that readers can intuitively gauge the approximate lower bound portion of information removed.

[1] Ren, Yi, and Danica J. Sutherland. "Learning dynamics of llm finetuning." arXiv preprint arXiv:2407.10490 (2024).

---

> ### Author Response · Authors · 2025-11-21
>
> Thank you very much for your strong endorsement of our paper; it is truly encouraging for us. Thank you! Also, we greatly appreciate the thoroughness of your comments and the helpful suggestions, which will certainly strengthen our paper. Below, we provide our responses to the points you have kindly raised.
>
> ---
>
> **Weakness / Question 1: Overly dense and somewhat disorganized presentation**
>
> We fully agree that the presentation feels dense. This was primarily due to the 9-page limitation at ICLR, we had to stuff our massive experiment under a dense presentation (currently, even a single new line will cause a new page). Fortunately, in the discussion phase, we are allowed to use the 10th page, which enables us to improve the narrative flow and clarity.
>
> In detail, following your suggestion, we unified the experimental methods into a new Section 3 and all experiment results into Section 4. **Please refer to Revision Note 1 for details**. This restructuring improves the clarity and logical flow of the paper. Thank you for your excellent suggestion!
>
> **Question 2: Scatter Size in Fig. 3**
>
> Thank you for your careful observation. In the revision, we directly supplemented the numerical results (**please refer to Revision Note 7**), we believe that the numerical results are more direct.
>
> ---
>
> Once again, thank you for the time you have devoted to our paper and for your excellent review. It has greatly improved the quality of our work. We look forward to further discussion with you.

---

> > ### Comment · Reviewer_5eyE · 2025-11-24
> >
> > Thank you for your comments. I am glad my suggestions are helpful for your revision. The revised paper now presents a clearer line of logic. Thank you for your excellent work!

---

> > > ### Author Response · Authors · 2025-11-25
> > >
> > > Thank you very much for your encouraging reply. We truly appreciate your suggestion, which has greatly improved the organization of the paper. Thank you again for your support and for helping us strengthen the work. We will thank all the reviewers in our acknowledgements.

---

### Official Review · Reviewer_TE1T · 2025-11-03

**Soundness:** 4
**Presentation:** 4
**Contribution:** 3
**Rating:** 10
**Confidence:** 2

**Summary:**

The paper proposes a novel interpretation of the ICL mechanism by viewing it as a process to remove redundant information in the query’s hidden states for the targeted tasks. Specifically, the paper applies a low-rank filter on the hidden states for informational removal and finds it boost the performance on the designed tasks significantly. On top of that, the paper conducts further analysis to prove that a few-shot ICL implicitly performs a similar information removal process. Finally, the paper identifies attention heads that are responsible for such removal behaviors and shows that they are crucial for the success of ICL.

**Strengths:**

- The paper includes well-designed illustrations that well support the claims.
- The paper provides a perspective for demystifying ICL that is novel yet aligns with many previous observations.
- To support the claim that ICL performs information removal, the paper provides decently comprehensive analyses that dives deep into the activities of the hidden states of the model using self-crafted analysis methods.
- The paper includes experiments and discussions to compare their newly discovered Denoysing Heads against the previous induction heads.

**Weaknesses:**

The paper provides many insights on the mechanism of ICL, but the potential future directions built on these findings are less clear.

**Questions:**

What are possible sources of the accuracy improvement from instruction given that instructions do not seem to contribute to the information removal process?

---

> ### Author Response · Authors · 2025-11-21
>
> Thank you very much for your strong endorsement of our paper; it is truly encouraging for us. Thank you! Also, we greatly appreciate the thoroughness of your comments and the helpful suggestions, which will certainly strengthen our paper. Below, we provide our responses to the points you have kindly raised.
>
> ---
>
> **Weakness: Potential future directions**
>
> Thanks for pointing this out. With the help of reviewer M6i4’s excellent suggestion, we proposed a method that amplifies the attention head to enhance the ICL performance (**Please refer to Revision Note 2**). We believe this may represent a promising paradigm: improving model controllability through mechanistic understanding, for which we have provided a prototype. We consider this a novel and exciting direction.
>
> **Question: Possible sources of the accuracy gain from instructions**
>
> Thanks for your valuable question. We can offer some conjectures here: the instruction may inject a certain amount of direct logits steering into the residual stream, causing the output components aligned with this instruction to be amplified. For example, an instruction “Please predict the sentiment of this sentence” could bias the output logits to “Positive” and “Negative”, by (maybe) some matching to the training-time distribution, without effect on the information content of the query. Therefore, we may infer that this mechanism is likely a form of direct amplification. Further investigation will need to be carried out in future work.
>
> ---
>
> Once again, thank you for the time you have devoted to our paper and for your excellent review. It has greatly improved the quality of our work. We look forward to further discussion with you.

---

### Official Review · Reviewer_M6i4 · 2025-11-03

**Soundness:** 3
**Presentation:** 3
**Contribution:** 3
**Rating:** 6
**Confidence:** 2

**Summary:**

This paper proposes a new mechanistic interpretation of in-context learning: rather than copying label tokens via induction heads, large language models perform a task-oriented information removal that compresses query hidden states toward a low-rank, task-verbalization subspace (TVS). The authors demonstrate (1) that injecting a trained low-rank filter into a layer’s residual stream converts near-zero zero-shot performance into strong task-specific outputs; (2) that few-shot demonstrations induce geometric changes in hidden states — increased eccentricity and covariance flux into the learned TVS — consistent with implicit information removal; and (3) that a subset of attention heads  causally contribute to this aligning operation: ablating them reduces covariance flux, eccentricity, and downstream accuracy. The paper thus argues DHs complement induction heads and together explain a broader range of ICL phenomena.

**Strengths:**

The paper’s framing of the “information removal → task-verbalization subspace” is conceptually novel and shifts the focus from mere copying to selective suppression

The paper is well-written and easy to follow. The low-rank injection experiment is simple, interpretable, and effectively demonstrates that a small learned linear filter can steer outputs.


The identification of Denoising Heads (DHs) and the analysis of their interaction with induction heads provide a richer mechanistic understanding of in-context learning.

**Weaknesses:**

Ablating a head is a valid intervention, but zeroing a head output changes the residual stream and thus all downstream activations; the observed drops in Covariance Flux / Eccentricity / Accuracy may partly reflect these propagated network dynamics rather than a head performing a localized denoising operation.

The method appears to identify DHs only on a subset of layers. If many DHs remain unidentified, ablation results could be underestimated or misattributed.


The DH identification uses fixed relative-change thresholds (e.g., −3.5% / −5%). It is unclear how sensitive results are to threshold choice and whether effects are statistically significant across random seeds / datasets / prompts.

**Questions:**

Can you show that amplifying DH outputs (scale >1) increases Covariance Flux/Eccentricity and improves accuracy in settings where the model is weak (e.g., few-shot with noisy labels)?

---

> ### Author Response · Authors · 2025-11-21
>
> Thank you for your insightful review of our paper. We greatly appreciate the thoroughness of your comments and the helpful suggestions, which will certainly strengthen our paper. Below, we provide our responses to the points you have kindly raised.
>
> ---
>
> **Weakness 1: Long effects of zeroing attention heads**
>
> We believe what you intended to say is that the information removal is mediated by a circuit, so ablating any head within it will influence the hidden states on the layer the circuit operates. We appreciate your excellent intuition, but please notice that: the change of covariance flux and eccentricity (Fig. 7 and similar figure families in the Appendix) is evaluated immediately, i.e., on the hidden states right after the ablation is done, therefore, we in fact only detect the terminal effect produced by the last head in this propagation chain. Any upstream circuit before this head, if exists, cannot be detected because it does not immediately induce geometric effects on the hidden states.
>
> **Weakness 2: Identify DHs only on a subset of layers**
>
> We agree with you. Our experiment setting on some larger models is scanning only some layers for the DHs, so that the DHs are under-localized, i.e., some of the DHs are not identified. Therefore, the ablation experiments based on such under-localization can cause weaker results than on a fully localized DH set. However, as we have more clearly emphasized in the revised version, our goal is not to identify all DHs, but to use them as a handle for testing the causal relationship between information removal and ICL performance.
> That is, in a word, if ablating only a subset of heads is sufficient to produce a clear causal effect, then it can even more strengthen our conclusion. And, as shown in Table 2, even weaker-than-normal results are still significant, which further confirms the causal relationship between DH and ICL accuracy.
>
> **Weakness 3: Fixed thresholds**
>
> We understand your concern. However, as noted earlier, our goal is not to identify all DHs, but to use them as a handle for testing the causal relationship between information removal and ICL performance. Therefore, we only need to ensure that the chosen threshold is broadly permissive, meaning that any mild threshold can demonstrate the significance of DH ablation relative to random ablation.
> To address this point, we added an experiment showing that the choice of threshold is very flexible: the results remain robust across a mild range of −20% to 0%. **Please refer to Revision Note 3.**
> In addition, following your suggestion, we have included a statistical test to evaluate the probability that the DH set could be selected by chance. **Please refer to Revision Note 5.**
>
> **Question 1: Head output amplifying**
>
> We appreciate your excellent intuition. We have added an application prototype that enhances ICL performance by amplifying the DH outputs. **Please refer to Revision Note 2.**
>
> ---
>
> Once again, thank you for the time you have devoted to our paper and for your excellent review. It has greatly improved the quality of our work. We look forward to further discussion with you. If you feel that your concerns have been satisfactorily resolved, we would be grateful if you could consider this when updating your evaluation.

---

### Author Response · Authors · 2025-11-21
**Revision Note 2025/11/20 (AOE)**

Dear PCs, SACs, ACs, and Reviewers,

We appreciate the reviewer’s overall positive feedback, which is greatly encouraging to us. **To address the valuable comments raised by the reviewer, we have made the following adjustments to the paper.** (We have highlighted the **technical** revisions in red.)

1. **[Reviewer 5eyE: Paper Writing Framework]** Thank you to reviewer 5eyE for the wonderful writing suggestion. We consider the suggested writing framework elegant and have therefore revised the narrative structure of the paper accordingly. Specifically:
   1. We created a new Section 3 to unify the previously scattered descriptions of the experiment methodology and to establish a clearer logical progression of these methods.
   2. All experimental results have been moved to Section 4.
   3. In addition, to accommodate this restructuring, the previous Section 2.2 (Experiment Settings) has been moved into Section 4, rather than remaining in Section 2. Section 2 is now renamed “Background” and is used solely for related works.
   4. We also moderated the tempo of the paper, and added brief recalls of methodological details in the experiment section, so that we can help readers understand more easily, and do not need to jump back to Section 3 while reading Section 4.
   5. The "contribution summary" part has been rewritten.
   This revision is almost entirely a replacement of paragraphs (with minor adjustments to transitional passages that are not central to the paper), and the technical content of the paper remains unchanged.

2. **[Reviewer M6i4, TE1T: Application & Amplifying the DH]** We appreciate the reviewer’s excellent intuition. We have added an application prototype that enhances ICL performance by amplifying the DH outputs. This serves as a frugal and effective method prototype, especially in scenarios where labels contain noise. This addition has been included at the end of Section 4.4 (page 9), and the corresponding results are shown in Figure 14.

3. **[Reviewer M6i4, sEGe: Sensitivity of the Ablation Threshold]** (Appendix E) We added a discussion on the sensitivity of the ablation threshold, which confirms that any non-extreme threshold can reliably distinguish DHs from random heads. A footnote describing this result has also been added to the main text.

4. **[Reviewer sEGe: Mathematical Grounding of Metrics]** (Appendix G) We added a mathematical proof for using covariance as an information measure. Specifically, we show that the covariance loading on principal component directions serves as an equivalent measure of entropy.

5. **[Reviewer M6i4: Statistical Significance]** (Table 2) We have added the p-values of the ablation results of DH against RH. Specifically, these values represent the p-test probability of accepting the hypothesis that “the DHs are in a randomly sampled attention head set.”

6. **[Reviewer sEGe: Generative Tasks]** (Appendix G) We added a discussion on generative tasks. Through eccentricity-based tests, we provide prototypical evidence that information removal also occurs in the English–Chinese translation task. Also, we include an additional limitation section concerning the complexity of the information filter.

7. **[Reviewer 5eyE: Numerical Results for Covariance out of Rank]** (Appendix I) We added numerical results for Covariance out of Rank.

8. **[Reviewer sEGe: More Models]** We added experimental results for the 3B Instruct and 13B Instruct models. These results are broadly consistent with our main findings. Please see Table 2 and the Appendix I for details.

---

**Other minor revisions:**

- Minorly revised Fig. 1 and Fig. 16 for aesthetic and typo.
- Revised Sec. 4.1 and Table 5 since we added a new model.
- Section 4.4 now specifies the number of demonstrations used.
- Table 2 includes an added note regarding the range of scanned layers.
- Layout adjustments corresponding to the above revisions, including updating the placement and size of several figures and tables, as well as adjusting some hyperlinks and footnotes.
- Several typos, wording issues, and grammatical errors have been corrected.
- [2025/11/21 ~03:00 AOE] Fixed one figure import error.
- [2025/11/24 ~05:00 AOE] Very minor revision: Line 185: corrected the model amount.
- [2025/11/30 ~01:00 AOE] Revised some typos.

Again, thank you for the time and thoughtful reviews you devoted to this paper. Your efforts have made this paper even stronger.

Authors

---

### Author Response · Authors · 2025-11-29
**Review and Reviewer-Author Discussion Summary (1/2)**

Dear PCs, SACs, ACs, and Reviewers,

Thank you very much for your valuable contributions to our work. To assist the newly assigned AC and help reduce their workload, we provide below a summary of the key points from the reviews and the reviewer-author discussions.

---

**Strength.** Overall, we are grateful that the reviewers gave this paper a highly positive evaluation in the initial reviews. Specifically:

- **This paper proposes a new mechanism for explaining ICL based on a novel hypothesis, extending our understanding of ICL.**

  All four reviewers recognized this point (M6i4: Strength 1, TE1T: Strength 2, 5eyE: Strength 2, sEGe: Strength 1).

- **With reliable and interpretable methods (low rank injection and two metrics), this paper verifies the existence of the proposed mechanism.**

  After we revised the paper according to reviewer sEGe's suggestions, all four reviewers recognized this point (M6i4: Strength 2, TE1T: Strength 3, 5eyE: Strength 1, sEGe: Official Comment).

- **This paper identifies a new type of attention heads (Denoising Head) that conduct the mechanism above, addressing the limitation that traditional induction-head-based explanations cannot generalize to unseen label settings.**

  All four reviewers recognized this point (TE1T: Strength 4, M6i4: Strength 3, 5eyE: Strength 3, sEGe: Strength 2).

- **The experiments are carefully designed, extensive, and solid.**

  Three reviewers explicitly highlighted this (TE1T: Strength 3, 5eyE: Strength 4, sEGe: Strength 4).

- **The paper is clearly written and easy to follow.**

  After we revised the writing following reviewer 5eyE's suggestions, three reviewers (M6i4: Strength 2, TE1T: Strength 1, 5eyE: Official Comment) highlighted this.

---

> ### Author Response · Authors · 2025-11-29
> **Review and Reviewer-Author Discussion Summary (2/2)**
>
> **Concerns and Our Addressing.** During the discussion period, we actively addressed the reviewers' concerns, which were recognized by the reviewers and led to increased scores. Specifically:
>
> - **Concerns about experiment design.**
>
>   - (M6i4: Weakness 3, sEGe: Weakness 1.2) **Using fixed thresholds to identify and ablate denoising heads.**
>
>     **Our Addressing.** We added experiments in **Revision 3** showing that any reasonable threshold leads to correct results. Please refer to our responses to reviewers M6i4 and sEGe for details. Reviewer sEGe explicitly acknowledged our efforts and considered this weakness resolved.
>
>   - (sEGe: Weakness 2, 3) **Task, scale, and instruction-tuned models.** The original experiments are on smaller models, classification tasks, and a comparison against an instruction-tuned model is needed.
>
>     **Our Addressing.** We added experiments in **Revision 6** and **Revision 8** on (a) a generative task, (b) a larger model, (c) an instruction-tuned model and obtained consistent results. Reviewer sEGe explicitly acknowledged our efforts and considered this weakness resolved.
>
>   - (M6i4: Weakness 1, 2) **Concerns about ablation setup.** (a) Ablation analysis may not directly identify the heads that conduct the hidden-state denoising writing operation, but instead may locate intermediate nodes that transmit this process. (b) Ablation analysis is not a full scan of every layer.
>
>     **Our Addressing.** We provided a detailed explanation of our experimental setup and clarified that (a) our method can precisely identify the heads that perform the denoising writing operation, and (b) we did not scan all layers due to computational constraints. However, this setup, where only a subset of heads is identified and ablated, actually makes our conclusions more reliable, since we obtain strong results under weaker experiment conditions. Please refer to our responses to reviewers M6i4 for details.
>
> - **Presentation issues.**
>
>   - (sEGe: Weakness 1) **The intuition from our metrics to “information removal” is not obvious.**
>
>     **Our Addressing.** We added a mathematical proof that directly links our metric (the eigenvalues of the covariance matrix) to information entropy in **Revision 4**. Reviewer sEGe explicitly acknowledged our efforts and considered this weakness resolved.
>
>   - (5eyE: Weakness) **Overdense writing style.**
>
>     **Our Addressing.** As shown in **Revision 1**, we reorganized the structure of the paper. Reviewer 5eyE explicitly acknowledged our efforts and considered this weakness resolved.
>
>   - (5eyE: Question, M6i4: Weakness 3) **More descriptive statistics and visualization results.**
>
>     **Our Addressing.** As shown in **Revision 5 and 7**, we added more intuitive descriptive statistics and visualization results.
>
> - (TE1T: Weakness, M6i4: Question) **Lack of application, try to amplify the Denoising Head.**
>
>     **Our Addressing.** We added experiments in **Revision 2** showing that amplifying the denoising head can enhance ICL accuracy, serving as an effective and efficient application prototype based on our findings.
>
> **Recognition of our revision from reviewers.** Two of the four reviewers (5eyE, sEGe) explicitly acknowledged that most of their concerns had been addressed, and reviewer sEGe raised the score on 25 Nov (4→8). We believe that we have also properly addressed the remaining two reviewers' concerns.
>
> ---
>
> Above, we have faithfully summarized all reviewer comments and our corresponding responses, hoping that this will assist the AC's work. We are deeply grateful to the reviewers, AC, SAC, and PC, for their dedicated effort and excellent work. Their insightful feedback has further strengthened our paper. The authors offer their sincere respect and appreciation to all involved!
>
> Sincerely,
> Authors

---

### Meta-Review · Area_Chair_degJ · 2025-12-25

**Summary:**

This submission proposes a mechanistic view of in-context learning (ICL) as task-oriented information removal: few-shot demonstrations act as implicit filters that suppress task-irrelevant information in query representations. The paper introduces two probing metrics (Covariance Flux through a Task-Verbalization Subspace and Eccentricity), and identifies a class of “Denoising Heads” whose ablation degrades these metrics and ICL accuracy. Reviewers broadly agree the empirical study is extensive and the framing is interesting; the main concerns focused on (i) whether the metric-to-“information removal” interpretation is sufficiently grounded, (ii) robustness and identification choices for Denoising Heads (e.g., thresholds / partial layer scans), and (iii) generality beyond the evaluated task family.


The paper provides a compelling mechanistic hypothesis, introduces useful probing metrics, and supports the claims with extensive experiments and causal interventions. While some generalization and causal-localization limitations remain, the rebuttal and revisions substantially strengthened the methodological grounding and robustness.

**Reviewer Concerns:**

### Concerns addressed by the rebuttal/revision

- **Threshold / robustness for DH identification:** The authors added analyses showing results are robust across reasonable thresholds.

- **Grounding the metric interpretation:** The authors added a proof/derivation linking their covariance-based metric to information entropy.

- **Generality checks (tasks/models/instruction-tuned):** Additional experiments on a generative task, larger models, and instruction-tuned models were added, with consistent trends.

- **Actionability / “application” angle:** They added an experiment showing that amplifying denoising heads can improve ICL performance.

- **Presentation clarity:** The dense writing concern was addressed via restructuring and clearer organization, acknowledged by a reviewer.


### Remaining / partially outstanding concerns

- **Scope / generalization:** The mechanism is still mainly validated on a particular family of tasks; broader behaviors (e.g., more complex reasoning/generation regimes) remain less explored.

- **Causal localization limitations:** Some reviewers raised that ablation effects can propagate and DH sets may be underestimated when scanning only subsets of layers; these remain practical limitations even if the main causal story is plausible.

**Reviewer Scores:**

- TE1T: Likely unchanged at 10, but I discount this score in deliberation because the reviewer self-reports low confidence.

- 5eyE: Likely remains 10; the review is detailed and the main weakness (presentation density) was addressed by restructuring.

- sEGe: Per the discussion summary, this reviewer explicitly increased the score from 4 to 8 after revisions.

- M6i4: Originally 6 with concerns about ablation/threshold sensitivity; given the added robustness checks and statistical tests, I expect a modest increase (e.g., to 8) or staying at 6 but with reduced concern.

---

### Decision · Program_Chairs · 2026-01-26

Accept (Poster)